# Visualizing the disordered nuclear transport machinery in situ

Miao Yu[1,2,3,9], Maziar Heidari[4,9], Sofya Mikhaleva[1,2,3,9], Piau Siong Tan[3,9], Sara Mingu[1,2], Hao Ruan[1,2], Christopher D. Reinkemeier[1,2,3], Agnieszka Obarska-Kosinska[5], Marc Siggel[4,7,8], Martin Beck[5], Gerhard Hummer[4,6 ✉] & Edward A. Lemke[1,2 ✉]

The approximately 120 MDa mammalian nuclear pore complex (NPC) acts as a gatekeeper for the transport between the nucleus and cytosol[1]. The central channel of the NPC is filled with hundreds of intrinsically disordered proteins (IDPs) called FG-nucleoporins (FG-NUPs)[2,3]. Although the structure of the NPC scaffold has been resolved in remarkable detail, the actual transport machinery built up by FG-NUPs— about 50 MDa—is depicted as an approximately 60-nm hole in even highly resolved tomograms and/or structures computed with artificial intelligence[4–11]. Here we directly probed conformations of the vital FG-NUP98 inside NPCs in live cells and in permeabilized cells with an intact transport machinery by using a synthetic biology-enabled site-specific small-molecule labelling approach paired with highly time-resolved fluorescence microscopy. Single permeabilized cell measurements of the distance distribution of FG-NUP98 segments combined with coarse-grained molecular simulations of the NPC allowed us to map the uncharted molecular environment inside the nanosized transport channel. We determined that the channel provides—in the terminology of the Flory polymer theory[12]—a 'good solvent' environment. This enables the FG domain to adopt expanded conformations and thus control transport between the nucleus and cytoplasm. With more than 30% of the proteome being formed from IDPs, our study opens a window into resolving disorder–function relationships of IDPs in situ, which are important in various processes, such as cellular signalling, phase separation, ageing and viral entry.

IDPs are flexible, dynamic macromolecules that lack a fixed tertiary structure and can adopt a range of conformations to perform various functions across the cell. IDPs are highly relevant for human physiology and have central roles, among others, in neurodegenerative ageing diseases and cancer. IDPs are also key players in phase separation and are involved in the formation of biomolecular condensates[13–21]. In the nanosized NPC, which has a total molecular weight of approximately 120 MDa in mammals, there are hundreds of IDPs enriched in phenylalanine (F) and glycine (G) residues, known as FG-NUPs[1]. The FG-NUPs form a permeability barrier in the central channel of the NPC, which regulates nucleocytoplasmic transport by restricting the passage of large cargo unless it presents a nuclear localization sequence or a nuclear export sequence[2,3]. Nuclear transport receptors can specifically recognize these sequences and efficiently shuttle the cargo through the barrier. With recent advances in cryo-electron tomography, crystallography, proteomics and artificial intelligence (AI)-based structure prediction, approximately 70 MDa of the NPC scaffold enclosing the central channel has been resolved with near-atomic resolution[4–11]. However, signals from the highly dynamic FG-NUPs are by and large not accessible to those structural biology techniques, and the actual transport machinery inside the central channel—another approximately 50 MDa— is not captured, leaving an approximately 60-nm hole in the centre of the scaffold structure. Consequently, the protein conformational state inside the NPC remains elusive, which has led to several partially conflicting hypotheses for the morphologies of the FG domains in their functional state[22–28]. With approximately 30% of the entire eukaryotic proteome being intrinsically disordered, the problem that the conformational state is not easily studied in cells extends far beyond NPC biology. Besides magnetic resonance and scattering techniques[13,14], single-molecule fluorescence of purified and labelled proteins has become a powerful tool for probing the conformations of proteins in solution; advanced studies have even shown that this is possible in cells if such probes are microinjected[29–31]. However, the NPC is assembled only in late mitosis and during nuclear growth in interphase[32], and its labelling thus requires genetic encoding. Established fluorescent protein-based technologies such as GFP or self-labelling protein tags such as SNAP-tag[33], however, do not readily enable the extraction of multiple distance distributions for the same protein, owing to the sheer size of the fluorescent label and the inherently limited freedom of labelling.

[1]Biocenter, Johannes Gutenberg University Mainz, Mainz, Germany. [2]Institute of Molecular Biology Mainz, Mainz, Germany. [3]Structural and Computational Biology, European Molecular Biology Laboratory, Heidelberg, Germany. [4]Department of Theoretical Biophysics, Max Planck Institute of Biophysics, Frankfurt am Main, Germany. [5]Department of Molecular Sociology, Max Planck Institute of Biophysics, Frankfurt am Main, Germany. [6]Institute of Biophysics, Goethe University Frankfurt, Frankfurt am Main, Germany. [7]Present address: Centre for Structural Systems Biology, Hamburg, Germany. [8]Present address: European Molecular Biology Laboratory Hamburg, Hamburg, Germany. [9]These authors contributed equally: Miao Yu, Maziar Heidari, Sofya Mikhaleva, Piau Siong Tan. ✉e-mail: gerhard.hummer@biophys.mpg.de; edlemke@uni-mainz.de

In this study, we developed a method to probe distance distributions of FG-NUPs inside the NPCs by combining fluorescence lifetime imaging of fluorescence resonance energy transfer (FLIM–FRET) with a site-specific synthetic biology approach. We show that the methods deliver quantitative results when using permeabilized cells with functional transport machinery, and offer sound agreement with qualitative measurements from live cells. We focused on NUP98 because it is the essential constituent of the NPC permeability barrier and is accessible to this technology[34–36]. By measuring the distance distribution for 18 labelled chain segments of NUP98 in the NPC using FLIM–FRET, we showed that the FG domain is exposed to—in the terminology of the Flory polymer model[12]—'good solvent' conditions inside the NPC. This enables the protein to adopt much more extended conformations in the functional state than in the highly collapsed state of single chains in solution at 'poor solvent' conditions. We combined our residue-specific measurements with coarse-grained molecular dynamics (MD) at residue resolution. This enabled us to integrate the distance distribution data and the recently solved scaffold structure[7] into a molecular picture of FG-NUP distribution and motion in the central channel of a functional NPC.

## In-cell site-specific labelling of NUP98

High-precision fluorescence measurements of the conformation of FG-NUPs in their functional state require the introduction of labelling tags with minimal linkage errors and minimal disruption of the structures and functions of the labelled proteins. To this end, we performed site-specific labelling with a non-canonical amino acid (ncAA) using genetic code expansion (GCE)[37]. We used the pyrrolysine orthogonal tRNA–synthetase suppressor pair to reassign the amber stop codon (TAG) to incorporate the ncAA *trans*-cyclooct-2-en-L-lysine (TCO*A) at that site. The chemical functionality of this ncAA residue can then be reacted with an organic fluorophore containing a tetrazine moiety to undergo an inverse-electron-demand Diels–Alder reaction (click chemistry[38]). Thus, the dye is stably attached to the protein via a small chemical linker, causing minimal disruption to the protein structure and its function. One potential downside of this technique is that it is not mRNA-specific, leading to background labelling of untargeted proteins with their naturally occurring stop codons being suppressed. To circumvent this problem, we utilized our recently developed film-like, synthetic orthogonally translating organelles (OTOs) to form a distinct protein translational machinery on the outer mitochondrial membrane surface[39]. These organelles exclusively reassigned two amber codons for the target FG-NUP and incorporated TCO*A at the two specified sites with high selectivity, ensuring minimal interference with endogenous protein translation and negligible background staining (see Fig. 1a and Extended Data Fig. 1 for the improved contrast when comparing OTO technology with conventional GCE). The incorporated ncAAs were reacted with a mixture of donor and acceptor dyes for FRET measurements. This results in FRET species mixed with donor-only and acceptor-only species. Therefore, the chosen measurement method must be able to distinguish the FRET species from the other two. Furthermore, quantitative high-precision FRET measurements have high requirements on the properties of the FRET dye pair, such as photostability, Förster radius, monoexponential decay of fluorescence lifetime and fluorescence emission clearly distinguishable from background.

We first performed live-cell labelling of NUP98. We co-transfected COS-7 cells with plasmids encoding NUP98 with an amber codon (NUP98[A221TAG]) and the OTO-GCE system, and treated the cells with various cell-permeable dyes, including JF549-tetrazine, JF646-tetrazine, silicon rhodamine-tetrazine and TAMRA-tetrazine (see Methods). We could hardly identify a clear nuclear envelope due to the low signal-to-noise ratio (from nonspecific dye sticking), unless we highly overexpressed NUP98 (Extended Data Fig. 2).

We also performed labelling with cell-impermeable dyes, which are more hydrophilic and show less nonspecific sticking. To deliver the dyes to the nuclear envelope, we permeabilized the cells with low-dosage digitonin, in which the plasma membrane was selectively permeabilized while leaving the nuclear membrane and endoplasmic reticulum intact[40]. We chose a FRET dye pair of AZDye594-tetrazine (orange) and LD655-tetrazine (red) for their exceptional photostability and suitable Förster radius ($R_0 \sim 7.7$ nm; see Methods), and because they are spectrally distinct from typically green cellular autofluorescence. To verify that our procedure did not perturb the functionality of the NPC, we performed a transport assay on permeabilized COS-7 cells labelled with LD655-tetrazine[41] (Fig. 1b and Extended Data Fig. 3). A large inert cargo (70-kDa FITC-labelled dextran) was excluded from the nucleus, suggesting that both the permeability barrier and the nuclear envelope were intact, whereas IBB–MBP–GFP (a triple fusion of the importin-β-binding domain (which is recognized by the import receptor importin-β) with maltose-binding protein (to make the construct bigger) and green fluorescent protein) supplied with transport mixtures (which contained the nuclear transport receptor importin-β; see Methods) was actively imported into the nucleus, demonstrating that NPCs containing labelled NUP98 in the permeabilized cells were fully functional. Therefore, we performed the following FRET measurements of the NUP98 FG domain with AZDye594 and LD655, unless otherwise stated.

## FLIM–FRET measurements of NUP98

FLIM–FRET is exquisitely sensitive to the spatial distance between pairs of donor and acceptor fluorescent dyes. As one of the few FRET methods that provide quantitative FRET information without a priori knowledge about the actual labelling stoichiometry, FLIM–FRET is also independent of fluorophore concentration and excitation intensity, providing an ideal tool for probing the dimension of FG-NUPs in the cellular milieu[42]. By quantifying the decrease in donor fluorescence lifetime when it undergoes FRET coupling with an acceptor molecule, we determined the average spatial distances between the dyes. The robustness of the method was further enhanced by combining FLIM–FRET measurements with acceptor photobleaching, from which we could measure the fluorescence lifetime of the donor-only population and the cellular background with high precision. In the Supplementary Text (equations (15–17)) and Supplementary Figs. 1 and 2, we further detail how the combination of FLIM–FRET and acceptor photobleaching increases robustness of the analysis.

To make sure that the spectral properties of the chosen dye pair are independent of the labelling site, we labelled 19 different positions of the NUP98 FG domain with only the donor dye. Extended Data Fig. 4 shows that the donor fluorescence lifetimes and anisotropies were similar for each labelling site, demonstrating that the fluorophores experience a similar microenvironment within the NPC and that the dyes are mobile enough to allow for high-precision FRET distance measurements. Another concern is that intermolecular FRET could occur, that is, FRET between different NUP98 molecules in the same NPC, owing to the high FG-NUP density in the nanosized central channel. To verify that no intermolecular FRET was measured, we transfected COS-7 cells with a plasmid encoding NUP98 with only a single amber codon (NUP98[A221TAG]) and treated them with a mixture of donor and acceptor dyes, such that each modified copy of NUP98 could only be singly labelled with either donor or acceptor dye. We selected nuclear envelopes with an acceptor intensity per pixel (excited by a 660-nm laser) below a determined threshold at which to measure FRET to eliminate risks of intermolecular FRET (see Methods; Fig. 2 and Supplementary Fig. 3). We then measured the average fluorescence lifetime of the donor dye before and after acceptor photobleaching. If FRET occurs, the donor intensity and the fluorescence lifetime will increase when the acceptor is bleached selectively by a high-power laser[43]. We found the

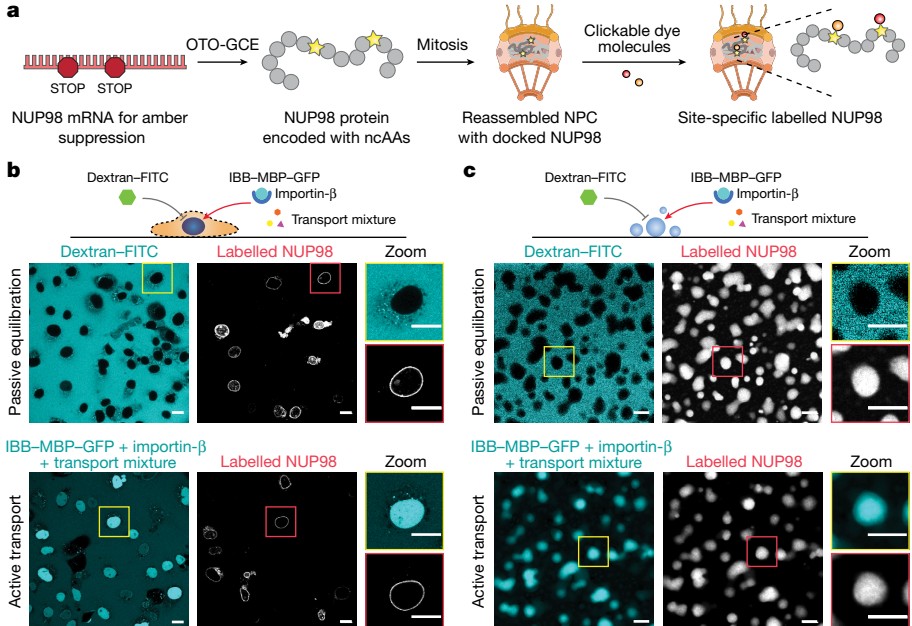

**Fig. 1 | Site-specific labelling of NUP98 in the functional state inside the NPC and comparison with phase-separated condensates in vitro. a**, Schematic of site-specific labelling of target NUP98 inside the NPC. The genetic code was exclusively expanded for target NUP98. ncAAs were introduced into NUP98 at specific sites by synthetic orthogonal translating organelle-enabled genetic code expansion (OTO-GCE). Tetrazine-modified dye molecules were added to the cells and reacted with ncAAs (click chemistry). **b**, Passive exclusion (70-kDa dextran was excluded) and facilitated assay, or active transport assay (IBB–MBP–GFP supplied with transport mixture was imported) showing that the NPCs were functional with site-specific labelled NUP98 in permeabilized COS-7 cells (with the plasma membrane selectively permeabilized with low-dosage digitonin). Scale bars, 20 μm. **c**, Purified NUP98 FG domain was phase-separated in vitro by rapidly diluting a denatured highly concentrated stock solution into physiological buffer. The permeability of the droplet-like condensates formed was measured rapidly where they still obeyed liquid-like characteristics[26]. The droplets recapitulated the function of the permeability barrier, as shown in the passive exclusion and facilitated transport assays. Scale bars, 5 μm. For **b**,**c**, $n$ = 3 experiments were repeated independently with the same conclusion.

donor signal to be unchanged in the permeabilized cells, thus excluding the existence of intermolecular FRET (Fig. 2b). However, when we performed the same assay on the living cells labelled with JF549 and JF646, where we could identify a clear nuclear rim, we noticed an increase of donor lifetime after acceptor photobleaching (Fig. 2c). This indicates the presence of intermolecular FRET under live-cell labelling conditions, where due to the poorer signal to noise, nuclear rims could only be identified when NUP98 was highly overexpressed. Intermolecular FRET does not permit to measure at the same time quantitative intramolecular FRET, and thus we continued with permeabilized cell measurements for the rest of the work.

Next, we conducted the acceptor photobleaching assay on the permeabilized cells expressing double-amber-mutated NUP98 (NUP98[A221TAG–A312TAG]; Fig. 2d). We observed an increase in both the donor intensity on the nuclear rim and its average fluorescence lifetime when the acceptor was photobleached, confirming the occurrence of FRET. With these results, we validated the experimental setup as sufficiently sensitive for measuring FRET between two labelled sites of NUP98 in the permeabilized cells with a functional transport machinery.

We then created a series of NUP98 mutants to form a set of chain segments of different length between the labels for FLIM–FRET measurements. We chose A221TAG as the reference site and kept it constant while varying the second site along the FG domain (Fig. 2e). We define $N_{res}$ as the number of amino acid residues between the two labelled sites and $R_E$ as the root-mean-square inter-residue distance between the fluorophores at these sites. We measured 18 chain segments in COS-7 cells using our developed pipeline (Fig. 2a and Supplementary Figs. 3–6; see Methods for details). In brief, our pipeline involved auto-segmentation to select the nuclear rim from each cell as a region of interest and extraction of measured donor fluorescence intensity profiles before and after acceptor photobleaching, defined as $I(t)$ and $I'(t)$, respectively.

By subtracting $I(t)$ from $I'(t)$, the signals from the donor-only population and the background were eliminated, and the difference was taken as the pure FRET population (here we refer to the FRET population as the NUP98 chains specifically labelled with a pair of donor and acceptor dyes). We directly observed the differences in FRET efficiencies among all mutants in the lifetime fluorescence decay profiles (Fig. 2e). We detected a clear trend by which a smaller $N_{res}$ showed a larger difference before and after acceptor photobleaching, indicating a higher FRET efficiency and smaller $R_E$. To further visualize and compare the fluorescence lifetimes across cells, we converted individual fluorescence decay curves from the time domain into a phasor plot—a graphical view enabling a cell-by-cell analysis of the complicated lifetime curves (Fig. 2f and Extended Data Fig. 5; see Methods)[44]. Each point in the phasor plot represents the fluorescence decay of the nuclear envelope of a single cell. The phasor plot revealed not only some heterogeneity across cells but also an overall trend, in which greater $N_{res}$ exhibited more left-shifted phasor values, corresponding to longer fluorescence lifetime and greater $R_E$. These results suggested that FLIM–FRET could be successfully used to spatially distinguish and map the chain dimensions of the NUP98 FG domain.

## Extracting chain dimensions of NUP98

We determined the chain dimensions of the NUP98 FG domain in situ according to the polymer scaling law, $R_E \sim N_{res}^{\nu}$, which relates the root-mean-square inter-residue distance $R_E$ to the chain length $N_{res}$, where $\nu$ is the scaling exponent[12]. In Flory's homopolymer theory[12], $\nu \sim 0.3$ indicates that the polymer is very compact, as self-interactions dominate over interactions with the poor solvent; at $\nu = 0.5$ those interactions are balanced, and at $\nu \sim 0.6$, the interactions between the polymer segments and the solvent are maximized. Disordered protein

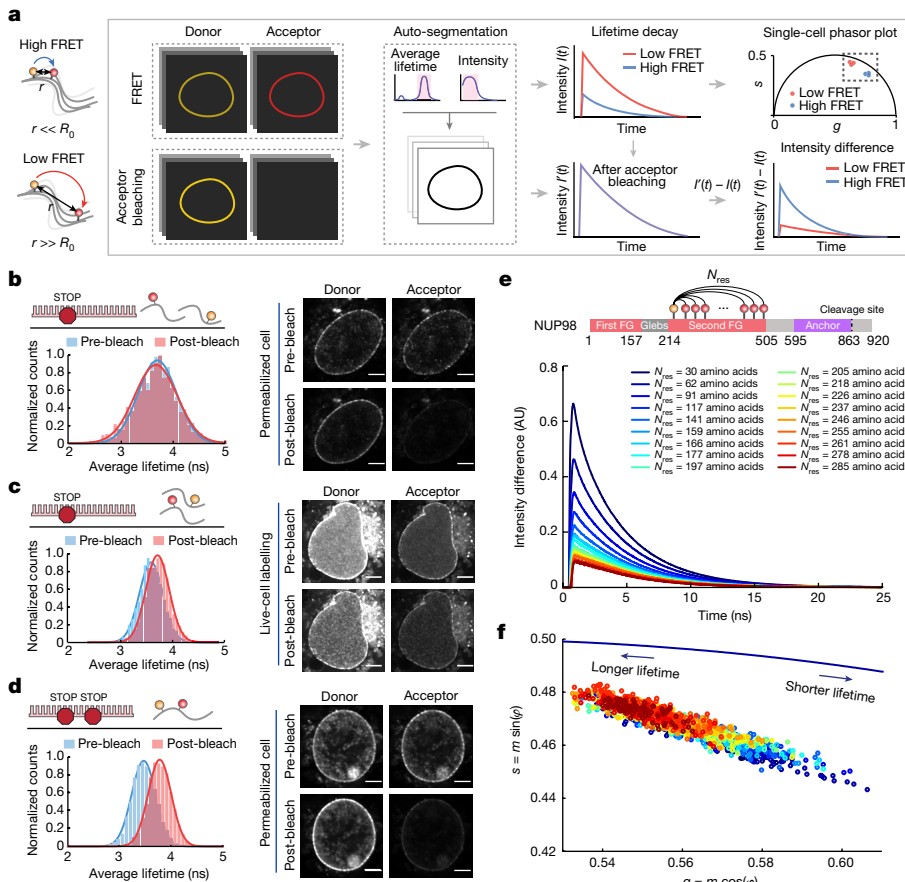

**Fig. 2 | FLIM–FRET measurements of the NUP98 FG domain inside the NPC.**
**a**, Schematic of the FLIM–FRET analysis pipeline. Different chain segments of the NUP98 FG domain were labelled with a FRET dye pair, and the donor fluorescence intensity was measured on a cell-by-cell basis. Each nuclear rim was selected as a region of interest, and the measured donor fluorescence intensity profiles before and after acceptor photobleaching were extracted and analysed. **b**,**c**, Acceptor photobleaching assays were performed for a single-amber-mutated sample (NUP98[A221TAG]) in permeabilized cells labelled with a AZDye594–LD655 mixture (**b**) and living cells labelled with a JF549–JF646 mixture (**c**). In **b**, the average fluorescence lifetime of the donor dye did not change before and after acceptor photobleaching, indicating the absence of intermolecular FRET. In **c**, the average fluorescence lifetime of the donor dye changed before and after acceptor photobleaching changed, indicating that intermolecular FRET was detected in highly overexpressing living cells with a bright nuclear rim. **d**, Acceptor photobleaching assay was performed for a double-amber-mutated sample (NUP98[A221TAG–A312TAG]) labelled with the AZDye594–LD655 mixture. The average fluorescence lifetime of the donor dye changed before and after acceptor photobleaching, validating the presence of intramolecular FRET. For **b**–**d**, $n = 5$ experiments were repeated independently with the same conclusion; scale bars, 5 μm. **e**, Fluorescence decay profile before photobleaching was subtracted from the one after photobleaching for the 18 different chain segments of the NUP98 FG domain. Each profile represents an averaged result of approximately 100 cells. The higher peak shows a greater difference in the intensity profiles, indicating higher FRET efficiency and smaller inter-residue distance. **f**, Phasor plot showing donor lifetimes of the measured 18 chain segments on a single-cell basis (here approximately 2,000 cells in total), in which each point represents the fluorescence decay of one nuclear rim. The left-shifted points represent longer lifetimes.

chains thus expand in good solvent conditions ($v > 0.5$) and compactify in poor solvents ($v < 0.5$)[45]. Flory's theory can be further extended to describe densely grafted polymer brushes where $v \sim 1$ (ref. 46). Note that the scaling law is derived for infinitely long homopolymers. Despite its limiting definition, the law has been applied to calculate an apparent scaling exponent for finite-length proteins[30,47]. In brief, the apparent scaling exponent captures a complex distance distribution in one number and thus provides excellent economy in describing how protein conformational changes are tuned by their environment.

Here, we measured lifetime decay curves for the 18 chain segments of NUP98, probing the FG domain (Extended Data Fig. 5a), and globally fitted them to the Gaussian chain model to extract the scaling exponent $v$ (see Supplementary Text). Gratifyingly, the lifetime decays showed qualitative agreement between living cells and permeabilized cells (Extended Data Fig. 6). However, as concluded before, we refrained from quantitively analysing the scaling law for living cells due to the co-existence of intermolecular and intramolecular FRET. In the permeabilized cells, we obtained a scaling exponent $v = 0.56 \pm 0.03$ (more than 0.5) using a global fit that makes use of the data obtained from all measured cells (approximately 2,000). This indicates that the probed NUP98 FG domain adopted a rather extended conformation inside the NPC in cells (Fig. 3). We also performed a mutant-by-mutant analysis, in which no scaling law model is assumed a priori, but the error is higher, as only approximately 100 cells go into the average of each mutant. Reassuringly, we obtained a scaling exponent $v = 0.55 \pm 0.05$, which agrees well with the global fitting (see Supplementary Text and Supplementary Figs. 7 and 8 for the details of the analysis as well as the error estimate). Encouraged by these results, we further performed a global fitting of our data to a self-avoiding walk-$v$ model, which provides a better description of protein chains under good solvent conditions[48]. We obtained a scaling exponent $v = 0.56 \pm 0.001$ (see Extended Data Fig. 7 and Supplementary Text), showing the robustness of our scaling exponent for different polymer models.

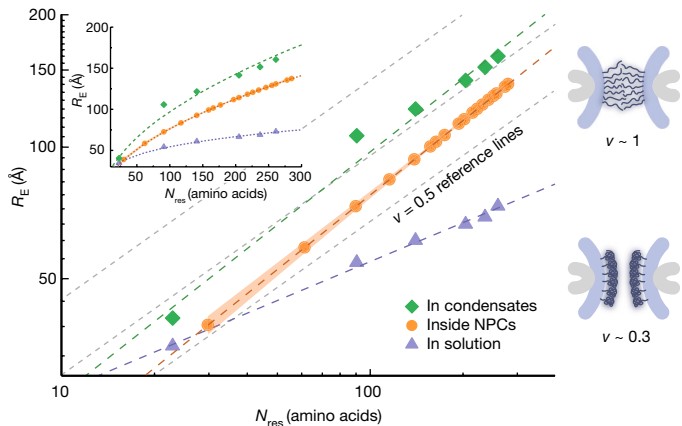

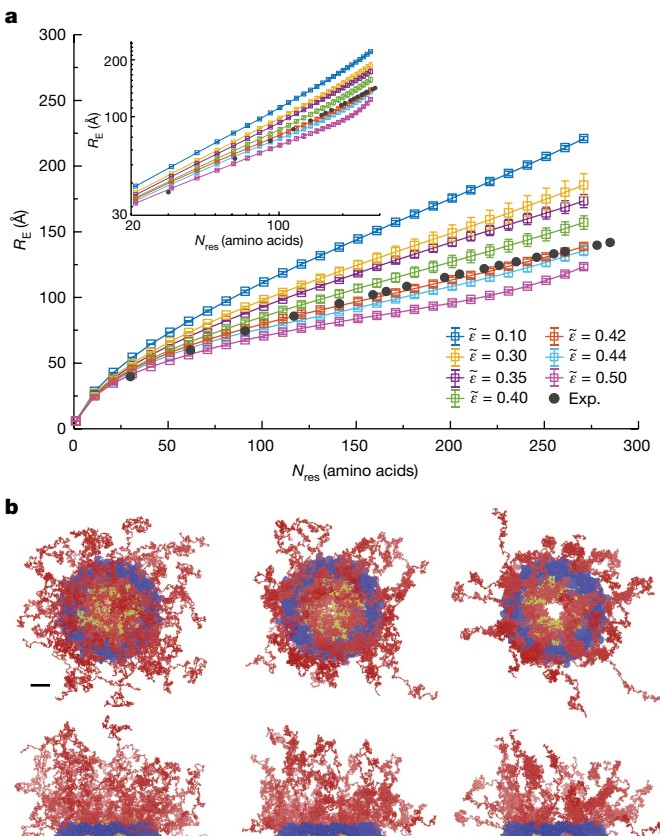

**Fig. 3 | Scaling law of the NUP98 FG domain.** On the left, the NUP98 FG domain showed more extended conformations inside the functional NPC (scaling exponent $v = 0.56 \pm 0.03$) and in the phase-separated condensate (NUP98 FG droplets on coverslips) ($v = 0.56 \pm 0.04$). The NUP98 FG domain at picomolar concentration showed a more collapsed conformation in physiological buffer on a single-molecule level ($v = 0.29 \pm 0.01$). Three reference lines at the theta-solvent state ($v = 0.5$; dashed grey lines) are plotted, because the prefactor of the scaling law is unknown for a heteropolymer under different solvent conditions. The NUP98 FG domain in the NPC was labelled with AZDye594 tetrazine and LD655 tetrazine. The purified NUP98 FG domain in the phase-separated condensate and in the solution was labelled with Alexa Fluor 594 maleimide and LD655 maleimide. On the right, schematics show the conformations of FG-NUPs inside the NPC associated with extreme scaling exponents. If $v \sim 1$, FG domains behave like densely grafted polymer brushes. If $v \sim 0.3$, FG domains adopt a collapsed conformation.

The result inside the NPC contrasts with our in vitro single-molecule FRET measurements on the purified NUP98 FG domain at picomolar concentration (Extended Data Fig. 8), for which the scaling exponent was $v = 0.29 \pm 0.01$ (less than 0.5). With water acting as a poor solvent for the hydrophobic side chains, the single NUP98 FG chain tends to become buried in a globule-like protein conformation in vitro. By contrast, the central channel of the NPC is enriched with other FG-NUPs, whose mutual attractive interactions establish conditions of a good solvent with expanded chains ($v > 0.5$), reminiscent of the chain conformations in polymer melts[12]. On top of that, the presence of nuclear transport receptors, which exist in large quantities in the NPC[49] (Extended Data Fig. 3e,f), and post-translational modifications (for example, glycosylation[50] and phosphorylation), as well as transport cargos such as proteins and RNAs, could also contribute to good solvent conditions.

In reconstituted condensates, NUP98 FG domains tend to gelate over time, potentially by forming β-structures[26,51,52]. During the required FLIM–FRET measurement time of approximately 5 min, the resulting distance could thus be affected by condensate ageing despite using freshly prepared FG droplets. With this caveat, we obtained a scaling exponent of $v = 0.56 \pm 0.04$ for FG condensates, consistent with good solvent conditions (Fig. 3) and with the ability to function as a permeability barrier (Fig. 1c and Extended Data Figs. 3b and 9). However, the large prefactor of 7.8 Å in the scaling fit, compared with 5.5 Å in the self-avoiding walk-$v$ model, indicates heterogeneities in the ageing condensate.

## FG-NUP motions revealed by modelling

The FLIM–FRET experiments assist MD simulations in providing us with a 3D view of the organization of FG-NUPs in functional NPCs. The disordered FG domains are grafted onto the NPC scaffold via folded domains. The recently published structure of the human NPC scaffold[7]

**Fig. 4 | Coarse-grained MD simulations of FG-NUPs in NPC model II.** **a**, Root-mean-square inter-residue distances $R_E$ of beads on the same NUP98 FG domain in the NPC as a function of the number of amino acid residues $N_{res}$ with different effective NUP–NUP interaction strength $\tilde{\varepsilon}$ on linear and (inset) log-scale. The symbols and error bars represent the average and standard error of the mean, respectively, as estimated from four blocks of size $10^4 \tau$. Lines are guides to the eye. The distances obtained from FLIM–FRET experiments (Exp.) are represented by filled circles, which overlap with the simulations for $\tilde{\varepsilon} = 0.42$. **b**, Top and side views of the NPC at the end of the MD simulations with $\tilde{\varepsilon} = 0.35$ (left), $\tilde{\varepsilon} = 0.42$ (middle) and $\tilde{\varepsilon} = 0.44$ (right) (scaffold is shown in blue, NUP98 is in yellow and other FG-NUPs are in red). Scale bar, 20 nm. See Supplementary Table 3 for details on NPC model II and Supplementary Fig. 18 for a view of the 48 NUP98 chains alone.

enabled us to anchor FG-NUPs with the correct positions, orientations and grafting densities. This allowed us to build a coarse-grained bead–spring polymer model[53] and perform MD simulations of the so-far elusive FG domains attached to the NPC scaffold (see Methods). We first parameterized the effective NUP–NUP interaction strength as $\tilde{\varepsilon} = 0.44$ (defined in equation (14) in the Methods) by matching the phase behaviour of in vitro-reconstituted NUP98 FG condensates that mimic the permeability barrier[27,54] (Extended Data Fig. 10 and Supplementary Video 1). However, if the same interaction strength of $\tilde{\varepsilon} = 0.44$ was applied to the whole-NPC simulations, the inner-ring FG-NUPs collapsed onto the scaffold to form surface condensates, despite only weak direct interactions between FG-NUPs and the scaffold ($\tilde{\varepsilon}_{scaffold} = 0.1$). The surface condensates left a void at the centre of the pore with a diameter of approximately 20 nm, which would not seem consistent with the function of the NPC to block the unaided passage of large cargo. After a slight adjustment of the interaction strength to $\tilde{\varepsilon} = 0.42$, the FG-NUPs recovered the $R_E$ matching our FLIM–FRET data (Fig. 4a, Supplementary Figs. 9–11 and 18 and Supplementary

Videos 2–7). We observed that under such a condition, the FG-NUPs formed extended coil configurations and the inner-ring FG-NUPs fluctuated extensively to form a dynamic barrier across the central channel (Fig. 4b and Supplementary Video 3). Remarkably, the optimal FG-NUP interaction strength determined in this way nearly coincides with the critical strength $\tilde{\varepsilon}_c \approx 0.42$ at which condensation occurs. The large FG-NUP motions seen in the MD simulations thus amount to critical fluctuations that create a highly dynamic polymer network well suited for the rapid but selective molecular transport in the permeability barrier[55–58]. In an NPC model with explicit solvent (Supplementary Table 4), the extension of NUP98 follows the experimental FRET measurement at an interaction strength of $\tilde{\varepsilon} = 0.42$, again close to the respective critical value of $\tilde{\varepsilon}_c \approx 0.41$ (see Supplementary Text and Supplementary Figs. 12 and 13). We note that $\tilde{\varepsilon}$ should be considered effective, as it accounts, for example, for the presence of cargo and nuclear transport receptors unresolved in our in situ experiments. Phosphorylation, glycosylation[50] and other post-translational modifications of the FG-NUPs in the NPC will alter $\tilde{\varepsilon}$ and may in turn impact their state, given the sensitivity to $\tilde{\varepsilon}$ for states close to the critical point (Supplementary Fig. 12).

We probed for the possible effects of sequence heterogeneity by adapting the stickers-and-spacers concept[59] for our MD simulations. In the FYW model, we treated aromatic residues (F, Y and W) as mutually attractive stickers separated by weakly interacting spacer regions (see Supplementary Text and Supplementary Fig. 14). As for the homopolymer models, we calibrated the effective sticker interaction strength $\tilde{\varepsilon}_{FYW}$ against the FLIM–FRET measurements of the NUP98 FG domain and applied it to all FG-NUPs in the NPC. We obtained very good agreement of the calculated NUP98 extension with the FLIM–FRET data for $\tilde{\varepsilon}_{FYW} = 3.5$, which again lies just close to the critical value (Supplementary Figs. 15–17), as in the homopolymer models. We conclude that a more detailed model accounting for sequence heterogeneity does not substantially alter the conclusions.

We further explored the possibility of heterogeneity in the chain ensembles by comparing the calculated extensions of the $8 \times 6 = 48$ NUP98 chains grafted to the 8-fold symmetric NPC scaffold at six non-symmetry-equivalent positions (Supplementary Figs. 19 and 20). Consistently across the homopolymer models I and II (Supplementary Tables 2 and 3) and the heteropolymer FYW model, we found that the NUP98 chains at the UCc positions in the cytoplasmic ring are somewhat more extended than at the Uno positions in the inner ring on the nuclear side (Supplementary Fig. 20). Our MD simulations thus indicate some heterogeneity in NUP98 chain extension depending on their precise 'grafting' point within the NPC, including in the scaling behaviour (Supplementary Fig. 20d). A detailed account of electrostatic interactions could further enhance the heterogeneity[60]. The unimodal distributions of the inter-residue distances provide no indication of co-existing collapsed and extended populations (Supplementary Fig. 21).

## Discussion

Here we developed an experimental approach using site-specific fluorescent labelling of IDPs in mammalian cells and FLIM to directly decipher their plasticity by FRET measurements in the functional state. We showed that this approach works for the sub-resolution permeability barrier of the NPC, a nanocavity with a diameter of approximately 60 nm, filled with approximately 50 MDa of highly concentrated FG-NUPs. By measuring the inter-residue distances of different segments of the labelled FG-NUP98 using FLIM–FRET in permeabilized cells with functional transport machinery, we obtained the distribution from which we could estimate an apparent scaling exponent and revealed that the intact NPC environment provides a good solvent ($v > 0.5$) in which the FG domains adopt extended conformations compared with the collapsed solution state in vitro.

Owing to recent advances in determining the structure of the NPC scaffold, the actual grafting points for the FG domains are now known with high confidence[4–11]. We synergistically combined our residue-specific FLIM–FRET measurements with computational modelling and investigated the conformational behaviour and interaction mechanisms of FG-NUPs at the molecular level with residue resolution. We found that a simple polymer model could capture the motions of FG-NUPs. The model reproduced the FLIM–FRET results at a near-critical interaction strength, which defines the energetic threshold for forming protein condensates and is associated with large fluctuations in the dynamic polymer network. For weaker interactions, the FG-NUPs were too loose to generate a polymer network. For stronger interactions, the FG-NUPs formed a surface condensate, the collapse of which onto the scaffold was driven by the geometry of the grafting points, not by the weak direct interactions with the scaffold. Indeed, the chain extension is exquisitely sensitive to the NUP–NUP interaction strength $\tilde{\varepsilon}$ in the NPC (Fig. 4a), whereas in bulk condensates, this dependence is negligible (Extended Data Fig. 10b).

A key finding is that the ensembles of FG-NUPs within functional nuclear pores inside cells are distinct from the chain ensembles sampled in solution, untethered from the pore. The intriguing inference presented is that the apparent solvent quality of the nuclear pore is better than that of a bulk, aqueous solvent, with conformational statistics measured in the pore resembling what one would observe in at least a theta solvent ($v = 0.5$) and more closely, a good solvent ($v > 0.5$). Our study removes much speculation about the conformational state of FG-NUPs in the NPC and provides a sound coarse-grained model with amino acid precision to explain the function of the permeability barrier. The measured apparent scaling exponent $v = 0.56 \pm 0.03$ disaccords with, for example, the polymer brush model[46] ($v \sim 1$) and the forest model[24] ($v \sim 0.3$), which describes the low charge-content, cohesive FG domain as a globular structure. Remarkably, we also showed that the parameterization based on in vitro reconstitution studies failed to reproduce a functional pore. Despite having similar permeability barrier properties as the intact NPC, the bulk condensate formed from phase separating NUP98 is an incomplete approximation of the actual permeability barrier, the materials properties of which are modulated by the anchoring of a distinct number of FG-NUPs with 3D precision on a half-toroidal NPC scaffold. In terms of nuclear transport selectivity, there are consequences: whereas a surface condensate would leave a substantial hole at the centre, we found the hole to be filled by FG-NUPs at near-critical conditions (Fig. 4b). These results emphasize the importance of interrogating the permeability barrier in situ to reconcile different transport models and understand the molecular basis for nuclear transport.

In this work, a range of technologies have been used, which finally enabled us to measure the distance distribution of a key FG-NUP inside the functional transport machinery of the NPC. The combination of fluorescent lifetime technologies and chemical synthetic biology tools compensates for the weaknesses or ambiguities that can originate from either of the methods alone. We show that our labelling technology is also live-cell compatible at the cost of a lower signal-to-noise ratio, which originates from stronger background sticking of the tested membrane-permeable dyes (Extended Data Fig. 2). Although this problem might not be an issue for studying biological systems that are more abundant, for the NPC we could only solve it by increasing expression levels. However, owing to the fact that 48 NUP98 chains are cramped into the nanosized cavity of the NPC, with approximately 250 FG-NUPs in total (Fig. 4), intermolecular FRET can become a concern. We show a procedure to detect intermolecular FRET; with this, the method should be generally extendable to other systems. To avoid intermolecular FRET within the nanosized NPC, we had to work at very low expression levels. Sufficient signal-to-noise was then ensured by mild permeabilization of the plasma membrane, which enabled us to use hydrophilic dyes that stick less. At the same

time, we validated that the nuclear transport machinery is still intact in situ. Owing to the potential protein truncation problem arising from translation termination at the reassigned codon in GCE[39,61], we refrained from probing the FG-NUPs with N-terminal anchoring domains (for example, FG-NUP214), where the truncated proteins could dock on the nuclear pore and interfere with labelling or even transport functionality. We currently could only infer the molecular behaviours of other FG-NUPs by MD simulations. It is highly desired to experimentally probe all FG-NUPs in living cells when more advanced GCE and highly fluorogenic cell-permeable dyes compatible with GCE are available in the future. However, as NUP98 has been shown to be the only NUP that is vital to reconstitute functional transport activity of intact NPCs[36], our study captures fundamental properties of NPC function.

The most advanced computational modelling tools, such as the AI-based AlphaFold, can very precisely predict the structures of folded proteins. However, when applied to IDPs, they still require experimental constraints and validation[62]. It is necessary to probe IDPs with in situ experimental measurements. As we showed here, a protein chain with globular-like character in vitro can adopt a much more expanded conformation in the functional state due to the cellular environment. Our work is an example of how the structural knowledge from cryo-electron tomography—which revealed the anchoring sites of IDPs—paired with in situ FLIM–FRET—which revealed the conformational state of the proteins inside the nanosized NPC—can yield a complete coarse-grained picture of a cellular machine enriched in IDPs inside a cell. The tools developed here could be applied generally to study the plasticity and functions of many other IDPs in the cell, filling a major technology gap in the field.

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

## Methods

### Cell culture, transfection and labelling

**Cell culture.** COS-7 cells (87021302, Sigma) were maintained in Dulbecco's modified eagle medium (41965-039, Thermo Fisher) supplemented with 10% v/v fetal bovine serum (F7524, Sigma), 1% penicillin–streptomycin (15140-122, Thermo Fisher), 1% L-glutamine (25030-081, Thermo Fisher) and 1% sodium pyruvate (11360-070, Thermo Fisher) at 37 °C and 5% $CO_2$, and passaged every 2–3 days up to 15–20 passages. The COS-7 cells were authenticated by the manufacturer, validated by morphology and regularly tested for mycoplasma contamination, with negative results.

**Cell transfection.** Cells were trypsinized (trypsin-EDTA; 25300-054, Thermo Fisher) and seeded into a 35-mm imaging dish (81158, ibidi) 24 h before transfection. The cells were transfected at a confluency of 60–70% with plasmids of interest listed in Supplementary Table 1 in Supplementary Information (for amber suppression with synthetic organelles, for example, pcDNA3.1-hsNUP98[TAG]-BoxB and pcDNA3.1-TOM20-FUS-$\lambda_{N22}$-PylRS(AF)-tRNA at a mass ratio of 1:1) using jetPRIME according to the manufacturer's protocol. After 4–5 h, the medium was changed, and 10 mM HEPES with 50 µM *trans*-cyclooct-2-en-L-lysine (TCO*A; SciChem; for in-cell labelling) or *t*-butyloxycarbonyl-L-lysine (BOC; Iris Biotech; for control measurements) were added, respectively.

**Live-cell labelling with cell-permeable dyes.** At 20–24 h post-transfection, COS-7 cells were washed with fresh culture medium supplemented with 50 µM BOC and 10 mM HEPES and incubated at 37 °C for 2 h to get rid of the excessive TCO*A. Then, the cells were incubated with 250 nM cell-permeable dye (Janelia Fluor 549-tetrazine (Tocris), Janelia Fluor 646-tetrazine (Tocris), silicon rhodamine-tetrazine (Spirochrome) or TAMRA-tetrazine (click chemistry tools)) in the serum-free culture medium at 37 °C for 45 min. After the labelling, the cells were washed with fresh culture medium four times in 2 h at 37 °C. Cells were imaged immediately at room temperature in the phenol red-free culture medium supplemented with 10 mM HEPES. The lid of the imaging dish was kept closed for the duration of cell imaging for 2 h.

**Cell labelling with cell-impermeable dyes.** At 20–24 h post-transfection, COS-7 cells were washed twice with transport buffer[40] (TB; 20 mM HEPES, 110 mM KOAc, 5 mM NaOAc, 2 mM Mg(OAc)$_2$, 1 mM EGTA and 2 mM DTT, pH 7.3, supplemented with PEG6000 (5 mg ml$^{-1}$) to avoid osmotic shock[41]), and permeabilized for 2 min with 20 µg ml$^{-1}$ digitonin (A1905, AppliChem) in TB. After the permeabilization, cells were washed twice with TB to remove digitonin and labelled with a dye solution containing 33.3 nM AZDye594-tetrazine (Click Chemistry Tools) and 66.6 nM LD655-tetrazine (Lumidyne Technologies) in TB for 2 min. Here we optimized the molar ratio between donor and acceptor dyes as 1:2 to reduce the donor-only population. To remove residual dyes, the cells were washed in TB and incubated at 37 °C for 30 min. Cells were imaged immediately at room temperature. The lid of the imaging dish was kept closed for the duration of cell imaging for 2 h.

### Passive exclusion assay and active transport assay for labelled cells

**Passive exclusion assay.** COS-7 cells were transfected, permeabilized and labelled with 100 nM LD655-tetrazine as described above. After the last wash, the imaging dish was mounted on the custom-built confocal microscope. After gently removing the washing buffer, 0.5 µM of 70-kDa FITC–dextran (53471, Sigma) in TB was added to the dish, incubated for 10 min and the cells were imaged.

**Active transport assay.** To allow the import complex to form, 0.5 µM IBB–MBP–GFP cargo was first pre-incubated with 1 µM importin-β on ice for 10 min, and then combined with the rest of the transport mix (5 µM RanGDP, 4 µM NTF2 and 2 mM GTP in TB, where importin-β, Ran and NTF2 were purified as previously described[55,63]). After that, the complete import mixture was added to the permeabilized and labelled COS-7 cells as described above.

To ensure that the NPC was functional during the 2-h time window of FLIM–FRET measurements for each imaging dish, passive exclusion assay and active transport reactions were performed for both the freshly labelled cells and the cells incubated in TB at room temperature for 2 h after the labelling steps (Extended Data Fig. 3c,d).

### Immunostaining of endogenous importin-β

After labelling with 100 nM LD655-tetrazine in TB as described above, the COS-7 cells were immediately fixed with 2% PFA in PBS for 10 min or kept at room temperature for 2 h and then fixed. After washing with PBS twice, the cells were permeabilized with 0.5% Triton X-100 in PBS for 15 min. Then, the cells were washed with PBS twice and incubated with blocking buffer (3% BSA in PBS) for 90 min. The cells were subsequently immunolabelled with anti-KPNB1 antibody (ab2811, Abcam; at 1:1,000) and anti-mouse Alexa Fluor 488 secondary antibody (A-11001, Thermo Fisher; at 1:1,000). Immunofluorescence demonstrated that endogenous importin-β colocalized at the nuclear envelope after permeabilization and was retained during the 2-h time window of FLIM–FRET measurements (Extended Data Fig. 3e,f).

### Protein expression, purification and labelling

**NUP98 FG domain purification.** *Homo sapiens* NUP98 FG domain (1–505 amino acids, with or without the Gle2-binding domain (GLEBS; 157–213 amino acids), a structured domain in between the two FG domains) was cloned into pQE-14his-TEV vectors. We also designed a construct without GLEBS, because the structured GLEBS domain might misfold and trigger aggregation when rapidly changing the buffer condition from denaturing to native in the droplet assay (see 'In vitro droplet assay and FLIM–FRET measurements'). The GLEBS domain is outside the residue segments probed by the in-cell experiments. Unless otherwise stated, we used the NUP98 construct without GLEBS for in vitro experiments.

*Escherichia coli* BL21 AI cells containing NUP98 FG construct with or without GLEBS were grown in terrific broth medium containing 50 µg ml$^{-1}$ of kanamycin at 37 °C and 200 rpm, and protein expression was induced with 0.02% (w/v) arabinose and 1 mM IPTG at $OD_{600}$ = 0.6. After 16 h of expression at 18 °C, cells were harvested by centrifugation and lysed in a cell disruptor (CF1, Constant Systems) in the lysis buffer containing 6 M GdmCl, 0.2 mM Tris(2-carboxyethyl)phosphine (TCEP), 20 mM imidazole and 50 mM Tris-HCl, pH 8. The lysate was centrifuged for 1 h at 12,000$g$ at 4 °C to remove cell debris, and the supernatant was incubated with Ni-beads for 2 h at 4 °C. The Ni-beads with lysate were loaded in polypropylene tubes (Qiagen), washed twice with 2 M GdmCl and 20 mM imidazole, pH 8, and eluted with buffer containing 2 M GdmCl and 500 mM imidazole, pH 8. To remove the His-tag, the elution was dialysed in 0.5 M GdmCl and 50 mM Tris-HCl, pH 8, and cleaved overnight with TEV protease at room temperature. Proteins were incubated again with Ni-beads in 2 M GdmCl and 50 mM Tris-HCl, pH 8, to remove cleaved His-tags, TEV protease and nonspecific proteins with Ni-bead affinity. The flow-through containing the NUP98 FG domain was collected and further purified by size-exclusion chromatography (Superdex 200, Akta pure protein purification system, Cytiva) in 2 M GdmCl, 0.2 mM TCEP and 50 mM Tris-HCl, pH 8. Fractions were analysed by SDS–PAGE and stained with Coomassie blue. Pure fractions were pooled and concentrated to around 15 mg ml$^{-1}$ in 4 M GdmCl using 3-kDa MWCO centrifugal filters (Merck Millipore), with the concentration measured by a BCA protein assay kit (Thermo Fisher). The proteins were flash-frozen and stored at −80 °C.

**NUP98 FG domain labelling in vitro.** For labelling, the purified NUP98 FG domain with single or double cysteine mutations was exchanged

to 4 M GdmCl, 1× PBS, 0.1 mM EDTA and 0.2 mM TCEP, pH 7. Labelling with Alexa Fluor 594 maleimide (A10256, Thermo Fisher) and LD655-maleimide (Lumidyne Technologies) was done at the molar ratio of 1:2 (dye:protein) overnight at 4 °C. The reaction was quenched with 10 mM DTT in 4 M GdmCl and 1× PBS, pH 7. Unreacted dye was washed off using a 3-kDa MWCO centrifugal filter, and the labelled protein was further purified with Superdex 200. Pure fractions were chosen, pooled and concentrated as described above, and the final concentration was measured by the absorbance spectrometer Duetta (Horiba). The proteins were flash-frozen and stored at −80 °C.

## FLIM−FRET imaging setup

The custom-built FLIM−FRET imaging setup (Supplementary Fig. 6) was equipped with picosecond pulsed laser diode heads including the wavelengths of 485 nm (LDH-D-C-485, PicoQuant), 560 nm (LDH-D-TA-560, PicoQuant) and 660 nm (LDH-D-C-660, PicoQuant). The laser heads were controlled through a multichannel picosecond diode laser driver (Sepia II PDL 828, PicoQuant). The beams were coupled into a single-mode polarization-maintaining optical fibre (KineFLEX-P-2-S-405/640-2.5-2.5-p2) and fibre coupler (60FC-4-RGBV11-47, Schäfter + Kirchhoff). The beam travelled through a Glan-laser polarizer (Thorlabs) and was directed into a laser scanning system (FLIMbee, PicoQuant). The three galvo mirrors in the scanning system were imaged onto the backfocal plane of the objective (×60 SR Plan Apo IR, 1.27 NA; Nikon) with 200-mm tube lens. The fluorescence emission was focused onto a pinhole (100 μm for cell measurements, 50 μm for droplet measurements and 100 μm for single-molecule FRET measurements), and then separated into parallel and perpendicular components using a 50/50 polarizing beam splitter (Thorlabs). Each component was further separated by two sets of beamsplitters (ZT561 RDC and T647 LPXR, Chroma), passed through three sets of bandpass filters (green channels: 525/50 BrightLine HC; orange channels: 609/57 BrightLine HC, Semrock; red channels: ET700/75m, Chroma), and focused onto the single-photon counting detectors (green channels and orange channels: PMA Hybrid 40, PicoQuant; red channels: τ-SPAD, PicoQuant). The signals from the photon detectors were recorded by a TCSPC system (HydraHarp 400, PicoQuant) at a time resolution of 16 ps. Data acquisition was carried out with SymPhoTime 64 software v2.6 (PicoQuant).

## Determination of Förster radius $R_0$

The Förster radius $R_0$ is the distance between a pair of fluorophores at which the FRET efficiency is 50%, which was calculated as[43],

$$R_0 = 0.211 \sqrt[6]{\frac{\kappa^2 \phi_D J(\lambda)}{n^4}} \ (R_0 \text{ in Å}) \tag{1}$$

where $\kappa^2$ is the orientation factor, $n$ is the refractive index ($n = 1.375$ for cell measurements[64], and $n = 1.426$ for in vitro condensates[65]), $\phi_D$ is the quantum yield of the donor without energy transfer, and $J(\lambda)$ is the overlap integral between the donor emission and acceptor absorption spectra at wavelength $\lambda$, given by[43],

$$J = \int F_D(\lambda) \varepsilon_A(\lambda) \lambda^4 d\lambda / \int F_D(\lambda) d\lambda \tag{2}$$

where $F_D(\lambda)$ is the radiation emission intensity of the donor at wavelength $\lambda$, and $\varepsilon_A(\lambda)$ is the extinction coefficient of the acceptor. The emission spectra of the donor and the excitation spectra scan of the acceptor were measured on a Leica SP8 STELLARIS microscope using $xy\lambda$ or $xy\Lambda$ acquisition mode, for single-labelled cells with the donor or the acceptor, respectively. The acquired spectra were then compared with the spectra measured for free dyes in TB using an absorbance spectrometer (Duetta) and no spectral shift was detected. $\kappa^2$ can be assumed to be approximately two-thirds when the dyes can freely rotate, which is expected to be the case as all our dyes have a C5 flexible linker between the conjugating group and the

chromophore. Here we assumed near-parallel excitation and emission dipoles. We also measured the fluorescence anisotropies for the sample labelled with the donor dye inside the NPC (0.27 ± 0.015), in the FG condensates in vitro (0.28 ± 0.003), and in the solution on a single-molecule level (0.15 ± 0.003). The measured anisotropies were found to be smaller than 0.3 in all cases, and the error in the distance measurements was shown to be below 10% in such a regime[43]. The $R_0$ for in cell measurements was determined as 77.1 Å, and for in vitro condensate measurements was 77.3 Å. Note that the apparent scaling exponent is robust to the error in $R_0$ measurements (see Supplementary Fig. 7).

## FLIM−FRET for cell measurements

The average power of laser excitation was optimized to collect enough photons from the cell within a reasonable time but avoid photon pile-up and other artefacts in the fluorescence lifetime measurements. The instrument response function was measured on a daily basis using a freshly prepared saturated solution of KI and erythrosine B[66]. The temporal offsets of the parallel and perpendicular detectors were pre-aligned by the measured instrument response functions. The cell measurements were performed using the following imaging settings: pixel size of 100 nm, image size of 256 × 256 pixels, pixel dwell time of 150 μs and the time resolution of 16 ps. The fluorescence photons were detected in T3 mode and collected from the perpendicular and parallel detectors for each colour individually.

To measure FLIM−FRET, the morphology of the labelled cell was first checked with 660-nm laser excitation to ensure that no GLFG bodies (normally a sign of overexpressed NUP98 in the cell[67]) existed in the nucleus. The acceptor intensity per pixel of the nuclear rim was then checked with 660-nm laser excitation to further assess the expression level of the mutant NUP98, ensuring that cells with similar expression levels and not highly overexpressed mutant NUP98 were chosen. Specifically, we determined such an acceptor intensity range based on the criteria that, on the nuclear rim, the relative FRET efficiency (that is, the proximity ratio, $E_{rel.} = \frac{I_A}{I_A + I_D}$; $I_A$ and $I_D$ are the total acceptor and donor fluorescence intensities, respectively, excited by a 560-nm laser) did not correlate with the acceptor intensity per pixel excited by a 660-nm laser (Supplementary Fig. 3). This intensity threshold was further verified by acceptor photobleaching assays on cells that expressed single-amber-mutant NUP98 and labelled with donor and acceptor dye mixture. The average fluorescence lifetime of the donor dye did not change before and after acceptor photobleaching as shown in Fig. 2b, indicating that no intermolecular FRET could be detected.

After checking the expression level of the mutant NUP98 with 660-nm laser excitation, we imaged the selected cell for 5 min using 560-nm laser excitation with an average power of 40 μW at 40 MHz, and then for 30 s using 660-nm laser excitation with an average power of 35 μW at 40 MHz. Next, the acceptor labelling was photobleached using 660-nm laser excitation with an average power of 300 μW at 40 MHz for 2 min. The donor signal was measured again post-photobleaching for 5 min using 560-nm laser excitation with an average power of 40 μW at 40 MHz.

The recorded images were processed using an automatic segmentation pipeline developed based on the software package PAM in MATLAB[68]. The nuclear rim was selected as a region of interest (ROI) using a thresholding algorithm according to the intensity and average lifetime of each pixel. The time-resolved donor fluorescent intensity profiles before and after acceptor photobleaching were extracted from the selected ROI, respectively. The total fluorescence decay was calculated by combining the parallel and perpendicular fluorescence decays ($I_\parallel$ and $I_\perp$, respectively)[69],

$$I(t) = (1 - 3L_2)GI_\parallel(t) + (2 - 3L_1)I_\perp(t) \tag{3}$$

where $L_1$ and $L_2$ are factors accounting for polarization mixing caused by the high numerical aperture objective lens, and $G$ is the factor accounting for the difference in the detection efficiencies $\eta$ between parallel and perpendicular polarization, given by,

$$G = \frac{\eta_\perp}{\eta_\parallel} \tag{4}$$

The detected fluorescence decays in parallel and perpendicular channels, $D_{par}$ and $D_{per}$, respectively, for a reference sample are expressed as[64]

$$D_{par}(t) = \frac{1}{3}D_0\exp(-t/\tau)\left\{1 + \frac{2}{5}(2-3L_1)\exp(-t/\rho)\right\} \tag{5}$$

$$D_{per}(t) = \frac{1}{3}D_0\exp(-t/\tau)\left\{1 - \frac{2}{5}(1-3L_2)\exp(-t/\rho)\right\} \tag{6}$$

To determine $L_1$ and $L_2$, the fluorescence decays of YFP with known fluorescence rotational relaxation time ($\rho = 16$ ns)[70] was measured and globally fitted with equations (5 and 6).

$G$ factor was determined as a ratio of the average intensities of the perpendicular and parallel donor channels when measuring a solution of Tris(2,2′-bipyridyl)dichlororuthenium(II) chloride ([Ru(bpy)$_3$]Cl$_2$). For each mutant, the total fluorescence decays for approximately 100 cells were added up for further fitting analysis (see the sections about FLIM analysis methods and fitting the scaling law for NUP98 FG in the NPC in the Supplementary Text).

## In vitro droplet assay and FLIM–FRET measurements

The purified and labelled NUP98 FG domain was mixed with unlabelled protein at a molar ratio of 1:5,000 in 2 M GdmCl and 1× PBS, pH 7. Of such a mixture, 1 µl was then quickly mixed with 24 µl TB supplemented with 5 mg ml$^{-1}$ PEG6000 in a chambered coverslip (81507, ibidi) and imaged immediately with the custom-built confocal microscope. The final total concentration of NUP98 in the system was 10 µM for unlabelled and 2 nM for labelled. The trace amount of GdmCl left in solution was 80 mM (a negligible level).

For the passive exclusion assay and the facilitated/active transport assay on the phase-separated condensates, unlabelled NUP98 and the LD655-labelled condensates were mixed with the ratio described above, to avoid the crosstalk with the fluorescently labelled cargos (that is, FITC or GFP). After mixing the protein with TB for 5 min, the buffer was carefully replaced by 0.5 µM 70-kDa FITC–dextran in TB or 0.5 µM IBB–MBP–GFP in the transport mixture as described for the labelled cells avoiding disturbing the condensates on the coverslip surfaces and imaged immediately.

To measure FLIM–FRET on the NUP98 FG condensates, we focused on the first 5 min as the condensates behaved more liquid-like during this time[26] (for example, FG-droplets merged quickly as shown in Extended Data Fig. 9a). We applied the same conformation distribution model (that is, the Gaussian chain model) to fit the lifetime curves of the FG condensates as we used for in cell measurements (see Supplementary Text).

The formed condensates were measured for 5 min using 560-nm laser excitation with an average power of 70 µW at 40 MHz, and the imaging settings were as follows: pixel size of 200 nm, image size of 256 × 256 pixels, pixel dwell time of 100 µs and time resolution of 16 ps. The fluorescence photons were detected in T3 mode and collected from the perpendicular and parallel detectors for each colour individually. The FG-droplets were selected as the ROI based on the intensity and average lifetime in the SymPhoTime 64 software. The time-resolved donor fluorescent intensity profiles were extracted from the ROI pixels for analysis.

To ensure that no intermolecular FRET was detected, single-cysteine-mutated NUP98 was labelled with either the donor dye or with the mixture of donor and acceptor dyes. The labelled protein was then mixed with the unlabelled protein with the same concentration as described above to perform the droplet assay. The donor fluorescence lifetimes of the donor-only and the donor–acceptor mixture showed no difference, indicating that no intermolecular FRET could be detected (Extended Data Fig. 9b,c).

## Phasor transformation of FLIM data

To analyse the lifetime decays on a single-cell basis, the raw intensity profile $I(t)$ of each selected nuclear rim was plotted as a single point in a phasor plot by applying the Fourier transform to the measured decay data, given by[71]

$$g(\omega) = m\cos(\varphi) = \frac{\int_0^T I(t)\cos(2\pi ft)\,dt}{\int_0^T I(t)\,dt} \tag{7}$$

$$s(\omega) = m\sin(\varphi) = \frac{\int_0^T I(t)\sin(2\pi ft)\,dt}{\int_0^T I(t)\,dt} \tag{8}$$

where $g(\omega)$ and $s(\omega)$ are the $x$ and $y$ coordinates of the phasor plot, respectively, $m$ and $\varphi$ are the modulation and the phase delay of emission with respect to the laser excitation that has the repetition rate of $f$, that is, 40 MHz, and $T$ is the repeat frequency of the acquisition, that is, 25 ns. To establish the correct scale for the plotted phasor points, the coordinates of the phasor plot were first calibrated by applying Fourier transform to the measured instrument response function trace and setting it as the zero lifetime[71]. Each phasor point from the acquired FLIM data was then calibrated accordingly using the same calibration parameters so that the final phase plot was referenced relative to the calibration standard. The above procedure was performed with a self-written code in MATLAB.

## Single-molecule FRET measurements

Single-molecule FRET measurements were performed with the custom-built confocal microscope mentioned above. The sample was illuminated with 560-nm and 660-nm lasers in pulsed interleaved excitation mode at a repetition rate of 32 MHz. Photon counts were recorded with a resolution of 16 ps. Purified and double-labelled NUP98 was diluted to the final protein concentration of 50 pM in 1× PBS supplemented with 10 mM fresh DTT. The donor dye was excited with a 560-nm laser at an average power of 70 µW and a 660-nm laser at an average power of 20 µW. Data acquisition was carried out with the SymPhoTime 64 software (Picoquant).

Acquired data were analysed with the PAM software package[68] for burst search, and the identified bursts were further analysed in the BurstBrowser. The FRET efficiency $E$ in a burst is defined as[72],

$$E = \frac{I_A^D}{\gamma_{cor}I_D^D + I_A^D} \tag{9}$$

and the stoichiometry $S$ is defined as[67],

$$S = \frac{\gamma_{cor}I_D^D + I_A^D}{\gamma_{cor}I_D^D + I_A^D + I_A^A} \tag{10}$$

where $I_A^D$ is the acceptor fluorescence detected upon donor excitation, $I_D^D$ is the donor fluorescence upon donor excitation, $I_A^A$ is the acceptor fluorescence detected upon acceptor excitation, and $\gamma_{cor}$ is the factor accounting for the detection efficiency of the acceptor and donor channels. After performing the correction for the leakage of donor fluorescence into the acceptor channel ($\alpha = 1.00$) and direct acceptor

excitation ($\delta = 0.30$) and confirming minimal variation of quantum yields among different mutants, the $\gamma_{cor}$ parameter was extracted from the apparent FRET efficiency, $E_{app}$, and the apparent stoichiometry, $S_{app}$ (ref. 72). A linear fit to a plot of $1/S_{app}$ versus $E_{app}$ yields intercept $a$ and slope $b$, which relates to $\gamma_{cor}$ in the following way,

$$\gamma_{cor} = (a-1)/(a+b-1) \tag{11}$$

We determined the $\gamma_{cor}$ parameter of labelled NUP98 in $1\times$ PBS to be 2.51. Single-molecule transfer efficiency histograms are shown in Extended Data Fig. 8. Each dataset shows the donor-only species at zero FRET efficiency (owing to incomplete labelling or photophysically inactive acceptor such as due to bleaching) and a FRET population. The FRET population was selected and fitted with an asymmetric Gaussian function to determine the centre of the population while accounting for non-linear effects at high FRET efficiency. The fitted values were taken as the average FRET efficiency, which is defined as[73],

$$\langle E \rangle = \int_0^\infty E(r)\rho(r)\,\mathrm{d}r = \int_0^\infty \frac{1}{1+\left(\frac{r}{R_0}\right)^6}\rho(r)\,\mathrm{d}r \tag{12}$$

where $\rho(r)$ describes the distance distribution of the inter-residue distance. A typical single-molecule FRET (smFRET) experiment does not contain enough information to retrieve a model-free distance distribution, and one must therefore choose a model to fit. Here we adopted the Gaussian chain model, which assumes that the monomers occupy zero volume and excluded volume effects are not considered. Despite its simplicity, it is commonly used for the analysis of IDPs[30,74]. The distance distribution function takes the form given by the Gaussian chain model as[73]

$$\rho(r) = 4\pi r^2 \left(\frac{3}{2\pi\langle r^2\rangle}\right)^{\frac{3}{2}} e^{-\frac{3r^2}{2\langle r^2\rangle}} \tag{13}$$

where the root-mean-square inter-residue distance $\sqrt{\langle r^2\rangle} \equiv R_E$. By comparing the fitted $\langle E \rangle$ with the calculated average FRET efficiency derived from the model, $R_E$ was obtained for each mutant.

## MD simulation methods
We performed MD simulations to model the large-scale conformations and motions of FG-NUPs in the NPC and in in vitro-reconstituted NUP98 FG condensates. To ensure efficient sampling, we used a coarse-grained polymer model, in which each amino acid is represented by a single bead without local structure.

**Composition and structure.** The model complements the recently resolved scaffold structure[7] of the human NPC with a nearly complete set of FG-NUPs. In our model I, we did not include NUP153 and POM121, whose anchor positions are not yet resolved with high confidence (see Supplementary Table 2). In our model II, we included NUP153 and POM121 (see Supplementary Table 3). The FG-NUPs were grafted onto the NPC scaffold in the constricted state (PDB ID: 7R5K) at established positions as indicated[7]. The NUP98 simulation models included the GLEBS domain, which is outside the residue segments whose distances were probed in simulations or experiments.

**Energy function.** We used a FENE potential $U_{FENE}$[53] to describe the disordered FG-NUPs, Lennard–Jones (LJ) interactions $U_{LJ}$ between the CG beads, and a potential confining NUP98 chains. The total potential energy is

$$U = U_{LJ} + U_{FENE} + U_C = 4k_BT \sum_{i<j, r_{ij}<r_c} \tilde{\epsilon}_{ij}\left[\left(\frac{\sigma}{r_{ij}}\right)^{12} - \left(\frac{\sigma}{r_{ij}}\right)^6\right]$$

$$+ 4k_BT \sum_{\langle i,j=i+1\rangle, r_{ij}<2^{\frac{1}{6}}\sigma}\left[\left(\frac{\sigma}{r_{ij}}\right)^{12} - \left(\frac{\sigma}{r_{ij}}\right)^6 + \frac{1}{4}\right]$$

$$- \sum_{\langle i,j=i+1\rangle} 0.5k_{FE}R_{FE}^2 \ln\left[1 - \left(\frac{r_{ij}}{R_{FE}}\right)^2\right]$$

$$+ k_c \sum_{i\in NUP98, POM121, r_{xy,i}>R_c}(r_{xy,i} - R_c)^2 \tag{14}$$

The length and interaction energy scales of the LJ potential between a pair of beads $i$ and $j$ are $\sigma$ and $\tilde{\epsilon}_{ij}$, respectively. The latter is normalized by the thermal energy $k_BT$, where $k_B$ is the Boltzmann constant and $T$ is the system temperature. We fixed the interaction strength between scaffold and FG beads at $\tilde{\epsilon}_{ij} = \tilde{\epsilon}_{scaffold} = 0.1$, which ensures that chains do not stick to the scaffold. The interaction strength between FG beads is given uniformly by $\tilde{\epsilon}_{ij} = \tilde{\epsilon}$. The sums over $\langle i,j=i+1\rangle$ extend over the bonded beads of the FG-NUPs. The bond contraction and stretching are controlled by the repulsive LJ and logarithmic terms, whose maximum bond limit is $R_{FE} = 1.5\sigma$ with $k_{FE} = 80\,k_BT$. To mimic the membrane envelope underneath the scaffold, we applied an axial confinement on NUP98-FG and POM121-FG beyond a radius of $R_c = 81\sigma \equiv 48.6$ nm, where $r_{xy} = \sqrt{x^2+y^2}$ and $k_c = 80\,k_BT$.

**MD simulations.** We used the LAMMPS software package[75] to simulate the polymeric systems. The systems were thermalized with a Langevin thermostat[76] (at $k_BT = 1$) with a damping coefficient of $10\tau$. We used a uniform mass $m$ for all monomers and a characteristic time scale $\tau = \sqrt{m\sigma^2/k_BT}$. The time step of all simulations was set to $0.01\,\tau$ for the single-chain and condensate simulations and to $0.001\,\tau$ for NPC simulations. We used block averaging with four non-overlapping blocks to estimate uncertainties. In the MD simulations of the FG-NUPs attached to the NPC scaffold, the residues of the FG-NUPs grafted to the scaffold (see Supplementary Tables 2 and 3 for details) and the scaffold itself were kept frozen.

We determined the bond length $\sigma$ by matching the root-mean-square inter-residue distance distribution of the CG model of a single NUP98 FG chain (1–499) to that in the Martini 2.2 level of coarse graining[77,78]. With a focus on the geometric extensibility, we worked in the limit of weak non-bonded interactions between distant amino acids. For the Martini simulations of the single chain, we used $\alpha$-scaling ($\alpha = 0.1$)[79], and for the CG model, a weak cohesive strength between FG beads ($\tilde{\epsilon} = 0.1$). The Martini MD simulations were performed with GROMACS 2020.6 (refs. 80,81). The first 499 N-terminal amino acids of NUP98 were converted into a Martini coarse-grained model using the martinize.py python script[77,78]. A cubic simulation box of length 30 nm was built and solvated with coarse-grained Martini water and 10% anti-freezing water. Ions were added to neutralize the system. The system was initially energy minimized and then equilibrated for 500 ns in NVT and NPT ensembles, respectively, using the velocity scaling thermostat[82] and the Berendsen barostat[83] at temperature $T = 300$ K and pressure $P = 1$ atm. The time constant of the thermostat was 1 ps, and that of the barostat was 5 ps with a compressibility of $3 \times 10^{-4}$ bar$^{-1}$. For a 10-μs long production run in the NPT ensemble, we used a velocity scaling thermostat[82] and a Parrinello–Rahman barostat[84] with time constants of 1 ps and 12 ps, respectively. As shown in Supplementary Fig. 9b, the probability distribution of the end-to-end distance of the Martini model peaked at the same position as that of the CG model with $\sigma \equiv 0.6$ nm, and the averaged end-to-end distance of both models were equal. In the following, we fixed the FENE bond length at $\sigma \equiv 0.6$ nm

In a second step, we adjusted the interaction energy between the FG beads ($\tilde{\epsilon}$) by matching the model to the measured thermodynamic

properties of the NUP98 FG chain (1–499) condensate formation[27,54]. As shown in Extended data Fig. 10a, 500 chains were simulated inside a simulation box size of $360 \times 90 \times 90\ \sigma^3$. We adjusted the interaction strength $\tilde{\epsilon}$ to match the experimentally measured concentration of a NUP98 FG condensate[27,54] (see Extended data Fig. 10b). We note that in this way, we also got a good fit to the measured concentration of the dilute phase and thus the transfer free energy. The simulations of the condensate and NPC systems were performed for durations exceeding $1.8 \times 10^6\ \tau$ and $6 \times 10^4\ \tau$, respectively. We used the VMD software to visualize all systems[85].

**Comparison to smFRET and FLIM–FRET experiments.** We calculated the root-mean-square distances between the labelled sites across NUP98 FG chains and simulation trajectories to compare to the smFRET and FLIM–FRET experiments. As in the experiments, we kept one site fixed (amino acid position 221 for NUP98 in the NPC) and swept across the C-terminal residues. For an isolated NUP98 FG (1–499) chain, the results are shown in Supplementary Fig. 9c. We found that the experiments are best explained with a NUP–NUP interaction strength of $\tilde{\epsilon} = 0.5$, which is mostly in the collapsed state (Supplementary Video 8). According to the radius of gyration of single chains (Supplementary Fig. 9a), the midpoint of the coil–globule transition is at $\tilde{\epsilon} = 0.44$. We conclude that isolated NUP98 FG chains in aqueous solution are mostly collapsed.

### Reporting summary

Further information on research design is available in the Nature Portfolio Reporting Summary linked to this article.

## Data availability

The NPC scaffold used in the MD simulation is PDB ID: 7R5K. The source data for the main and extended data figures as well as the coordinates for MD simulations are provided in Supplementary Information. All other data are available in the main text or Supplementary Information. Source data are provided with this paper.

## Code availability

Initial configurations and trajectories of the MD simulations are available at https://doi.org/10.5281/zenodo.7648957 under CC-BY license.

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

**Acknowledgements** We thank all of the members of the Lemke laboratory for helpful discussions; the core facilities of the Faculty of Biology at Johannes Gutenberg University Mainz; and the Protein Production Core Facility at the Institute of Molecular Biology Mainz for expert assistance. M.Y. was funded by the MSCA Individual Fellowship (TFNUP 89410) and a Humboldt Research Fellowship for Postdoctoral Researchers. E.A.L. acknowledges funding from the ERC-ADG grant 'MultiOrganelleDesign', as well as CRC1551 'Polymer concepts in cellular function' of the Deutsche Forschungsgemeinschaft (DFG project number 464588647) and SPP2191 (DFG project number 419070619). E.A.L., G.H. and M.B. acknowledge funding from CRC 1507 'Membrane-associated protein assemblies, machineries, and supercomplexes' of the Deutsche Forschungsgemeinschaft (DFG project number 450648163). M.H., A.O.-K., M.S., M.B. and G.H. thank the Max Planck Society for support.

**Author contributions** E.A.L. and G.H. conceived the project. M.Y., S. Mikhaleva, P.S.T. and S. Mingu designed and performed the cell experiments. M.Y., S. Mingu and H.R. designed and performed the in vitro-reconstituted condensates and smFRET experiments. M.H. performed the MD simulations with help from M.S. M.Y. and M.H. analysed the experimental and simulation data, and co-wrote the original draft together with E.A.L. and G.H. C.D.R., A.O.-K. and M.B. contributed analytical tools or reagents. All authors contributed to manuscript editing and approved the final manuscript.

**Competing interests** The authors declare no competing interests.

**Additional information**
**Correspondence and requests for materials** should be addressed to Gerhard Hummer or Edward A. Lemke.

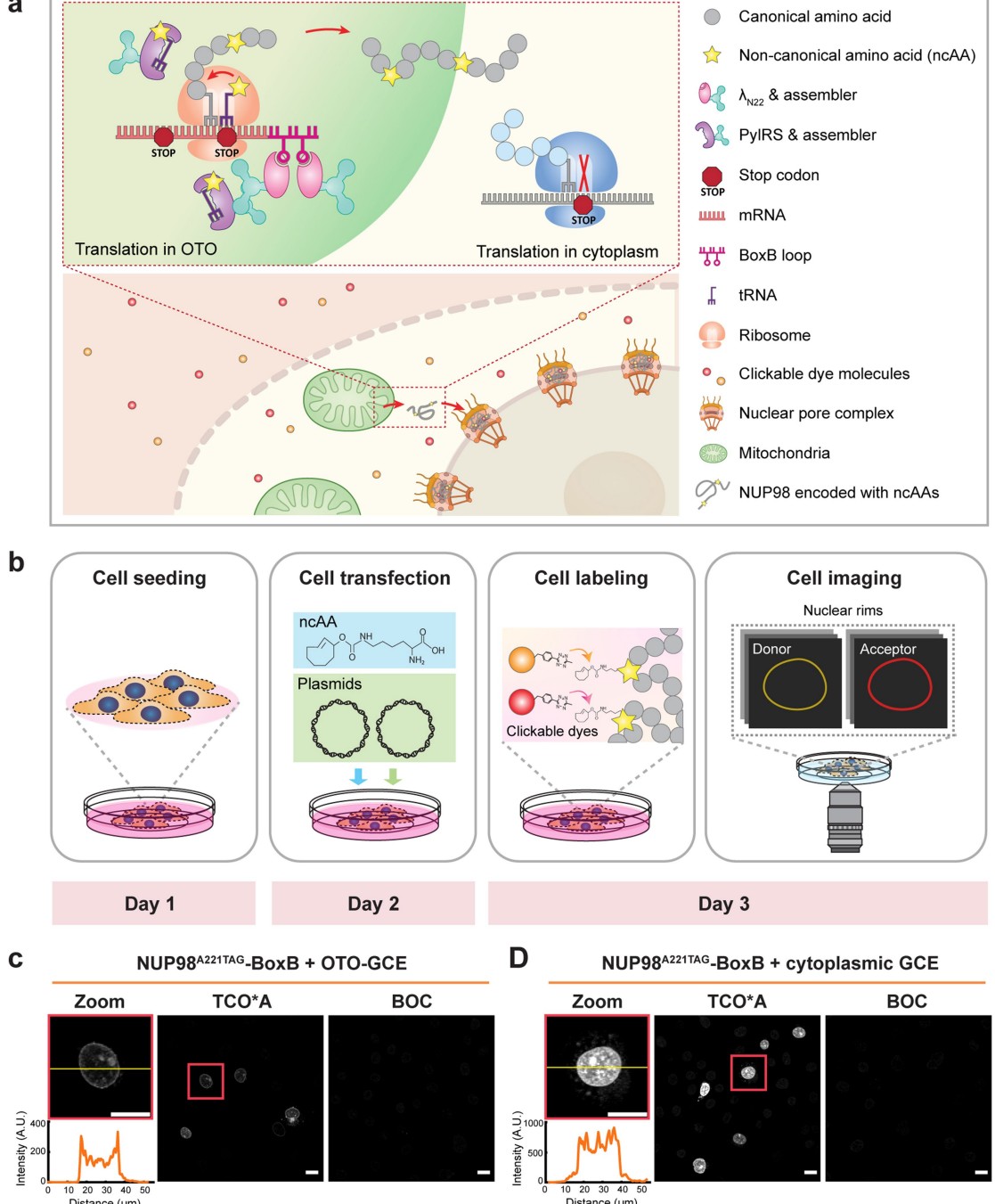

**Extended Data Fig. 1 | Comparison of genetic code expansion (GCE) using cytoplasmic and orthogonal translating film-like organelles (OTOs).** **(a)** Schematic of the OTOs that form a distinct protein translational machinery on the outer mitochondrial membrane surface. The OTOs preferably expanded the genetic codon of the target NUP98, ensuring minimal interference with the endogenous protein translation in the cytoplasm. Non-canonical amino acids (ncAAs) were introduced into NUP98 at the sites specified by amber mutation. The genetically encoded NUP98 with ncAAs incorporated into the nuclear pore complex. The ncAAs were labelled with tetrazine-modified organic dyes *via* click chemistry. **(b)** Schematic showing sample preparation pipeline including cell seeding, cell transfection with NUP98 and OTO-GCE plasmids, labelling the incorporated ncAAs with clickable dyes, and fluorescent lifetime imaging of labelled NUP98 on the nuclear rims using the custom-built FLIM-FRET optical setup. COS-7 cells were co-transfected with NUP98^A221TAG-boxB plasmid and **(c)** OTO-GCE system or **(d)** cytoplasmic GCE system in the presence of *trans*-cyclooct-2-en-L-lysine (TCO*A, an ncAA that can be labelled) or t-butyloxycarbonyl-L-lysine (BOC, a control ncAA that cannot be labelled), and labelled with LD655-tetrazine. Clearer nuclear rims and less off-target-amber-suppression-induced background labelling were observed in OTO-GCE system compared to the cytoplasmic GCE system. The BOC control groups prove that the observed off-target-labelling was not only due to non-specific sticking of the dye molecules. (Scale bars: 50 μm).

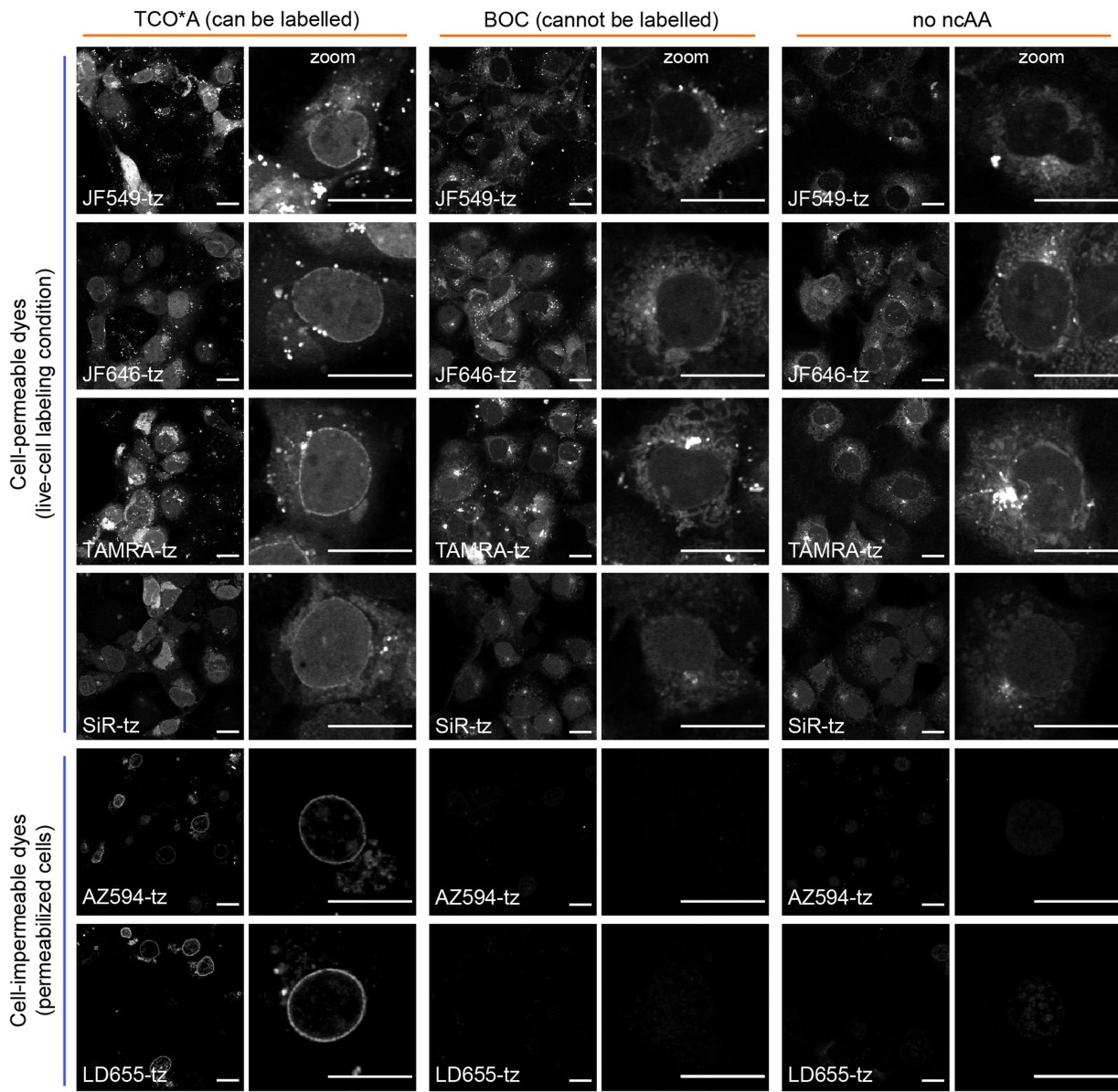

**Extended Data Fig. 2 | Labelling of NUP98[221TAG] with various cell-impermeable and cell-permeable tetrazine dyes in COS-7 cells in the presence of TCO*A or BOC or no ncAA.** The reporter protein was expressed in the presence of trans-cyclooct-2-en-L-lysine (TCO*A, an ncAA that can be labelled) or t-butyloxycarbonyl-L-lysine (BOC, a control ncAA that cannot be labelled). Cells are labelled with the cell-permeable dyes (JF549, JF646, SiR, and TAMRA) under live cell conditions. For the cell-impermeable dyes (AZDye594 and LD655), we permeabilized the plasma membrane with low-dosage digitonin in order to deliver the dyes across the plasma membrane. (Scale bars: 20 μm).

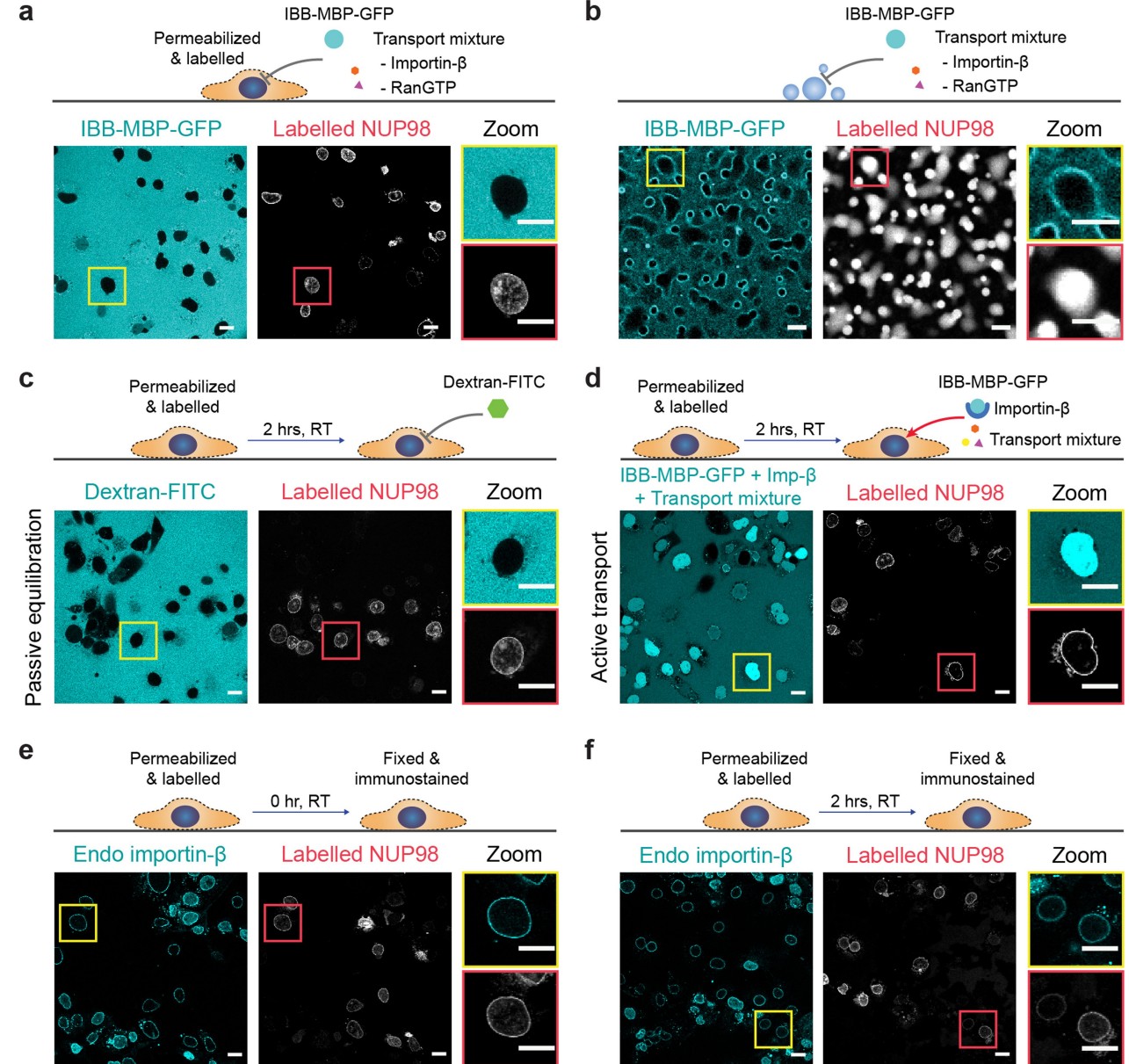

**Extended Data Fig. 3 | Control experiments of transport functional assay.**
IBB-MPB-GFP cargo in the transport mixture without importin-β and RanGTP
was not imported into (**a**) the labelled COS-7 cells (Scale bars: 20 μm) or (**b**) the
*in vitro* NUP98 FG-condensates. (Scale bars: 5 μm.) 2 hrs after labelling, the
permeabilized COS-7 cells maintained their transport functions in (**c**) passive
exclusion assay (70 kDa dextran was excluded) and (**d**) facilitated/active

transport assay (IBB-MBP-GFP supplied with transport mixture was imported).
Immunofluorescence reveals that endogenous importin-β colocalized at the
nuclear envelope directly after (**e**) permeabilization and labelling, and (**f**) was
retained after 2 hrs. For each condition, *n* = 3 experiments were repeated
independently with same conclusion.

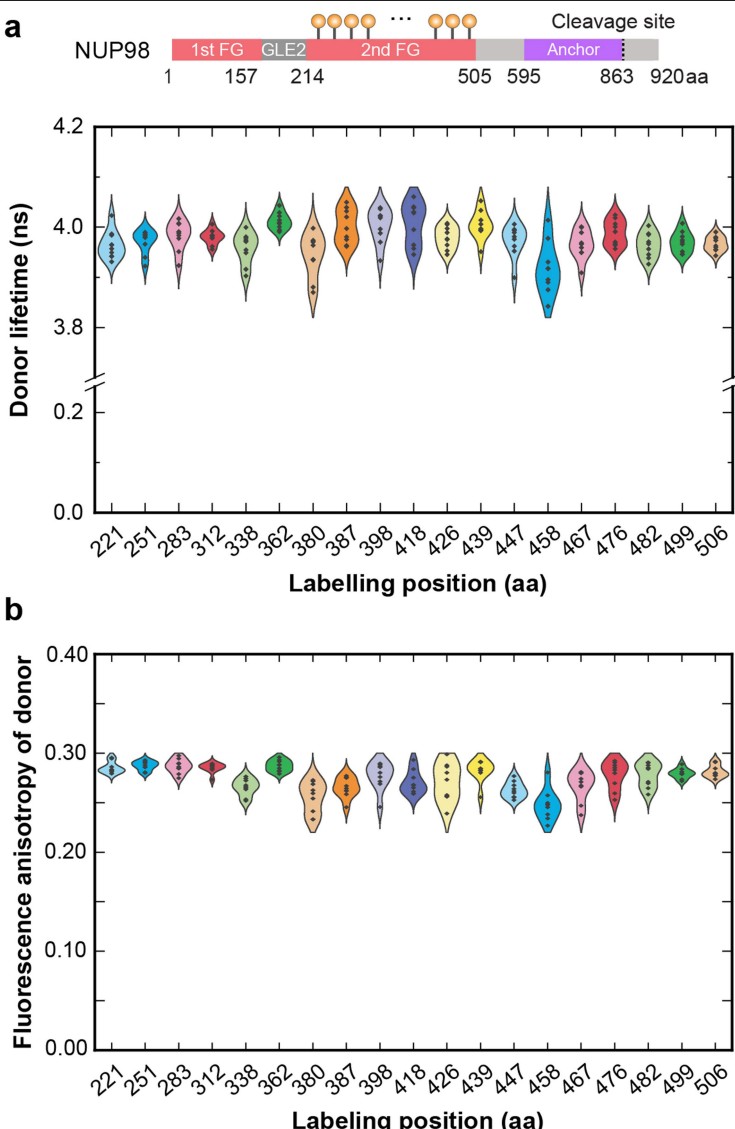

**Extended Data Fig. 4 | Measurements of donor fluorescence lifetime and anisotropy among different labelled sites inside the NPC.** Different positions along the FG-domain of NUP98 labelled with the donor dye showed similar **(a)** fluorescence lifetime and **(b)** fluorescence anisotropy of the donor dye inside the NPC ($n$ = 9 cells for each mutant), indicating the fluorophores at different labelling sites shared similar fluorescent properties.

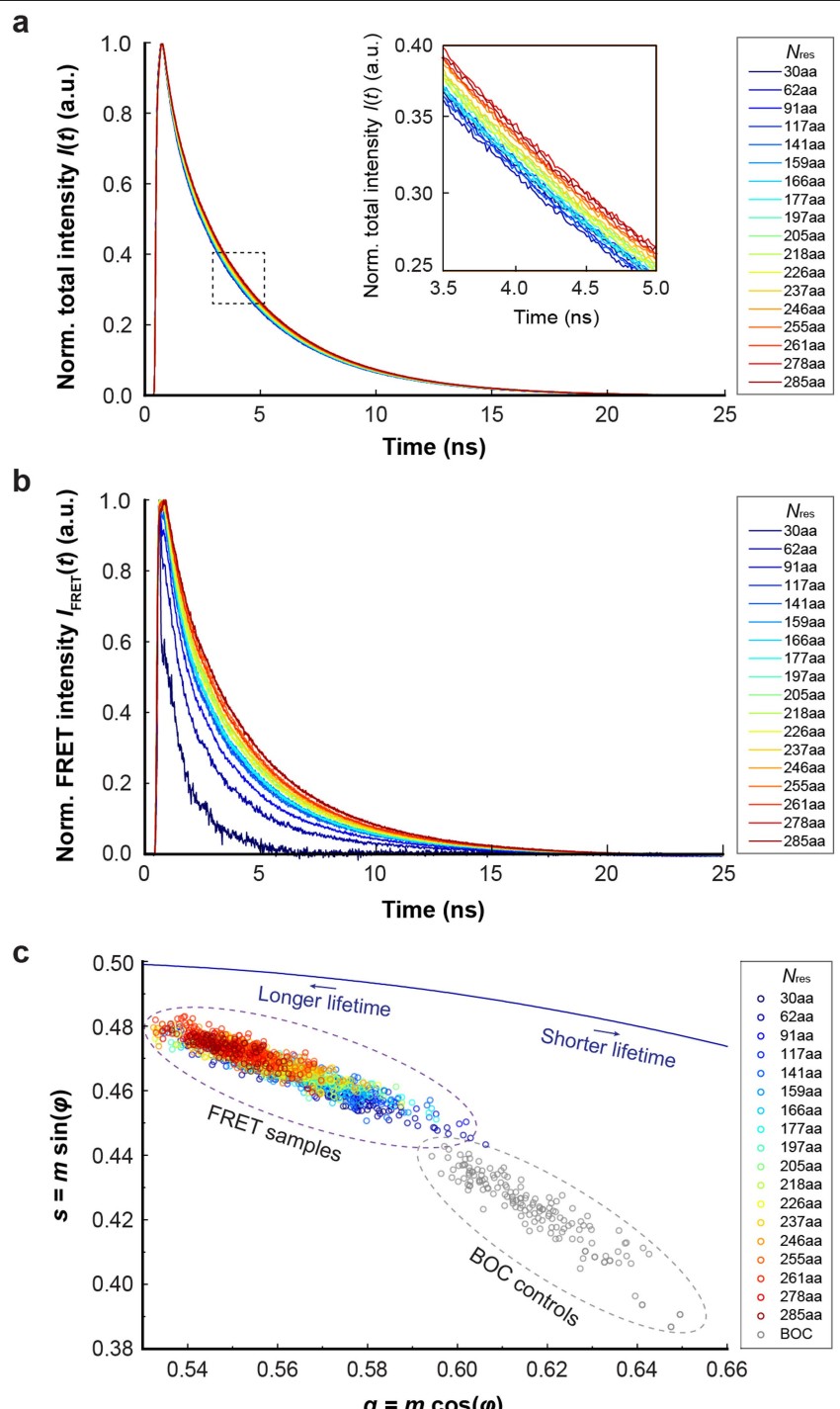

**Extended Data Fig. 5 | Fluorescent lifetime analysis of the eighteen FLIM-FRET chain segments probed within the NUP98 FG domain inside the NPC.** **(a)** Normalized total fluorescence intensity profiles of the donor channel $I(t)$ for the eighteen chain segments of NUP98 FG domain for the FLIM-FRET measurements in functional NPCs. Each profile represents an averaged result of ~100 cells. **(b)** Normalized FRET intensity profiles $I_{FRET}(t)$ by subtracting the fitted donor-only signal $I_{Donly}(t)$ and fitted background signal $I_{bg}(t)$ from the total signal $I(t)$ in (a) by using Supplementary equation 15. **(c)** Extended phasor plot for Fig. 2f where the lifetime decays of NUP98 encoded with a nonreactive ncAA BOC are also included. The lifetime decays of the FRET samples could be well separated from the BOC controls. The averaged lifetime decay of the BOC controls was used to determine the cellular background signal for the lifetime fitting procedure.

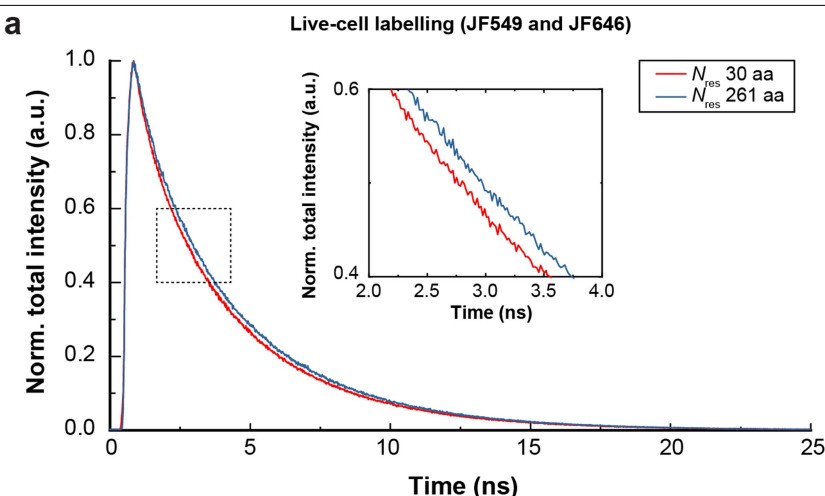

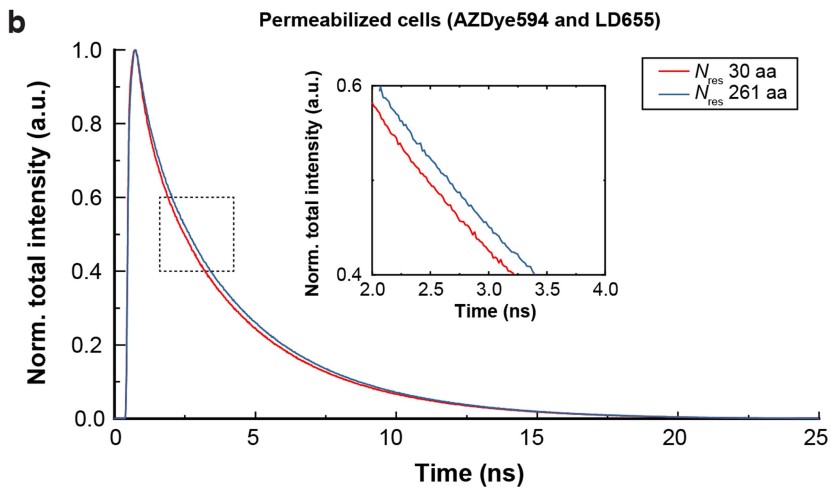

**Extended Data Fig. 6 | FLIM-FRET measurements of NUP98 inside the NPC show qualitative agreement between live cells and permeabilized cells.** (a) Fluorescent lifetime curves of double-amber-mutated sample NUP98$^{221TAG-251TAG}$ ($N_{res}$ = 30 aa) and NUP98$^{221TAG-482TAG}$ ($N_{res}$ = 261 aa) labelled with donor (JF549) and acceptor dye (JF646) mixture. Each profile represents an averaged result of 6 cells. (b) Fluorescent lifetime curves of double-amber-mutated samples same as in (a) but labelled with donor (AZDye594) and acceptor dye (LD655) in permeabilized cells (same dataset as in Extended Data Fig. 5a for the selected mutants).

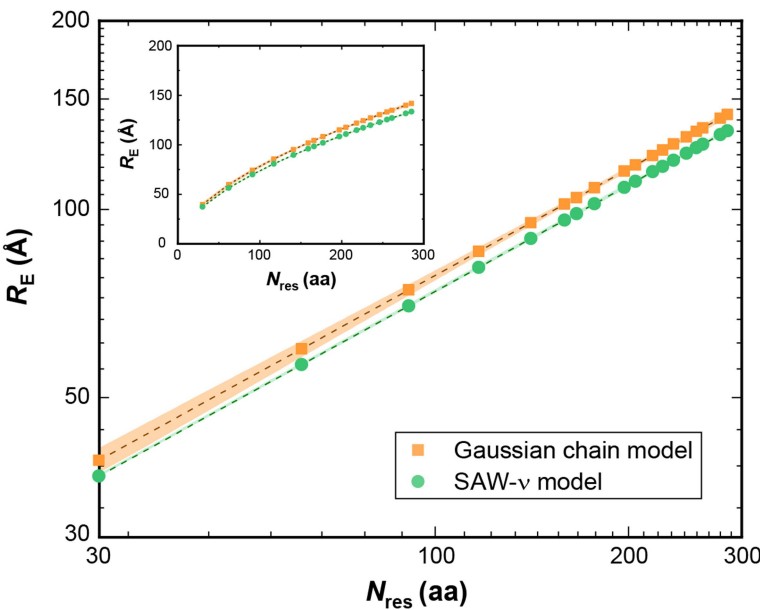

**Extended Data Fig. 7 | Comparison of the root-mean-square inter-residue distance $R_E$ obtained from global fitting with the Gaussian chain model and the SAW-$\nu$ model.** Global fitting was performed by fitting the lifetime decays of all mutants over ~2000 cells with Supplementary equations 15, 16 and 18 for the Gaussian chain model and Supplementary equations 18 and 19 for the SAW-$\nu$ model. We obtained $\nu = 0.56 \pm 0.03$ (Gaussian chain model) and $\nu = 0.56 \pm 0.001$ (SAW-$\nu$ model).

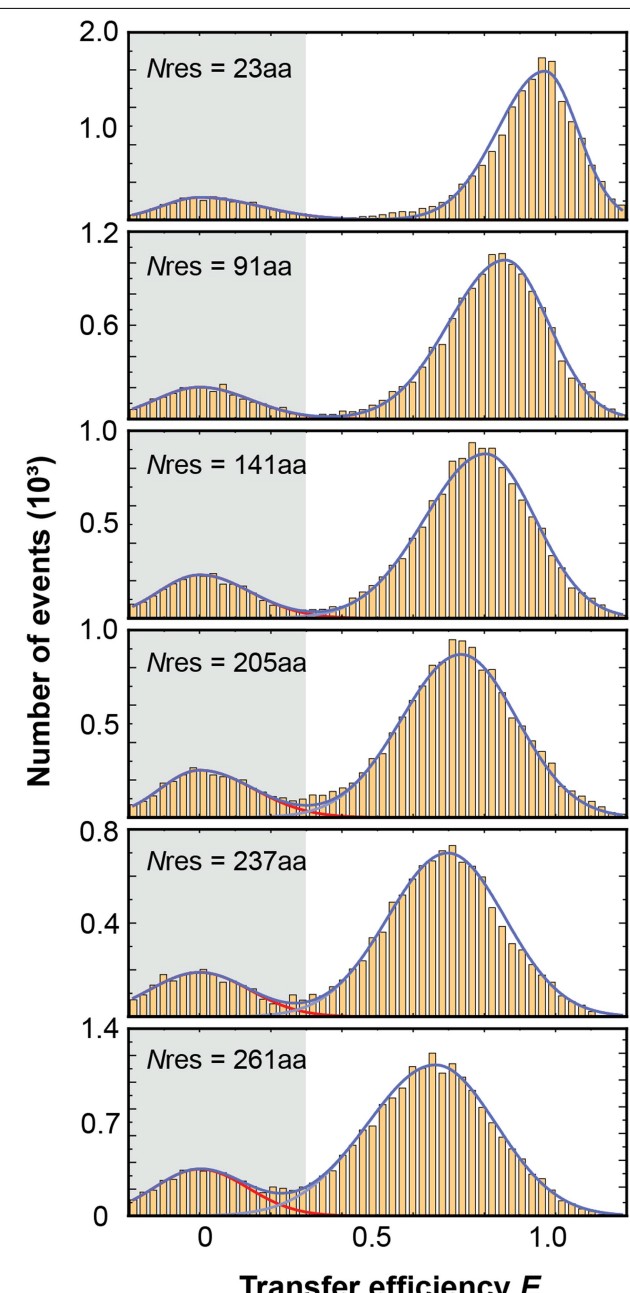

**Extended Data Fig. 8 | Single-molecule FRET measurements in solution.**
FRET efficiency histograms for different mutants of purified NUP98
FG-domain in 1x PBS. The peaks in the grey shadow indicate the donor-only
population. The FRET population was fitted with an asymmetric-Gaussian
function to retrieve the FRET efficiency.

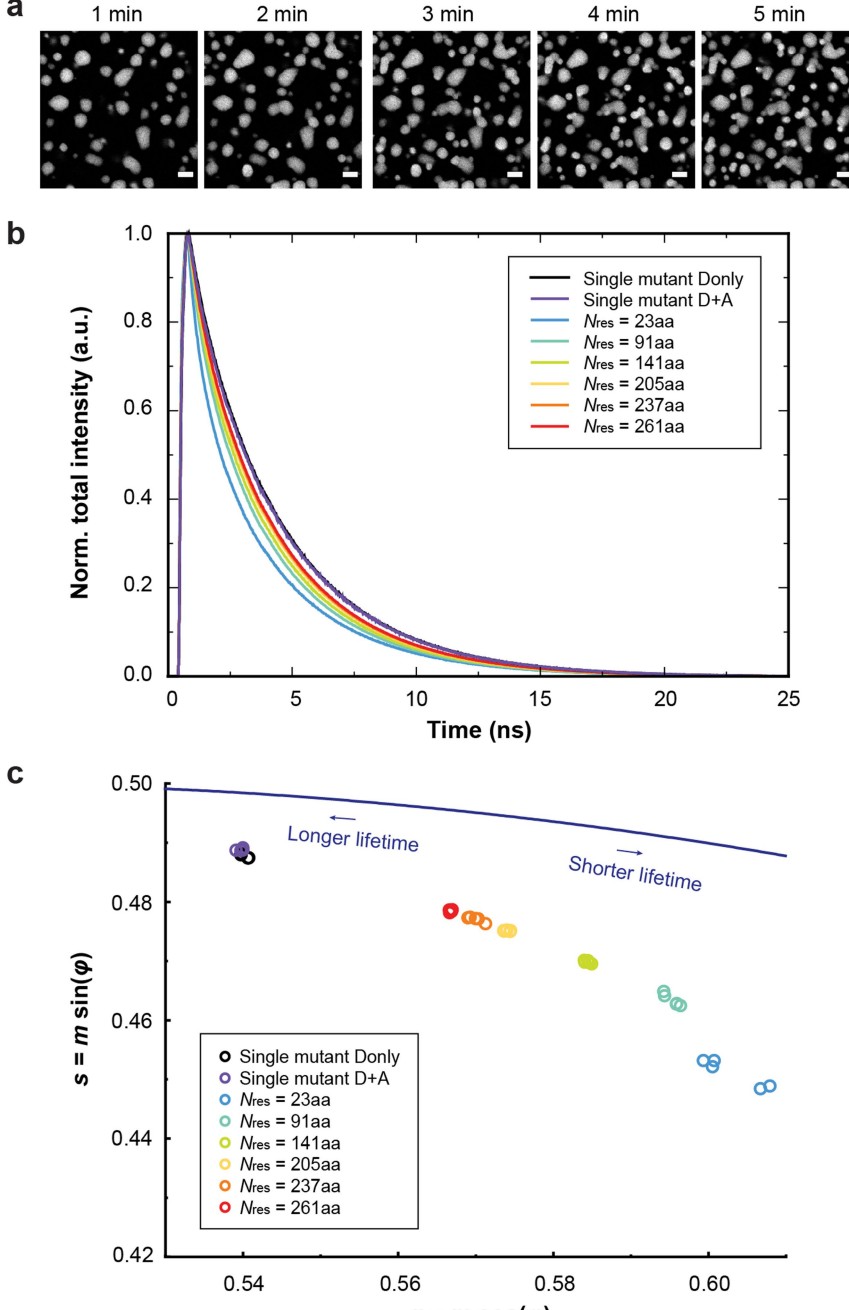

**Extended Data Fig. 9 | FLIM-FRET analysis of NUP98 FG condensates *in vitro*.**
**(a)** Time-lapse images showing phase separation of the purified NUP98 FG domain *in vitro*. The early stage of the formed condensates demonstrated liquid-like behavior where fast merging events were observed in line with other observations of the early state of NUP98 FG droplets before they harden out to gels[26]. Scale bar 5 μm. **(b)** Normalized donor fluorescence intensity profiles of six NUP98 double-mutants in phase-separated condensates that were allowed to settle on the coverslip (*n* = 5 independent experimental repeats). $N_{res}$ refers to the number of amino acid residues between the two labelled sites. The normalized donor fluorescence intensity profiles of the single-mutant NUP98[A221C] labelled with donor only or donor and acceptor mixture showed no difference, indicating no intermolecular FRET was detected and the lifetime differences detected in the double-mutants were due to intramolecular FRET. **(c)** Phasor plot showing the donor lifetimes of the six double-mutants for individual experimental repeats (*n* = 5). On the same phasor plot, the lifetimes for the single-mutant labelled with donor-only and a mixture of donor and acceptor were also compared, which showed no difference, ensuring no intermolecular FRET was involved during the measurements.

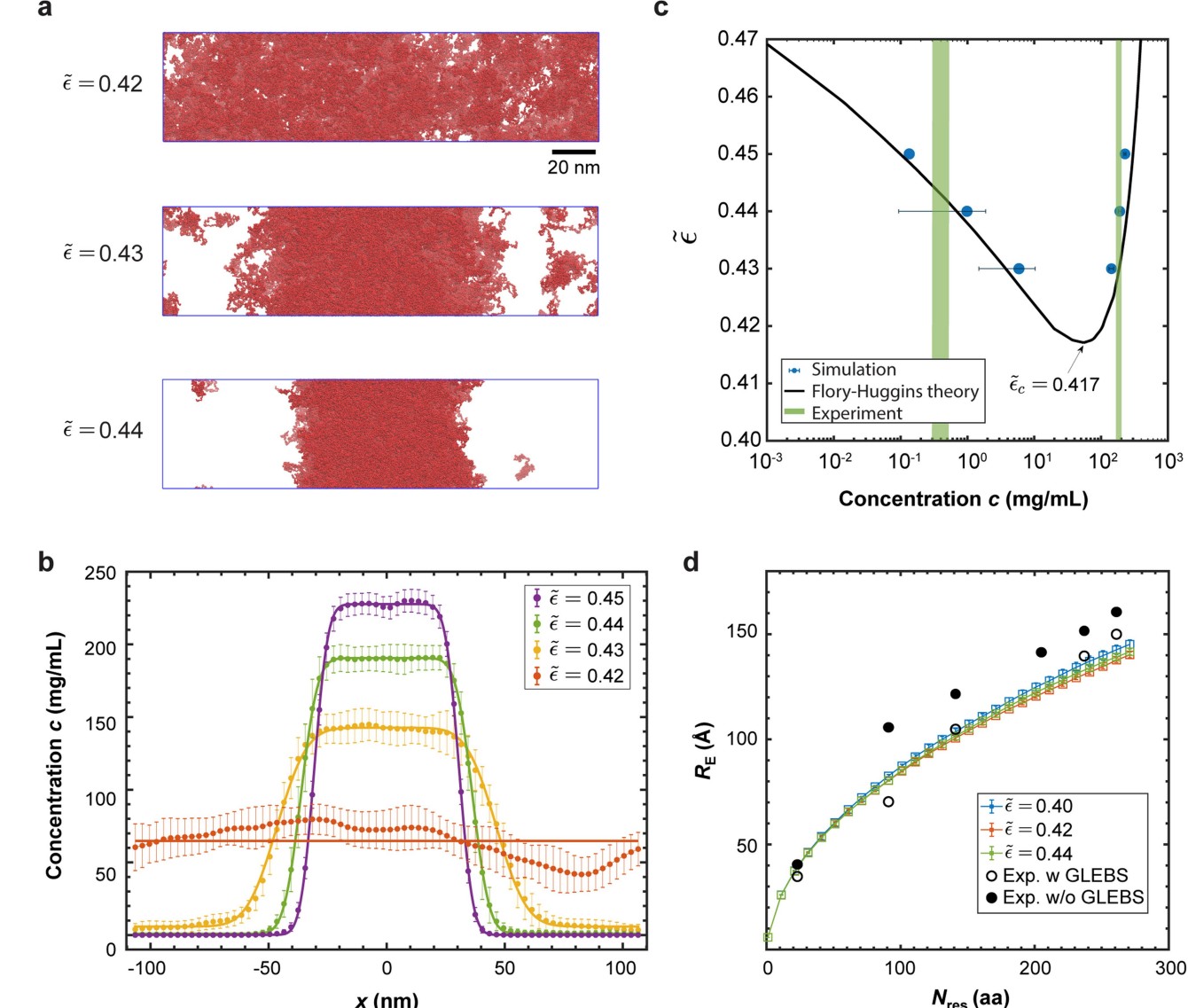

**Extended Data Fig. 10 | Coarse-grained MD simulations of NUP98 FG condensates. (a)** Condensate formation of homopolymer model of NUP98 FG chain at different cohesive interaction strengths $\tilde{\epsilon}$. Shown are side-views of an elongated simulation box (blue outline) with 500 NUP98-FG (1-499) chains (red). **(b)** Concentration profiles of the NUP98 FG system at different cohesive interaction strengths. The symbols and error bars represent the average and standard deviation, respectively, of time-averaged density profiles over $3\times10^5\tau$. For $\tilde{\epsilon} = 0.42$, at the critical interaction strength, the condensate is not stable and the chains are rather uniformly distributed inside the system. **(c)** Phase diagram of the NUP98 FG condensate in the plane of cohesive interaction strength $\tilde{\epsilon}$ and the concentration c of coexisting phases. The solid black line is the result of Flory-Huggins theory for a homopolymer of a length $N = 499$. The symbols and error bars represent the average and standard deviation, respectively, of time-averaged densities of dense and dilute phases over $3\times10^5\tau$. The critical cohesive strength

is obtained as $\tilde{\epsilon}_c \approx 0.417$. For $\tilde{\epsilon} = 0.45$, no chain escaped the condensate during the MD simulation. Based on capillary wave theory, we estimated the dilute-phase density by fitting a double error function to the averaged concentration profile, $c(x) = A\{\text{erf}[(x + B)/w] - \text{erf}[(x - B)/w]\} + c_{\text{dilute}}$, where $A = (c_{\text{dense}} - c_{\text{dilute}})/2\text{erf}(B/w)$. For $\tilde{\epsilon} = 0.45$, we obtained $c_{\text{dense}} = 227.7$ mg/mL, $c_{\text{dilute}} = 0.13$ mg/mL, $B = 30.69$ nm and $w = 6.41$ nm. The experimental values of the dense and dilute phases are marked in green bars[27,54]. **(d)** Root-mean-square inter-residue-distance $R_E$ of beads on the same NUP98 FG chain inside the condensate as a function of residue separation $N_{\text{res}}$. The symbols and error bars represent the average and standard error of the mean, respectively, as estimated from four blocks of size $8\times10^4\tau$. Issues with condensate aging and gelation during the FLIM-FRET experiments (empty circles: with GLEBS; filled circles: without GLEBS) likely explain the larger distances compared to the simulations.

# Reporting Summary

## Statistics

For all statistical analyses, confirm that the following items are present in the figure legend, table legend, main text, or Methods section.

| n/a | Confirmed | |
|---|---|---|
| ☐ | ☒ | The exact sample size (*n*) for each experimental group/condition, given as a discrete number and unit of measurement |
| ☐ | ☒ | A statement on whether measurements were taken from distinct samples or whether the same sample was measured repeatedly |
| ☐ | ☒ | The statistical test(s) used AND whether they are one- or two-sided<br>*Only common tests should be described solely by name; describe more complex techniques in the Methods section.* |
| ☐ | ☒ | A description of all covariates tested |
| ☐ | ☒ | A description of any assumptions or corrections, such as tests of normality and adjustment for multiple comparisons |
| ☐ | ☒ | A full description of the statistical parameters including central tendency (e.g. means) or other basic estimates (e.g. regression coefficient) AND variation (e.g. standard deviation) or associated estimates of uncertainty (e.g. confidence intervals) |
| ☐ | ☒ | For null hypothesis testing, the test statistic (e.g. *F*, *t*, *r*) with confidence intervals, effect sizes, degrees of freedom and *P* value noted<br>*Give P values as exact values whenever suitable.* |
| ☒ | ☐ | For Bayesian analysis, information on the choice of priors and Markov chain Monte Carlo settings |
| ☒ | ☐ | For hierarchical and complex designs, identification of the appropriate level for tests and full reporting of outcomes |
| ☐ | ☒ | Estimates of effect sizes (e.g. Cohen's *d*, Pearson's *r*), indicating how they were calculated |

*Our web collection on statistics for biologists contains articles on many of the points above.*

## Software and code

Policy information about availability of computer code

| Data collection | Symphotime v2.6 |
|---|---|
| Data analysis | Symphotime v2.6, PAM-PIE(https://gitlab.com/PAM-PIE/PAM), Matlab R2021a, LAMMPS(stable version 29Sep2021-Update3) software package, GROMACS(version 2020.6), VMD(version 1.9.3). |

For manuscripts utilizing custom algorithms or software that are central to the research but not yet described in published literature, software must be made available to editors and reviewers. We strongly encourage code deposition in a community repository (e.g. GitHub). See the Nature Portfolio guidelines for submitting code & software for further information.

## Data

Policy information about availability of data

All manuscripts must include a data availability statement. This statement should provide the following information, where applicable:
- Accession codes, unique identifiers, or web links for publicly available datasets
- A description of any restrictions on data availability
- For clinical datasets or third party data, please ensure that the statement adheres to our policy

The NPC scaffold used in the MD simulation is PDB ID: 7R5K. The source data for the main and extended data figures as well as coordinate for MD simulations are provided in supplementary information. Initial configurations and trajectories of the molecular dynamics simulations are available at https://doi.org/10.5281/zenodo.7648957 under CC-BY license. All other data are available in the main text or supplementary information.

## Human research participants

Policy information about studies involving human research participants and Sex and Gender in Research.

| | |
|---|---|
| Reporting on sex and gender | NA |
| Population characteristics | NA |
| Recruitment | NA |
| Ethics oversight | NA |

Note that full information on the approval of the study protocol must also be provided in the manuscript.

# Field-specific reporting

Please select the one below that is the best fit for your research. If you are not sure, read the appropriate sections before making your selection.

☒ Life sciences ☐ Behavioural & social sciences ☐ Ecological, evolutionary & environmental sciences

For a reference copy of the document with all sections, see [nature.com/documents/nr-reporting-summary-flat.pdf](http://nature.com/documents/nr-reporting-summary-flat.pdf)

# Life sciences study design

All studies must disclose on these points even when the disclosure is negative.

| | |
|---|---|
| Sample size | The sample size (~100 cells per mutant) in FLIM-FRET measurements was chosen so that the total photon numbers are sufficient enough for a very robust fitting of the fluorescent lifetime decay of together 18 mutants. |
| Data exclusions | No data were excluded from the analyses. |
| Replication | The transport assay of labelled cells and reconstituted in vitro condensates was repeated 3 times independently with same conclusion. The acceptor photobleaching assay to check intermolecular and intramolecular FRET was repeated 5 times independently with same conclusion. All attempts at replication were successful. |
| Randomization | Cells were seeded, transfected and labelled in imaging dishes, and the red fluorescent channel was used to guide the eye to identify cells with a fluorescently labelled nuclear rim. After image acquisition, a threshold was applied (as explained in Methods) to exclude cells with too high expression levels from further analysis. No further randomization was applied, all measured cells were analysed. |
| Blinding | The investigators were blinded to group allocation during data collection and analysis. |

# Reporting for specific materials, systems and methods

We require information from authors about some types of materials, experimental systems and methods used in many studies. Here, indicate whether each material, system or method listed is relevant to your study. If you are not sure if a list item applies to your research, read the appropriate section before selecting a response.

### Materials & experimental systems

| n/a | Involved in the study |
|---|---|
| ☐ | ☒ Antibodies |
| ☐ | ☒ Eukaryotic cell lines |
| ☒ | ☐ Palaeontology and archaeology |
| ☒ | ☐ Animals and other organisms |
| ☒ | ☐ Clinical data |
| ☒ | ☐ Dual use research of concern |

### Methods

| n/a | Involved in the study |
|---|---|
| ☒ | ☐ ChIP-seq |
| ☒ | ☐ Flow cytometry |
| ☒ | ☐ MRI-based neuroimaging |

## Antibodies

| | |
|---|---|
| Antibodies used | Anti-KPNB1 antibody (Abcam, ab2811) and anti-mouse-Alexa Fluor 488 secondary antibody (ThermoFisher, A-11001) |
| Validation | Antibodies were validated by the manufactures (for Anti-KPNB1, see https://www.abcam.com/kpnb1-antibody-3e9-ab2811.pdf; for |

| Validation | anti-mouse-AF488, see https://www.thermofisher.com/antibody/product/Goat-anti-Mouse-IgG-H-L-Cross-Adsorbed-Secondary-Antibody-Polyclonal/A-11001). |
|---|---|

## Eukaryotic cell lines

Policy information about cell lines and Sex and Gender in Research

| Cell line source(s) | COS-7 (Sigma 87021302) |
|---|---|
| Authentication | Authenticated by the manufacturer. Validation by morphology. |
| Mycoplasma contamination | All cell lines were regularly tested for mycoplasma contamination, with negative results. |
| Commonly misidentified lines (See ICLAC register) | No misidentified lines were used. |

