## [Peer Review File · Nature]

Manuscript Title: Visualising the disordered nuclear transport machinery in situ

Reviewer Comments & Author Rebuttals

Reviewer Reports on the Initial Version:

Referee #1 (Remarks to the Author):

The manuscript by Yu and colleagues explore the microenvironment within the nuclear pore complex, which is composed of intrinsically disordered protein domain FG-NUPs. They achieved this by first selectively incorporating into these proteins a ncAA at two distinct sites. The side chains of these nCAAs could be subsequently used to stochastically install a FRET pair in live cells. The authors demonstrate that the labeled NPCs retain proper transport function. They also confirm that the resonance energy transfer observed from these reconstituted, labelled NPCs are largely intramolecular and not intermolecular. Finally, the FLIM-FRET measurement of these labeled NPCs, and interfacing these dynamic distance measurements with existing structural information and computational simulation were used to gain insights into the dynamic nature of this microenvironment, which could not be structurally characterized due to its intrinsically disordered nature.

Although I am not an expert in this area, the ability to make dynamic distance measurements of IDPs in unstructured microenvironments within living cells strikes me as an important advance. The use of ncAA technology to facilitate site-selective labelling with minimal perturbation was key to achieving this. Application of this technology for intracellular labelling has been traditionally challenging, because of background noise from concurrent incorporation of the ncAA at endogenous UAG codons and their subsequent labelling. The authors also offer an elegant solution to this limitation by employing their recently developed film-like organelles, that enable mRNA-selective ncAA incorporation.

However, there are some concerns that should be addressed before publication:

1. The ncAA incorporation technology is highly context-dependent, which typically results in dramatically altered expression levels of different UAG mutants for the same protein. This issue is likely compounded when suppressing two UAGs in the same mRNA. Such significant differences in expression levels would likely impact subsequent measurements. Yet, there is little data offered here for characterizing these aspects for different mutants. The only data we see is post-labeling images, which lack much of the critical information. What are the relative expression levels for different double mutants? Do the truncated proteins (those failing to incorporate nCAAs) create complications, and how is this issue addressed? Is the fluorophore-labelling efficiency at different sites variable (differences in protein microenvironment) and does that impact measurements?
2. A key advance here is mRNA-selective ncAA/fluorophore tagging in living cells. While theoretically it makes sense, the authors do not provide any supporting data characterizing to what extent the OTO system reduces off-target labeling and background reduction. Given the potential of this

strategy for additional applications, the authors should provide data clarifying this aspect

3. As a non-expert in this area, it was not fully clear to me what the key take-home messages from the measurements made in the NPC were. If the authors could discuss this aspect in a more succinct way that is accessible to a general audience, and particularly highlight advances in our knowledge relative to the previous state-of-the-art, it would significantly strengthen the paper.

Referee #2 (Remarks to the Author):

This timely contribution comes on the heels of a series of papers that have provided detailed, tomographic maps showing the three dimensional structure and organization of the nuclear pore complex from different cell and species types. All these structures, impressive as they are, feature a prominent hole in the central lining of the pore because these FG-NUPs are conformationally heterogeneous, going by the name of intrinsically disordered regions. In this work, the authors combine a fluorescence lifetime and single molecular Förster resonance energy transfer experiments, focusing on NUP98, to provide a conformational annotation of a key NUP within the family of FG-NUPs. Characterization of the conformational ensemble is enabled by the use of synthetic biology aided selective labeling along the sequence of NUP98. These measurements, which are unprecedented, help lead to a statistical description of the conformational ensembles of NUP98. The key findings are that the ensembles within the pore, in live cells, are distinct from the ensembles sampled in dilute solution, untethered from the pore. Interestingly, the conformational ensembles bear statistical resemblance to those adopted in the context of condensates formed by NUP98. The intriguing inference presented is that the apparent solvent quality of nuclear pore is better than that of a bulk, aqueous solvent, with conformational statistics measured in the pore resembling what one would observe in a good solvent. The congruence between the conformational statistics within the pore and in condensates formed in vitro endorses the Görlich view of stickers-and-spacers architecture and mesh-like properties of the pore. Overall, this is a very timely contribution, of urgent importance, that is very well written. A key feature is the range of technologies that have been brought to bear. Additionally, the number of controls and the photo physical rigor are to be lauded. This is the only group of investigators that could have pulled off this contribution, so the originality and novelty are beyond dispute.

There are specific issues that came up during the reading of the MS that should be addressed in a suitable revision. These are as follows:

1) For the uninitiated, the phasor plots are going to be difficult to follow. The captions for these plots would benefit from being more informative, pedagogical even. Explain what is being plotted, what it means, and why these plots provide unequivocal insights. This is not clear at this juncture. Based on the Fourier transform pair shown in the methods section, one would have thought that the plane on which the analysis would be performed would be the amplitude and phase angle. Simply put, there are two issues with the phasor plots: As presented, they are unclear and the information to be processed by the reader is ambiguous. Second, it seems, based on the fact that these are Fourier transforms of the intensity data, that the analysis affords deeper insights that are not being pursued.

2) The apparent scaling exponent of 0.56 ± 0.05 , with the error bar, has a spread that ranges from that of an apparent theta solvent to that of an apparent good solvent. That is a significant range. The first question is if the errors are intended to be symmetrical? Second, if the conformational statistics are those of self-avoiding walks, i.e., if NUP98 within the pore belongs to the same universality class as self-avoiding walks, then the expectation would be that because of finite size, the inferred exponent should be 0.6 or larger. Instead, given the inferred exponent and the observation of significant compaction in dilute solutions, there is the very real possibility that the conformational ensemble might be a mixture of two populations viz., an ensemble of compact conformations and an ensemble of self-avoiding walks, which when mixed in a suitable proportion could give rise to an apparent exponent of 0.56. This would imply that the pore likely encompasses coexisting conformational ensembles. Can this be ruled out? If not, it would be useful to include this as a formal possibility in the discussions - see Ref. 53. Unfortunately, the simulations will not be able to provide clarity since they are simulations of an effective homopolymer.

3) The authors propose, based on the calculated phase diagram, that the tethered NUP98 system, modeled as a tethered homopolymer, is akin to this homopolymer at its critical point. First, it is worth noting that the excluded volume limit is in fact a critical point - please see the authoritative book of Schäfer on this topic. This, as shown by other others (<http://dx.doi.org/10.1529/biophysj.106.086264>), enables the analysis of a variety of interesting features, specifically whether or not the tethered chains have a projected end-to-end distance distribution that matches (without any parameterization) the form predicted by des Cloizeaux, Jannink, and Fisher. If so, this will imply a correlation hole and also provide deeper insights into how cargo negotiate the meshwork enabled by the crosslinking of the meshwork laid out by the self-avoiding chains with stickers. In this context, a relevant paper is one that was published by Wei et al., - <https://www.nature.com/nchem/journal/v9/n11/full/nchem.2803.html>.

4) Finally, there are some statements or phrases that are either unclear or unsubstantiated. For example, the authors claim that the motions of the chains are extensive and fast. Neither of these claims can be substantiated by the data - unless they are implicit, in which case they should be made explicit. The authors refer to a "critical polymer mixture" in the discussion section. What do they imply by this phrasing? I thought the last paragraph undermined the substance of the work. There is no doubt that this is technological tour de force. However, focusing the summary claims about novelty on technology / methodology does disservice to the prospect that this work likely provides a clear denouement to a long-standing debate. To this point, the average reader might be inclined to ask about the other NUPs within the NPC. Clearly, tackling all these other NUPs will require further studies and perhaps advances to the technologies used here. It would be useful to mention the other NUPs and know if the authors think that their observations for NUP98 transfer seamlessly over to all other NUPs (my guess is no).

Referee #3 (Remarks to the Author):

While the scaffold of the nuclear pore complex (NPC) has been determined to molecular resolution

by a combination of cryo-EM and X-ray studies, the properties of the permeability barrier are less well-determined as it is comprised of hundreds of intrinsically disordered polypeptide chains, which present unique challenges. Molecular dynamics and modeling will certainly be key methods to resolve these dynamic properties, but they must be appropriately parameterized and verified by experimentally determined constraints. This paper reports on an important one of these constraints, i.e., the scaling exponent in Flory homopolymer theory for polypeptides within intact NPCs. The sophisticated state-of-the-art approach is technically complex, yet the beauty of the result is its simplicity. The authors use their recently developed noncanonical amino acid approach to site-specifically incorporate a FRET dye pair into a single FG-polypeptide (Nup98), which is incorporated into functionally competent NPCs. This impressive approach in eukaryotic (human) cells was reported earlier by the corresponding author's group; nonetheless, this paper displays a dramatic illustration of the powerful utility of the method. By varying the amino acid spacing between two dye labels and using FLIM-FRET to obtain average distances between the dye pair, the extension of the polypeptide was determined. This extension depends on the quality of the solvent, i.e., in a 'good' solvent (solubilizes the polypeptide chain), the polymer will be largely extended, whereas in a 'bad' solvent, the polymer will collapse upon itself. The authors determined that the Nup98 chain within NPCs is largely extended, and hence in a good solvent environment. This allows the permeability barrier to remain largely open and fluid, allowing it to both block traffic of diffusing molecules, and yet allow rapid signal-dependent transport.

This paper is well-written, well thought out, and presents a compelling story. The figures are well-made. The in cellulo dye-labeling strategy should be widely applicable to a broad array of cell biological problems. However, they do not perform the dye labeling reaction or their observations in live cells, reducing impact. There are multiple areas in the manuscript where additional clarity is needed.

It is unclear why the FLIM signal is the parameter of choice for obtaining distance information. The difference in lifetimes for the various FRET pairs are tiny (Extended Data Fig. 5a), and these lifetimes are never explicitly reported. Because these differences are so small, the differences in donor only lifetimes (Extended Data Fig. 3) can have a significant impact on individual measurements (noting that the donor dyes can be in two places per experiment), though likely not on the overall trend. In contrast, the intensity differences before and after acceptor photobleaching (Fig. 2F) are much larger, suggesting more robust experimental values. Presumably, these can be converted to FRET efficiencies. In the simplest model, a FRET efficiency of 50% would result in a 50% decrease in donor lifetime, which is clearly not observed. This issue is never addressed or clarified. Notably, there is a wider spread in lifetimes for the condensates (Extended Data Fig. 9 vs Extended Data Fig. 3) – why? The R_e values for the condensates are larger than for the NPC (Fig. 3), which predicts lower FRET, not higher, which would be suggested by the shorter lifetimes. A comparison is made between R_e determined from smFRET and lifetimes (Fig. S1), though it is not clear why ensemble FRET data cannot be used (Fig. 2F data).

Various details of the molecular dynamics simulations need clarification/additional discussion. The models in Fig. 4B seem to include only half of the FG-Nup cloud, which presumably arises because they have included only half of the polypeptide chains (half-toroidal NPC scaffold, p.8). Their models also lack Nup153 and Pom121 (Extended Data Table 2), but no explanation is provided. Why don't

they use a full model? Presumably, the additional polypeptides in the central pore will have an effect, perhaps even blocking the hole that is observed at stronger interaction strengths. It is unclear why a uniform interaction strength is used. Is this reasonable? Doesn't interaction strength depend on amino acid? Don't some FG-polypeptides have stronger interaction strengths than others? It doesn't seem to make sense to use a one amino acid/bead model if all interactions are identical. Am I missing something? Are their NPCs in the permeabilized cells dilated or constricted? Does this make a difference for the modeling (which assumes a constricted state)? Their results are extremely sensitive to interaction strength, with completely different behavior by an interaction strength change of 10%. This does not seem to represent a very stable system, which could, for example, have very different properties with glycosylation, phosphorylation, and nuclear transport factor loading. Thus, it is not convincing at this stage to be making conclusions about opening and closing of the central channel.

Other issues:

The model system is permeabilized cells. This should be more prominently noted in the main text than a single mention on p. 5. This certainly should be noted in Fig. 1.

Fig. 3 – In panel 'A', why do the three lines have a different 'intercept'? I understand that an 'intercept' doesn't make sense on this log-log graph, but I would expect a common model for all three conditions. In this vein, the orange line is largely determined by the point at $N_{res} = 30$; without this point, a reasonable line through the $N_{res} = \sim 15$ 'intercept' could be drawn. This would appear to have a significant effect on the slope, and hence the main conclusion. In 'B', the cartoons do not match the much wider distribution observed in the polypeptide models in Fig. 4B – perhaps these do not represent Nup98, the protein labeled in their studies?

It would be helpful to include some discussion about what constitutes a 'good solvent', particularly as this applies to the polypeptides attached to the NPC. I get that this means that the polypeptide chains are more soluble. Does this mean that they can include some reagent for the 'in solution' conditions to extend the single polymer chains (e.g., salt, sucrose, PEG, cytoplasmic extract)?

Tethering the polypeptide chains to the NPC scaffold forces them together, eliminating the entropic loss needed to do this – why then, are the chains at $\epsilon = 0.44$ in the NPC (Fig. 4B) more extended than when free (Extended Data Fig. 7A)? Shouldn't it be the opposite because there is no entropic loss for bringing the chains together? It is unclear why a highly extended chain in Fig. 4B (most of which look largely 'isolated') is seeing a different environment than an isolated polypeptide in solution (which is much more compact). Are the water molecules implicit or explicit?

Presumably, Eqs. 10 and 11 were combined to estimate R_e from FRET efficiency. It is not clear what was used for $E(R)$, though this can be surmised to be the $1/R^6$ dependence of E . Combining these equations should therefore yield an R_e vs E curve. It would be useful to know where their data points lie on this curve to give an indication of the sensitivity of the measurements. For example, R_e ranges from ~ 25 Å to 170 Å in Fig. 3A. These endpoints seem to be outside the sensitivity range for FRET.

In general, the Methods is substantially under referenced. For example, a single reference on pp. 17-

19, and most of equations 1-11 are not referenced.

What is the angle between the excitation and emission dipoles of the donor dye? The fundamental anisotropy can be significantly lower if this angle is large (i.e., $> 20^\circ$).

Minor issues:

1) p. 3 – “structural biology techniques rely on averaging to enhance signals” is unnecessarily critical. The authors’ approach also relies heavily on averaging.

2) p. 5 – “were unaffected by the cellular environment” would be better as “were minimally unaffected by the cellular environment”. The donor lifetime distributions are not identical.

3) p. 5 – the sentence “If the acceptor is bleached selectively by a high-power laser and if FRET occurs, the donor intensity and the fluorescence lifetime will increase” is confusing.

4) p. 5 – “validated...for measuring FLIM-FRET”. They haven’t reported FLIM up to this point. Validated for FRET measurements is more accurate.’

5) p. 6 – “nuclear transport receptors, exist in large quantities’...Are these large quantities known for their conditions? No references. Would changing the transport receptor loading of the NPC change the solvent quality?

6) p. 17 – Please comment on the stability of the permeabilized cell morphology. 2 h is a long imaging time, and permeabilized cells can show instabilities that could influence the measurements.

7) p. 18 – the GLEBs binding domain was removed from the Nup98 polypeptide – which residues?

8) p. 20 – how are L1 and L2 determined? No reference provided.

9) Fig. 2 – Panel ‘D’ is referenced before panel ‘C’ is discussed. In panels ‘D’ and ‘E’, the specific mutants examined should be noted.

10) Fig. 4 – Panel ‘A’ has linear scales whereas the experimental data in Fig. 3A is log-log. This makes it difficult to compare these figures. Likewise, Extended Data Fig. 7D is linear. It would be helpful if these were shown with identical scaling – perhaps an inset would help. In panel ‘B’, it would be helpful to identify Nup98 (the subject of their study) in a different color.

11) Extended Data Fig. 2 – why isn’t this included in Fig. 1?

12) Extended Data Fig. 5 – panel ‘B’ could replace Fig. 2G.

13) Extended Data Table 2 – terms need to be defined, e.g. dynamic and frozen regions, frozen residue range, type number, etc. It is unclear if anchor domains are embedded in the scaffold structure and they are adding the intrinsically disordered regions, which are not visible in the structure.

14) Equation 14 – Isn't $A(\text{fret}) = 0$ after acceptor photobleaching? Doesn't this mean that the first term doesn't belong in Equation 15? Perhaps the photobleaching is incomplete and a small signal remains. Some comment should be made here.

Author Rebuttals to Initial Comments:

RESPONSE TO REVIEWER COMMENTS

Reviewer comments in black

Our response in blue.

We highlighted the changes in the main text, Extended Data, and SI accordingly in yellow.

Referee #1 (Remarks to the Author):

The manuscript by Yu and colleagues explore the microenvironment within the nuclear pore complex, which is composed of intrinsically disordered protein domain FG-NUPs. They achieved this by first selectively incorporating into these proteins a ncAA at two distinct sites. The side chains of these ncAAs could be subsequently used to stochastically install a FRET pair in live cells. The authors demonstrate that the labeled NPCs retain proper transport function. They also confirm that the resonance energy transfer observed from these reconstituted, labelled NPCs are largely intramolecular and not intermolecular. Finally, the FLIM-FRET measurement of these labeled NPCs, and interfacing these dynamic distance measurements with existing structural information and computational simulation were used to gain insights into the dynamic nature of this microenvironment, which could not be structurally characterized due to its intrinsically disordered nature.

Although I am not an expert in this area, the ability to make dynamic distance measurements of IDPs in unstructured microenvironments within living cells strikes me as an important advance. The use of ncAA technology to facilitate site-selective labelling with minimal perturbation was key to achieving this. Application of this technology for intracellular labelling has been traditionally challenging, because of background noise from concurrent incorporation of the ncAA at endogenous UAG codons and their subsequent labelling. The authors also offer an elegant solution to this limitation by employing their recently developed film-like organelles, that enable mRNA-selective ncAA incorporation.

We thank the reviewer for this enthusiastic summary of our work!

However, there are some concerns that should be addressed before publication:

1. The ncAA incorporation technology is highly context-dependent, which typically results in dramatically altered expression levels of different UAG mutants for the same protein. This issue is likely compounded when suppressing two UAGs in the same mRNA. Such significant differences in expression levels would likely impact subsequent measurements. Yet, there is little data offered here for characterizing these aspects for different mutants. The only data we see is post-labeling images, which lack much of the critical information. What are the relative expression levels for different double mutants? Do the truncated proteins (those failing to incorporate ncAAs) create complications, and how is this issue addressed? Is the fluorophore-labelling efficiency at different sites variable (differences in protein microenvironment) and does that impact measurements?

We thank the reviewer for pointing out the problems related to amber suppression technology. Developing a methodology to address those issues was key to the success of our work. The comment of the reviewer made us realize the need to better structure the main text and supplementary data. In response, we show more data and have revised the text to make it clearer how we addressed the

problem and where the respective solutions are described in detail. In the following, we address the key issues related to amber suppression:

(A) Addressing concerns of potentially different expression due to context dependence etc.

The use of sophisticated biophysical and quantitative FLIM-FRET measurements enabled us to address this concern as it can cope with the weaknesses inherent to GCE. We have tested 19 single and 18 double mutants in the NUP98 FG domain (221-506aa). For all we performed single colour and dual colour labelling under identical conditions. Our quantitative microscopy (where we detect absolute photon counts in every pixel) enabled us to quantify the intensity of all conditions even for very low expression levels, a regime where in our experience immunofluorescence and/or western blot are very unreliable. Importantly, very low expression is required to avoid intermolecular FRET, which is key to the success of the work (see **revised Fig. 2B and C** in the main text, where we saw intermolecular FRET in a highly overexpressing cell, but not in a low expressing cell).

To ensure contrast reliability independent of expression levels, we always selected nuclear envelopes in a selected fluorescence intensity range. Thus, even when the expression levels differed among mutants, we only analysed those cells with a similar expression level. We now show the data in the **revised Supplementary Fig. 1**, for all labelled mutants.

(B) Addressing concerns regarding truncation proteins.

NUP98 has its anchoring domain on the C-terminus (as marked in **revised Fig. 2A** in the main text). Truncated NUP98 proteins lack the C-terminus and thus cannot dock on the nuclear pore complex. Furthermore, only full-length protein that can be labelled with a donor and an acceptor dye contributes to a FRET signal. Therefore, truncations do not create complications for our measurements. This is one of the major reasons why we chose to work with FLIM-FRET and NUP98, which at the same time is also the most physiologically relevant NUP for nuclear transport function, as it was shown that Nup98 is sufficient to make functional NPCs in reconstitution assays¹. Our transport functional assays in Fig. 1B further proved that our GCE method does not interrupt nuclear pore functions.

By contrast, for FG-NUPs with the anchoring domains on the N-terminus (e.g., NUP214) we expect problems, because also truncated proteins can dock on the pore and compete with docking of the labelled protein at the NPC. Nevertheless, we are eager to measure other FG-NUPs in the future when the truncation problem is solved by more advanced genetic code expansion technology. In the revision, we added a discussion about the current challenges and future perspectives on using amber suppression technology to measure the conformations of other FG-NUPs (see at the bottom of **Page 10** in the main text).

(C) Differences in protein microenvironment that could impact measurements.

We chose sophisticated fluorescence anisotropy and lifetime measurements to detect any such effects and to eliminate such concerns. We always recorded fluorescence lifetime for donor-only species as well as their fluorescence anisotropy, which report any change in the quantum yield or rotational mobility of the dye due to microenvironment effects. The data is summarized for each mutant in new **revised Extended Data Fig. 4** and shows high consistence across labelling sites. It is thus safe to say that the differences of fluorophore properties were within the measurement error among all different labelling sites.

We note, that in our experience for dynamic and low complexity IDPs like FG-NUPs, we observe less microenvironment heterogeneity compared to e.g., folded proteins.

2. A key advance here is mRNA-selective ncAA/fluorophore tagging in living cells. While theoretically it makes sense, the authors do not provide any supporting data characterizing to what extent the OTO system reduces off-target labeling and background reduction. Given the potential of this strategy for additional applications, the authors should provide data clarifying this aspect.

We thank the reviewer for the suggestion to further highlight our technological advances. Indeed, without using the OTO systems, the signal-to-off-target-labelling ratio is unworkable, as here we target for low-expression level of NUPs.

We now directly compare the contrast in labelling for the classical (cytoplasmic) and our OTO film-like GCE system. As shown in **revised Extended Data Fig. 1C and D**, we co-transfected the plasmid pcDNA3.1-NUP98^{221TAG}-boxB together with either the OTO-GCE system or the cytoplasmic GCE system. The difference is striking: the nuclear envelopes are much clearer with the OTO-GCE system. By using t-butyloxycarbonyl-lysine (BOC), a control ncAA that cannot be labelled, we proved that the observed off-target-labelling was not due to non-specific sticking of the dye molecules.

3. As a non-expert in this area, it was not fully clear to me what the key take-home messages from the measurements made in the NPC were. If the authors could discuss this aspect in a more succinct way that is accessible to a general audience, and particularly highlight advances in our knowledge relative to the previous state-of-the-art, it would significantly strengthen the paper.

We thank the reviewer for this helpful suggestion. In response, we structured the manuscript better and revised our discussion part substantially (see also below the response to other reviewers) to make it more accessible to a general audience and highlight our main conclusion (see **Page 10** in the main text).

Referee #2 (Remarks to the Author):

This timely contribution comes on the heels of a series of papers that have provided detailed, tomographic maps showing the three dimensional structure and organization of the nuclear pore complex from different cell and species types. All these structures, impressive as they are, feature a prominent hole in the central lining of the pore because these FG-NUPs are conformationally heterogeneous, going by the name of intrinsically disordered regions. In this work, the authors combine a fluorescence lifetime and single molecular Förster resonance energy transfer experiments, focusing on NUP98, to provide a conformational annotation of a key NUP within the family of FG-NUPs. Characterization of the conformational ensemble is enabled by the use of synthetic biology aided selective labeling along the sequence of NUP98. These measurements, which are unprecedented, help lead to a statistical description of the conformational ensembles of NUP98. The key findings are that the ensembles within the pore, in live cells, are distinct from the ensembles sampled in dilute solution, untethered from the pore. Interestingly, the conformational ensembles bear statistical resemblance to those adopted in the context of condensates formed by NUP98. The intriguing inference presented is that the apparent solvent quality of nuclear pore is better than that of a bulk, aqueous solvent, with conformational statistics measured in the pore resembling what one would observe in a good solvent. The congruence between the conformational statistics within the pore and in condensates formed in vitro endorses the Görlich view of stickers-and-spacers architecture and mesh-like properties of the pore. Overall, this is a very timely contribution, of urgent importance, that is very well written. A key feature is the range of technologies that have been brought to bear. Additionally, the number of controls and the photo physical rigor are to be lauded. This is the only group of investigators that could have pulled off this contribution, so the originality and novelty are beyond dispute.

We thank the reviewer for the positive and thorough review of our work and for the insightful comments!

There are specific issues that came up during the reading of the MS that should be addressed in a suitable revision. These are as follows:

1) For the uninitiated, the phasor plots are going to be difficult to follow. The captions for these plots would benefit from being more informative, pedagogical even. Explain what is being plotted, what it means, and why these plots provide unequivocal insights. This is not clear at this juncture. Based on the Fourier transform pair shown in the methods section, one would have thought that the plane on which the analysis would be performed would be the amplitude and phase angle. Simply put, there are two issues with the phasor plots: As presented, they are unclear and the information to be processed by the reader is ambiguous. Second, it seems, based on the fact that these are Fourier transforms of the intensity data, that the analysis affords deeper insights that are not being pursued.

We thank the reviewer for the suggestions to help us improve the phasor plot. In the revision we have followed the advice of the reviewer to make the presentation less ambiguous and more informative.

In brief, in the fluorescence lifetime fields, phasor plots (frequency-domain) and lifetime decays (time-domain) are complementary tools to visualize data, i.e., the two plots contain the same quantitative information. Here we wanted to present the lifetime decays on a single-cell basis, and analysed the lifetime for each mutant. However, a time-domain plot of the lifetime decays of ~2000 cells does not look intuitive as all decay profiles overlay. That is why we decided to convert them into a phasor plot for a better display. In response, we now add more information at the bottom of **Page 6** in the main text and in the **revised Fig. 2F** to guide the readers.

2) The apparent scaling exponent of 0.56 ± 0.05 , with the error bar, has a spread that ranges from that of an apparent theta solvent to that of an apparent good solvent. That is a significant range. The first question is if the errors are intended to be symmetrical? Second, if the conformational statistics are those of self-avoiding walks, i.e., if NUP98 within the pore belongs to the same universality class as self-avoiding walks, then the expectation would be that because of finite size, the inferred exponent should be 0.6 or larger. Instead, given the inferred exponent and the observation of significant compaction in dilute solutions, there is the very real possibility that the conformational ensemble might be a mixture of two populations viz., an ensemble of compact conformations and an ensemble of self-avoiding walks, which when mixed in a suitable proportion could give rise to an apparent exponent of 0.56. This would imply that the pore likely encompasses coexisting conformational ensembles. Can this be ruled out? If not, it would be useful to include this as a formal possibility in the discussions - see Ref. 53. Unfortunately, the simulations will not be able to provide clarity since they are simulations of an effective homopolymer.

We thank the reviewer for the comment on the error in the scaling exponent and for raising the interesting question of heterogeneity. We first discuss the error analysis (point A) and then the heterogeneity comment (point B).

(A) Error in the scaling exponent. Motivated by the reviewer comment, we have revisited the analysis of the error in the scaling exponent to make it less complex, more robust, and better explained. For this, we now measured the control parameter (estimation of background from photobleaching experiments) for all mutants independently, and now provide two forms of error analysis.

Method 1: Global fitting. We employed the bootstrap resampling method for each mutant and added up the lifetime decays of sampled cells ($N = 100$). We then globally fitted the lifetime decays for all mutants to the scaling law. We repeated the resampling and global fitting procedure for 50 times and extracted the scaling exponent as $\nu = 0.56 \pm 0.03$. Here the standard deviation mainly describes the cell-to-cell heterogeneity. The global fitting has a smaller error and is more robust, as it estimates one parameter from all cells (> 2000) across all mutants. See **revised Fig. 3** in the main text, where the orange error band shows the standard deviation estimated from the bootstrapping method.

Method 2: Local fitting of mutant by mutant. We first extracted R_E of each mutant. Such a fit relies on ~ 100 cells per mutant, and thus has inherently a larger error compared to a global fit of ~ 2000 cells. Also, in such a fit no “scaling” model is implied to the data, and it is thus less biased. The result is shown in the blue markers in the **new Extended Data Fig. 7**. Here, the error bars on each blue marker were estimated from bootstrapping, but only for each mutant (for ~ 100 cells). We then fitted the blue markers (mutant-by-mutant data) in the R_E versus N_{res} plot in Extended Data Fig. 7 to obtain the apparent scaling exponent which yielded $\nu = 0.55 \pm 0.05$.

The errors due to photon statistics etc. have recently been discussed in community-wide publications², and those are less than the errors compared to our in-cell measurements with cell-to-cell heterogeneity. Those errors are best estimated in our eyes by measuring many, many cells, and many mutants, and this is what we did. We also show in the **new Supplementary Fig. 3** that the scaling exponent, in contrast to the prefactor, is also more robust to “typical” error sources in FRET measurements, such as uncertainties in the FRET distance R_0 (i.e., the Förster radius).

Apart from the cell-to-cell heterogeneity, we think the error in the scaling exponent is mainly caused by the sequence heterogeneity. The global fit relies on much better statistics and the very low error estimated here might be considered a lower boundary, but probably closer to the reality, than the mutant-by-mutant analysis, which has intrinsically a much smaller data set, and probably depicts an

upper boundary on the error. The two mean agree within 0.01, and this increases confidence on the procedure.

In brief, the errors in both methods are symmetrical and it is beyond our signal-to-noise to detect if they could also be asymmetrical. In response, we now include both fitting methods and the results in the **revised Fig. 3** in the main text, **new Extended Data Fig. 7**, and **Supplementary Text (Page 4)**.

(B) Heterogeneity. The question of heterogeneity in the NUP98 population is intriguing. Driven by the reviewer's question, we have extended our analysis of the MD simulations and performed additional MD simulations specifically to address this question. The 8-fold symmetric human NPC contains $8 \times 6 = 48$ NUP98 chains in six non-symmetry-equivalent positions. In the **new Supplementary Fig. 18**, we now show the grafting positions of the six non-equivalent NUP98 chains in models I and II (we now also include NUP153 and POM121 in model II with the grafting sites described in **new Supplementary Table 3**). NUP98 chains are labelled as "U" chains and the labelling contains "C" if it is on the cytoplasmic side or "N" if on nucleoplasmic side. In the **new Supplementary Fig. 19**, we now compare the extension profiles obtained for these six positions. Across three different models of the NPC (including the FYW stickers-and-spacer-type model with sequence heterogeneity, as described in the following paragraph), we find consistently that the NUP98 chains in the cytoplasmic ring (UC_c position) are somewhat more extended than those in the inner and nuclear ring. Our MD simulations thus indicate some heterogeneity in NUP98 chain extension depending on their precise "grafting" point within the NPC.

In addition, we have extended our MD simulation model and probed for the effects of sequence heterogeneity. Specifically, we have adapted the stickers-and-spacers concept to FG-NUPs by treating aromatic residues (F, Y, W) as mutually attractive stickers separated by weakly interacting spacer regions³. The resulting FYW-stickers model is designed to capture variations in the number and positions of FG repeats and other aromatic rings within and between the different FG-NUPs (see **new Supplementary Fig. 13**). As for the homopolymer model, we carefully calibrated the attractive interaction strength against experimental data. As shown in the **new Supplementary Fig. 16**, for an interaction strength of $\tilde{\epsilon} = 3.25$ the FYW-stickers model nicely reproduces the extension of NUP98 determined in the FRET measurements. We also mapped the phase diagram of the FYW-stickers model and found that the cohesive strength of $\tilde{\epsilon} = 3.25$ is again just below the critical value, as the NUP98 condensate dissolved and uniformly distributed at this value while it forms condensates for higher interaction strengths (see **new Supplementary Fig. 15**). We conclude that a more detailed model accounting for sequence heterogeneity does not alter the conclusions.

In response, we added a discussion on **Pages 8 and 9 in the main text** of the intriguing possibility of coexisting populations of collapsed and extended chains, of the observed variations between non-symmetry-equivalent positions of NUP98 in the NPC, and of the extended simulations with the FYW-stickers heteropolymer model (with new citation added, Wang et al, Cell 2018³). We also included the detailed methods in the **Supplementary Text on Page 9**, and the results in **new Supplementary Figs. 13-16 and Supplementary Figs. 18-19**.

3) The authors propose, based on the calculated phase diagram, that the tethered NUP98 system, modeled as a tethered homopolymer, is akin to this homopolymer at its critical point. First, it is worth noting that the excluded volume limit is in fact a critical point - please see the authoritative book of Schäfer on this topic. This, as shown by other others (<http://dx.doi.org/10.1529/biophysj.106.086264>), enables the analysis of a variety of interesting features, specifically whether or not the tethered chains have a projected end-to-end distance distribution that matches (without any parameterization) the form predicted by des Cloizeaux, Jannink, and Fisher. If so, this will imply a correlation hole and also provide deeper insights into how

cargo negotiate the meshwork enabled by the crosslinking of the meshwork laid out by the self-avoiding chains with stickers. In this context, a relevant paper is one that was published by Wei et al., <https://www.nature.com/nchem/journal/v9/n11/full/nchem.2803.html>.

We thank the reviewer for raising the issue of excluded volume and its impact on the distance distribution of the FG-NUPs grafted to the NPC scaffold. In response, we have examined the end-to-end distance distribution of NUP98 chains in the simulations of models I, II, and FYW (**new Supplementary Fig. 20**). Consistent with the findings by Tran and Pappu⁴, we found that short end-to-end distances are underrepresented as a result at least in part of excluded volume. We also found that the theory of des Cloizeaux et al.⁵ describes the distributions well with some adjustments to the exponents. Most notably, with $p(x) \propto x^b \exp(-cx^d)$, we found that for NUP98 in NPC models I, II, and FYW, the exponent b is close to one and thus indicates an even more pronounced "correlation hole" than in the isolated chains considered in the theory of des Cloizeaux et al.⁵ and in the simulations of Tran and Pappu⁴. The exponent d is in the range of 2.1 to 2.6 close to the value of 2.4 found by Tran and Pappu⁴.

In response, we present these findings in the **new Supplementary Fig. 20** and discuss them at the end of **Page 9** in the **Supplementary Text**.

4) Finally, there are some statements or phrases that are either unclear or unsubstantiated. For example, the authors claim that the motions of the chains are extensive and fast. Neither of these claims can be substantiated by the data - unless they are implicit, in which case they should be made explicit. The authors refer to a "critical polymer mixture" in the discussion section. What do they imply by this phrasing? I thought the last paragraph undermined the substance of the work. There is no doubt that this is technological tour de force. However, focusing the summary claims about novelty on technology / methodology does disservice to the prospect that this work likely provides a clear denouement to a long-standing debate. To this point, the average reader might be inclined to ask about the other NUPs within the NPC. Clearly, tackling all these other NUPs will require further studies and perhaps advances to the technologies used here. It would be useful to mention the other NUPs and know if the authors think that their observations for NUP98 transfer seamlessly over to all other NUPs (my guess is no).

We agree that some of our statements were confusing and we thank the reviewer for pointing those out to us. In response, we have made substantial changes to the discussion on **Page 10-11 in the main text**. As we also explain in our response to reviewer 1, there are also current technological limitations (the truncation problem described above) that currently do not allow to probe other NUPs. We also thank the reviewer for alerting us of the confusion created by the way we had emphasized the methodological advance. In response, we have deleted the respective statement. In this way, we make it clear that the emphasis is on the understanding of the FG-NUPs in the nuclear pore complex. We also rephrased the statement about critical fluctuations and their connection to the formation of a dynamic permeability barrier.

Referee #3 (Remarks to the Author):

While the scaffold of the nuclear pore complex (NPC) has been determined to molecular resolution by a combination of cryo-EM and X-ray studies, the properties of the permeability barrier are less well-determined as it is comprised of hundreds of intrinsically disordered polypeptide chains, which present unique challenges. Molecular dynamics and modeling will certainly be key methods to resolve these dynamic properties, but they must be appropriately parameterized and verified by experimentally determined constraints. This paper reports on an important one of these constraints, i.e., the scaling exponent in Flory homopolymer theory for polypeptides within intact NPCs. The sophisticated state-of-the-art approach is technically complex, yet the beauty of the result is its simplicity. The authors use their recently developed noncanonical amino acid approach to site-specifically incorporate a FRET dye pair into a single FG-polypeptide (Nup98), which is incorporated into functionally competent NPCs. This impressive approach in eukaryotic (human) cells was reported earlier by the corresponding author's group; nonetheless, this paper displays a dramatic illustration of the powerful utility of the method. By varying the amino acid spacing between two dye labels and using FLIM-FRET to obtain average distances between the dye pair, the extension of the polypeptide was determined. This extension depends on the quality of the solvent, i.e., in a 'good' solvent (solubilizes the polypeptide chain), the polymer will be largely extended, whereas in a 'bad' solvent, the polymer will collapse upon itself. The authors determined that the Nup98 chain within NPCs is largely extended, and hence in a good solvent environment. This allows the permeability barrier to remain largely open and fluid, allowing it to both block traffic of diffusing molecules, and yet allow rapid signal-dependent transport.

This paper is well-written, well thought out, and presents a compelling story. The figures are well-made. The in cellulo dye-labeling strategy should be widely applicable to a broad array of cell biological problems. However, they do not perform the dye labeling reaction or their observations in live cells, reducing impact. There are multiples areas in the manuscript where additional clarity is needed.

We thank the reviewer for the thoughtful comments and efforts towards improving our manuscript! In particular we could not have phrased it better than "The sophisticated state-of-the-art approach is technically complex, yet the beauty of the result is its simplicity".

With respect to the live-cell labelling comment we provide an answer here. An additional extended answer can be found at the end of this point-by-point review (Appendix I) and its purpose is to show that all our claims and explanations are substantiated by our own measurements.

Only few dye pairs are suitable for high resolution FRET measurements, as a number of criteria need to be fulfilled, such as high photostability, a suitable Foerster distance, distinction from cellular autofluorescence, and an ideally monoexponential lifetime decay of the donor.

We now tested a large number of dyes for live-cell labelling experiments. We identified one workable condition where we were able to introduce a suitable FRET pair in live cells. For this system, we now include our core results in the main text (see **revised Fig. 2**) and extended figures (**new Extended Data Fig. 2 and Fig. 6**). The live-cell results are in good qualitative agreement with our assays in semi-permeabilized cells with validated transport functionality. However, in live cells we could not achieve the same signal-to-noise ratio needed for quantitatively extracting the results with residue resolution that is complementary to MD simulations. The major reason is simple: for live-cell labelling, the dyes need to be membrane permeable, and this intrinsic demand comes with an increased inherent tendency to stick. The list of possible dyes shrinks even further when considering that we need two dyes that form a perfect FRET dye pair suitable for quantitative measurements in the cell. For the best

live-cell dye pair (e.g., JF549 and JF646), we could only identify nuclear envelopes with sufficient contrast in highly overexpressing cells. In those cells, though, we detected intermolecular FRET that originates between different labelled NUP98s inside the same nanosized NPC (new main text Fig. 2C). Such a problem is to be expected in the NPC, as the NPC has octahedral symmetry with ~48 copies of NUP98 squeezed into a nanosized container. In contrast to permeabilized cell labelling, a nuclear envelope signal cannot be discriminated from background labelling for low expression levels in live cells.

We also like to stress again that the NPC transport assay in semi-permeabilized cells has established itself over the decades as the gold standard for transport assays in intact NPCs⁶⁻⁹, and we again validated that in our measurements the nuclear transport machinery is still intact *in situ*. Thus, we can provide a structural model for functional NPC machinery.

The reality is that the ideal live-cell dye compatible with genetic code expansion technology does not yet exist, let alone a dye pair suitable for FRET-based distance measurements. This is one of the great challenges in chemical biology for the long future.

In the extended answer, we go further into details and explain why a synthetic dye labelling strategy that offers the benefit of using 100-fold smaller probes than GFP or protein self-labelling tags, but with the same contrast as those larger tags, is currently beyond the state-of-the-art. In particular, we show that all available membrane-permeable dyes we could access have this problem. While many of those are workable under easier “conditions” like abundant cytoskeletal proteins or highly overexpressed conditions¹⁰⁻¹³, they fail in the regime probed in our challenging experiments). We also explain in Appendix I below why using SNAP-tag and HaloTag suffers less from this problem, because in those cases the fluorogenicity of some dyes could be enhanced by protein engineering. Our study, however, needs residue specificity of small probes to measure a scaling law, yet SNAP-tag and HaloTag are ~100-times bigger than a single amino acid.

At last, quantitative lifetime measurements are not possible with just any dye, as it has high requirements on mono-exponentiality of the decay, rotational mobility, and dye stability.

Nevertheless, our new results shows that it is possible to perform qualitative lifetime measurements in live cells. We are grateful to the reviewer for having challenged us to perform these experiments. With this proof-of-principle, we show that our methods will have broad applicability.

It is unclear why the FLIM signal is the parameter of choice for obtaining distance information. The difference in lifetimes for the various FRET pairs are tiny (Extended Data Fig. 5a), and these lifetimes are never explicitly reported. Because these differences are so small, the differences in donor only lifetimes (Extended Data Fig. 3) can have a significant impact on individual measurements (noting that the donor dyes can be in two places per experiment), though likely not on the overall trend. In contrast, the intensity differences before and after acceptor photobleaching (Fig. 2F) are much larger, suggesting more robust experimental values. Presumably, these can be converted to FRET efficiencies. In the simplest model, a FRET efficiency of 50% would result in a 50% decrease in donor lifetime, which is clearly not observed. This issue is never addressed or clarified. Notably, there is a wider spread in lifetimes for the condensates (Extended Data Fig. 9 vs Extended Data Fig. 3) – why? The R_e values for the condensates are larger than for the NPC (Fig. 3), which predicts lower FRET, not higher, which would be suggested by the shorter lifetimes. A comparison is made between R_e determined from smFRET and lifetimes (Fig. S1), though it is not clear why ensemble FRET data cannot be used (Fig. 2F data).

We thank the reviewer for raising the question about FLIM techniques. We apologize for not explaining this well enough. We reply first to point (A) about intensity-based vs lifetime-based FRET, and then to point (B) about the explicit lifetime, and the ‘smaller’ lifetime difference in the cell and ‘bigger’ lifetime difference in the condensates.

(A) Intensity-based FRET and lifetime-based FRET

There are mainly two FRET methods, intensity-based and fluorescence-lifetime/frequency-domain FRET methods, to extract a distance.

Intensity-based measurements are possible only if the exact stoichiometry of donor and acceptor labels is known. Even in vitro (biochemically), it is frequently hard to know the precise stoichiometry, because dyes can e.g., transit into dark states or bleach. Due to the complex cellular environment, dye sticking, and the off-target labelling issue in amber suppression (see point 2 on page 3 for reviewer 1), it is practically impossible to establish quantitative in-cell labelling and stoichiometry.

Though we used site-specific but random labelling by applying a dye mixture of known ratio (donor:acceptor=1:2), the real stoichiometry in the cell cannot be assumed to be the same. In the absence of a priori knowledge of the true D:A ratio, acceptor photobleaching cannot be used to detect anything else but an apparent FRET or proximity value. However, a conversion to distances needs knowledge about the exact stoichiometry of photophysically active acceptor and active donor during the measurement.

We also note that currently OTO amber suppression technology does not offer to bring two orthogonal ultrafast Diels-Alder reactions into the same protein. But even if this were possible, one would need a method or measurement configuration that is tolerant to “photophysical inactive” dyes if a quantitative measurement is required.

The unknown stoichiometry issue can be solved primarily by two established technologies:

- i) In single-molecular FRET (smFRET) measurements, by operating at the single-molecule level, the FRET-labelled species can be clearly distinguishable from donor-only or acceptor-only species. Photophysical inactive dyes give a distinct signal, and can thus be removed from data analysis¹⁴. However, as NUPs are highly packed in the sub-resolution NPCs, intensity-based FRET at the single-molecule level in functional NPCs is not yet feasible.
- ii) In contrast to intensity-based measurements, lifetime/frequency-domain measurements in general make it possible to determine the FRET efficiency as well as the fractions of FRET population, donor-only population, and the cellular background by fitting the exponential decay. This is the method we chose.

Finally, we used single-molecule FRET experiments of purified Nup98 to perform both, lifetime and intensity-based FRET analysis from the same measurement. In particular, the intensity-based measurements at single-molecule level are considered the gold standard to extract distance distributions from proteins including intrinsically disordered proteins^{15,16}. As shown in Supplementary Fig. 4, we got consistent results of R_E using the intensity-based and lifetime-based analysis, validating our FLIM pipeline.

In response, we have revised **the main text** in the middle of **Page 4**.

(B) The explicit lifetime and the ‘smaller’ lifetime difference in the cell and ‘bigger’ lifetime difference in the condensates.

The measured lifetimes are actually multi-exponential decays. Though the average photon arrival time (i.e., average lifetime) could be used to qualitatively compare the lifetime changes before/after acceptor photobleaching (e.g., the average photon arrival time changed from ~3.5 ns to ~3.8 ns in new Fig. 2D), this single number could not be directly converted to a distance. Because the average lifetime is dominated by the donor-only population. In order to quantitatively extract the end-to-end distance, one has to use the feature of the whole lifetime decay curve described by multi-components (donor-only, FRET population, and background). To enhance the accuracy, we combined acceptor photobleaching with lifetime measurements and ultimately probed ~2000 cells, which is a major achievement of the work. The data across 18 mutants as illustrated e.g. in the phasor plot in main text Fig. 2 that follows a clear trend shows how well this worked. And as also explained in more details to reviewer 2 regarding our error estimate (see Page 5, point A), we have improved these measurements and analysis further.

The difference in lifetimes for various FRET pairs in the cell looks small, because of the existence of the donor-only population, which can partially bury the FRET signals in the displayed lifetime curve, but the signal change is extractable with high confidence as shown in Fig 2. E and F. In the new **revised Supplementary Text on Page 6-7**, we discuss the effect of donor-only species on the fluorescence lifetime decays. We specifically use a toy model to go through the specific 50% FRET efficiency case mentioned by the reviewer.

Corresponding to new Supplementary Figs. 6 and 7, if we take for example an $R_E=90 \text{ \AA}$, it roughly corresponds to 50% in FRET efficiency (see Gaussian chain model in Supplementary Fig. 5A). Given $\tau_{\text{Donly}} = 3.9 \text{ ns}$ (average photon arrival time is 3.65 ns), the calculated average fluorescence lifetime (average photon arrival time) for $R_E=90 \text{ \AA}$ is listed in the table below. Therefore, for the complex in-cell system with donor-only species and cellular background, 50% FRET efficiency cannot be directly converted to 50% decrease in donor lifetime, and the observed decrease in donor lifetime would be much smaller than 50%.

Fraction of donor-only	Average lifetime ($R_E=60 \text{ \AA}$) (ns)	Change compared to the average lifetime of donor-only
0%	2.74	24.9%
10%	2.90	20.5%
30%	3.16	13.6%
50%	3.34	8.5%

Compared to the in-cell measurements, *in vitro* reconstituted condensates do not have non-specific sticking of dyes nor cellular autofluorescence. Therefore, the fraction of donor-only population is reduced. One can compare the cases with 30% and 50% donor-only population in Supplementary Fig S6 B vs C. That’s why there is a wider spread in lifetimes for the *in vitro* condensates compared to in-cell measurements.

In response, we have revised the **Supplementary Text on Page 6-7**, and **Supplementary Fig. 6 and 7**.

Various details of the molecular dynamics simulations need clarification/additional discussion. The models in Fig. 4B seem to include only half of the FG-Nup cloud, which presumably arises because they have included only half of the polypeptide chains (half-toroidal NPC scaffold, p.8). Their models also lack Nup153 and Pom121 (Extended Data Table 2), but no explanation is provided. Why don’t

they use a full model? Presumably, the additional polypeptides in the central pore will have an effect, perhaps even blocking the hole that is observed at stronger interaction strengths. It is unclear why a uniform interaction strength is used. Is this reasonable? Doesn't interaction strength depend on amino acid? Don't some FG-polypeptides have stronger interaction strengths than others? It doesn't seem to make sense to use a one amino acid/bead model if all interactions are identical. Am I missing something? Are their NPCs in the permeabilized cells dilated or constricted? Does this make a difference for the modeling (which assumes a constricted state)? Their results are extremely sensitive to interaction strength, with completely different behavior by an interaction strength change of 10%. This does not seem to represent a very stable system, which could, for example, have very different properties with glycosylation, phosphorylation, and nuclear transport factor loading. Thus, it is not convincing at this stage to be making conclusions about opening and closing of the central channel.

In response, we now explain that our original MD simulation model did not include all FG-NUPS because their anchoring points had not been resolved with high confidence. However, with this caveat but incorporating what we do know about the respective anchor locations, we built an extended NPC simulation model that now includes also NUP153 and POM121. As a result, the total mass of the FG-NUPS increased by approximately 17 %. The results of MD simulations with this extended model II are shown in **revised Fig. 4** in the main text. We found that the addition of the FG-NUPS did not alter the conclusions. In particular, we again found that an attractive interaction strength of $\tilde{\epsilon} = 0.44$ just below the critical value excellently reproduced the experimental measurements.

We fully agree with the reviewer that the strength of the interactions has to be carefully tuned because factors such as glycosylation, phosphorylation, and nuclear transport factor loading are not resolved at present at the required level of detail. In our modelling we consider the interaction strength $\tilde{\epsilon}$ as "effective" in an effort to implicitly account for the presence of other proteins and cargo, and of posttranslational modifications of the FG-NUPS, as these are factors that are not controlled in our *in situ* measurements. Therefore, we carefully tuned the interaction strength epsilon of the simulation model to match the *in situ* experimental distance measurements. We now discuss this relevant point on **Page 8 in the main text**.

In response to the question why a uniform interaction strength was used, we added the FYW stickers model designed to capture variations in the number and positions of FG repeats and other aromatic rings within and between the different FG-NUPS (see **new Supplementary Fig. 13**). As for the homopolymer model, we carefully calibrated the attractive interaction strength against experimental data. As shown in the **new Supplementary Fig. 16**, for an interaction strength of $\tilde{\epsilon} = 3.25$ the FYW stickers model nicely reproduces the extension of NUP98 determined in the FRET measurements, which is again just below the critical value for NUP98-condensate formation (see **new Supplementary Fig. 15**). We conclude that a more detailed model accounting for sequence heterogeneity does not alter the conclusions.

In response to the question of sensitivity to phosphorylation or glycosylation, "with completely different behavior by an interaction strength change of 10%", we agree that this is remarkable. However, this kind of sensitivity is a consequence of being close to the phase boundary for NUP condensate formation: the phase separated condensate differs sharply from the dilute solution, yet only a small change in the solvent and thermodynamic conditions decide on whether condensates form or not. We now address this point explicitly (see **Supplementary Figs. 13, 15 and 16**), as explained at the bottom of **Page 8 in the main text**.

Other issues:

The model system is permeabilized cells. This should be more prominently noted in the main text than a single mention on p. 5. This certainly should be noted in Fig. 1.

We thank the reviewer for the comment. We now make it clearer in both the main text and Fig. 1. And we now show that qualitatively our approach is also live-cell compatible (see our reply above on Page 9-10) and discuss live-cell measurements in greater detail in the main text on Page 4 and 6, and Fig. 2, as well as in **Extended Data Fig. 2 and 6**.

Fig. 3 – In panel ‘A’, why do the three lines have a different ‘intercept’? I understand that an ‘intercept’ doesn’t make sense on this log-log graph, but I would expect a common model for all three conditions. In this vein, the orange line is largely determined by the point at $N_{res} = 30$; without this point, a reasonable line through the $N_{res} = \sim 15$ ‘intercept’ could be drawn. This would appear to have a significant effect on the slope, and hence the main conclusion. In ‘B’, the cartoons do not match the much wider distribution observed in the polypeptide models in Fig. 4B – perhaps these do not represent Nup98, the protein labeled in their studies?

For an ideal homopolymer with infinite long chain length, we expect a scaling relation $R_E = \rho N^\nu$. In a log-log plot, ρ determines the intercept. However, the range of finite segment lengths that we probe here is not broad enough to determine ρ independently. For a heteropolymer in different solvents with different expansions, ρ is not expected to be constant. Obtaining good heteropolymer models to account for those effects is one of the challenges to be solved in biopolymer science field.

Thus, we only fitted the slope in the log-log graph to extract the apparent scaling, and did not show the prefactor to avoid confusion. Since we measured many mutants, any single point does not substantially impact the trend. For example, as suggested by the reviewer, removing the point at $N_{res} = 30$ gives a scaling exponent $\nu = 0.55$, very close to $\nu = 0.56$ with that point included. So, one point does not have significant impact on the scaling exponent, and the main conclusion remains that we detect a good solvent condition in the NPC. As detailed above in the response to reviewer 2 (Page 5, point A) and in the revised supplementary text page 4 and 5, we have also greatly expanded our error analysis using new measurements. Our bootstrap errors for the global fit and mutant-by-mutant analysis give a good estimate on the precision and confidence of the measurement.

We apologize for the confusion in Fig. 4B. The red polymers referred to all the FG-NUPs that have been simulated. We now changed the colour code and show our target NUP98 in yellow and revise the caption accordingly for **new Fig. 4B**. We also added the **new Supplementary Fig. 17**, in which we show the 48 NUP98 chains alone, with the other chains removed, for improved visual clarity.

It would be helpful to include some discussion about what constitutes a ‘good solvent’, particularly as this applies to the polypeptides attached to the NPC. I get that this means that the polypeptide chains are more soluble. Does this mean that they can include some reagent for the ‘in solution’ conditions to extend the single polymer chains (e.g., salt, sucrose, PEG, cytoplasmic extract)?

We thank the reviewer for alerting us that we had not made it clear that we use the boundary of a scaling exponent $\nu = 0.5$ to distinguish between good and poor solvents, as in the polymer literature (see, e.g., P. G. de Gennes, *Scaling concept in polymer physics*, Chapter 3, page 69-73). In Flory’s homopolymer theory¹² at $\nu = 0.5$ the polymer self-interactions and the interactions with solvent are balanced; at $\nu \sim 0.6$, the interactions between the polymer segments and the solvent are maximized. This definition of solvent quality also follows established practice for disordered proteins (see, e.g., Hofmann et al.¹⁵), where disordered protein chains expand in good solvents ($\nu > 0.5$) to increase their interactions with the solvent and compactify in poor solvents to minimize these interactions. We now

address this issue by pointing to the relevant literature and by clarifying this in the first paragraph on **Page 7 in the main text**.

In response to the question concerning possible “reagents”, in our previous work of smFRET measurements of single FG-NUP in solution, we didn’t observe conformational change in the polymer chain by adding import receptors, e.g., Importin β and NTF2 (1-10 μ M)^{17,18}. We think that in order to create a good solvent condition, the polymeric solvents need to reach a substantially high concentration, e.g., as high as the FG-NUPs (\sim mM level) and nuclear transport receptors inside the NPC. At the bottom of **Page 7 in the main text**, we rephrased the statement on how mutual attractive interactions between the FG-NUPs in the pore establish conditions of a good solvent with expanded chains, akin to the chain conformations in polymer melts¹⁹.

Tethering the polypeptide chains to the NPC scaffold forces them together, eliminating the entropic loss needed to do this – why then, are the chains at $\epsilon = 0.44$ in the NPC (Fig. 4B) more extended than when free (Extended Data Fig. 7A)? Shouldn’t it be the opposite because there is no entropic loss for bringing the chains together? It is unclear why a highly extended chain in Fig. 4B (most of which look largely ‘isolated’) is seeing a different environment than an isolated polypeptide in solution (which is much more compact). Are the water molecules implicit or explicit?

We thank the reviewer for raising these interesting questions. In response, we now explicitly address the question of chain collapse and extension in the different environments. Specifically to the point raised, at an interaction strength of $\tilde{\epsilon} = 0.44$ above the critical interaction threshold, isolated chains in dilute solution collapse driven by intrachain interactions. In our model, this results in a loss of configurational entropy that is compensated by a gain in energy. By contrast, at the same interaction strength in a bulk condensate, we have strong interchain interactions. As a consequence, the chains can maintain a high level of disorder (and thus high entropy) despite these strong interactions, and thus without incurring a major energetic penalty. Now in the NPC, the situation is closer to the bulk condensate with strong interchain interactions. In this regard, the picture in Fig. 4B may have been somewhat misleading because the comparably loose FG-NUPs at the outer rings dominate the view. As we now show in the **revised Fig. 4B**, the NUP98 chains are deeply immersed into a protein-dense environment and are thus nearly invisible. This becomes even clearer when comparing Fig. 4B with the **new Supplementary Fig. 17**, in which NUP98 is the only FG-NUP shown and all others are removed.

Further in regards to the entropy of condensation, Ng and Görlich²⁰ recently showed that NUP98 has a lower critical solution temperature, i.e., the condensate is entropically stabilized. This entropic stabilization points to a strong role of hydrophobic interactions, which strengthen with temperature. In the newly added FYW model (**new Supplementary Figs. 13-16**), we explicitly account for the hydrophobic interactions between aromatic residues.

In response to the question of solvent, we now make it clearer that in models I, II, and FYW the solvent is implicitly included, i.e., by tuning ϵ . However, to address this interesting question we have now built also a model in which solvent is included explicitly. As shown in **new Supplementary Fig. 11**, including the solvent shifts the phase boundary in phase diagram, but the main conclusions remain unchanged. In particular, an interaction just below the critical value best reproduces the observed polymer characteristics. Also in terms of the distance scaling of polymers in isolation and in the NPC, the inclusion of explicit solvent did not alter the conclusions (**new Supplementary Fig. 12**).

Presumably, Eqs. 10 and 11 were combined to estimate R_e from FRET efficiency. It is not clear what was used for $E(R)$, though this can be surmised to be the $1/R^6$ dependence of E . Combining these equations should therefore yield an R_e vs E curve. It would be useful to know where their data points

lie on this curve to give an indication of the sensitivity of the measurements. For example, R_E ranges from ~25 Å to 170 Å in Fig. 3A. These endpoints seem to be outside the sensitivity range for FRET.

$E=1/(1+(R_E/R_0)^6)$ is commonly used to describe a static polymer model (e.g., dsDNA) with a fixed end-to-end distance. However, for an intrinsically disordered protein that populates an ensemble of rapidly interconverting conformations, we cannot use the above equation (see e.g. point-by-point ref 15). By combining Eqs. 10 and 11 (now as Eqs. 12 and 13), we use the root mean squared end-to-end distance $\sqrt{\langle R_E^2 \rangle}$, simply referred as R_E , derived from a Gaussian chain model to describe such a conformational ensemble. Note that the R_E derived from Gaussian chain model cannot be directly compared to the R_E in a static model, where the sensitivity range is usually reported as from ~20 Å to ~100 Å. For example, in **new Supplementary Fig. 5**, we plot E vs R_E for $R_0 = 60$ Å (the Förster distance for a commonly used FRET dye pair of Alexa 488 and Alexa 594), and $R_0 = 77$ Å (the Förster distance of the FRET dye pair of AZDye 594 and LD 655 optimized for this work) in a static model (i.e., $E=1/(1+(R/R_0)^6)$) or in a Gaussian chain model. For $R_0 = 77$ Å, the Gaussian chain model show a broader distribution of R_E against E , compared to a static model (**Supplementary Fig. 5A**). One can estimate the sensitivity range by plotting the 1st derivative of E vs R_E , where the Gaussian chain model shows broader sensitivity range than a static model (**Supplementary Fig. 5B**).

We now include the above discussion in **Supplementary Text Page 6 (Comparison between a Gaussian chain model with a static model)** and also **new Supplementary Fig. 5**. We thank the reviewer for drawing our attention to our previous poor description.

In general, the Methods is substantially under referenced. For example, a single reference on pp. 17-19, and most of equations 1-11 are not referenced.

We thank the reviewer for pointing that out. We now add the references accordingly.

What is the angle between the excitation and emission dipoles of the donor dye? The fundamental anisotropy can be significantly lower if this angle is large (i.e., > 20°).

The fundamental anisotropy for labelled NUP98 in the NPC, in reconstituted condensates, and in solution on single-molecule level that we measured are 0.27 ± 0.015 , 0.28 ± 0.003 and 0.15 ± 0.003 , respectively. According to the following equation²¹,

$$r_0 = \frac{2}{5} \left(\frac{3\cos^2\beta - 1}{2} \right)$$

the angle between the excitation and emission dipoles of the donor dye are 27.7° (in the NPC), 26.6° (in reconstituted condensates) and 40.2° (in solution on single-molecule level). In response, we now add the data to new Extended Data Fig. 4 and main text methods page 24.

The fundamental anisotropies were found to be smaller than 0.3 in all cases, and the error in the distance measurements was proved to be below 10%²². We have also performed global fitting of the scaling law by increasing or decreasing R_0 by 10% (see **new Supplementary Fig. 3**). In both cases, the scaling exponent didn't change while only the prefactor changed. Therefore, we are confident that these effects do not substantially impact our accuracy nor alter our main conclusion of good solvent condition inside the NPC.

Minor issues:

1) p. 3 – “structural biology techniques rely on averaging to enhance signals” is unnecessarily critical. The authors' approach also relies heavily on averaging.

We agreed with the reviewer and revised accordingly. Thank you for pointing this out.

2) p. 5 – “were unaffected by the cellular environment” would be better as “were minimally unaffected by the cellular environment”. The donor lifetime distributions are not identical.

We agreed with the reviewer and revised accordingly.

3) p. 5 – the sentence “If the acceptor is bleached selectively by a high-power laser and if FRET occurs, the donor intensity and the fluorescence lifetime will increase” is confusing.

We thank the reviewer for pointing this out and revised the sentence.

4) p. 5 – “validated...for measuring FLIM-FRET”. They haven’t reported FLIM up to this point. Validated for FRET measurements is more accurate.’

We agreed with the reviewer and revised accordingly.

5) p. 6 – “nuclear transport receptors, exist in large quantities’...Are these large quantities known for their conditions? No references. Would changing the transport receptor loading of the NPC change the solvent quality?

Here the reviewer raises another interesting question and we apologize for being too brief on this and missing to cite references. In case of the *in situ* experiments, we consider it safe to assume that the NPCs contain transport receptors and cargo as shown by work in reference^{9,23}. In addition, we also performed immunostaining on the nuclear transport receptor importin- β and showed that endogenous importin- β colocalized at the nuclear envelope after semi-permeabilization and labelling, and was retained for longer than 2 hrs (see **new Extended Data Fig. 3E and F**). In the modelling, we consider the interaction strength epsilon as “effective” in an effort to implicitly account for the presence of other proteins and of posttranslational modifications of the FG-NUPS. Therefore, we tune epsilon to match the experimental distance measurements. We added references^{17,23} and revised our statement at the bottom of **Page 7 in the main text**. We also revised **Extended Data Fig. 3** and added the **Methods on Immunostaining of endogenous importin- β on Page 22**.

6) p. 17 – Please comment on the stability of the permeabilized cell morphology. 2 h is a long imaging time, and permeabilized cells can show instabilities that could influence the measurements.

As shown in **new Extended Data Fig.3C and D**, the semi-permeabilized and labelled COS-7 cells maintained their transport functions in passive exclusion assay and active transport assay after 2 hrs in line with the literature (e.g. point-by-point ref. 7). That the functionality of the transport activity is not changed over time, was a core selection criterium to restrict the measurements to the two-hour window.

7) p. 18 – the GLEBs binding domain was removed from the Nup98 polypeptide – which residues?

GLEBs binding domain refers to 157-213 aa. We added the residues in the **Methods Page 22**.

8) p. 20 – how are L1 and L2 determined? No reference provided.

We thank the reviewer for noticing that we missed to explain this properly before. We determined L_1 and L_2 by fitting the measured fluorescence decays of yellow fluorescent protein (YFP) with known

fluorescence rotational relaxation time. We have added the details and references to this common procedure^{24,25} in the **Methods Page 25**.

9) Fig. 2 – Panel ‘D’ is referenced before panel ‘C’ is discussed. In panels ‘D’ and ‘E’, the specific mutants examined should be noted.

We revised Fig.2 as suggested, thank you!

10) Fig. 4 – Panel ‘A’ has linear scales whereas the experimental data in Fig. 3A is log-log. This makes it difficult to compare these figures. Likewise, Extended Data Fig. 7D is linear. It would be helpful if these were shown with identical scaling – perhaps an inset would help. In panel ‘B’, it would be helpful to identify Nup98 (the subject of their study) in a different color.

We thank the reviewer for the suggestions. We have now included a log-log plot as an inset to **revised Fig. 4A** in the main text, as suggested. In addition, we now colour the NUP98 chains yellow in **revised Fig. 4B**. Since the NUP98 chains are deeply immersed into the FG-NUP network and thus barely visible in Fig. 4B, we added a **new SI Fig. 17** in which all other FG-NUPs are removed to reveal the NUP98 chains.

11) Extended Data Fig. 2 – why isn’t this included in Fig. 1?

We thank the reviewer for the suggestion. However, as suggested above by the reviewer, we now added more control experiments to further prove that the nuclear pores in the permeabilized and labelled cells were still functional and the nuclear transport receptors (e.g., importin- β) were retained at the nuclear envelop after 2 hrs. Therefore, we think it is better to keep all those control experiments in Extended Data Fig. 2 (Please see point 5) and 6) above). In our revision, this figure is **new Extended Data Fig. 3**.

12) Extended Data Fig. 5 – panel ‘B’ could replace Fig. 2G.

We thank the reviewer for the suggestion. However, we think it is easier to compare the difference in lifetimes across all the mutants when we zoom in for the FRET region on the phasor plot. Therefore, we prefer to keep the overview plot of both FRET and BOC in the Extended Data.

13) Extended Data Table 2 – terms need to be defined, e.g. dynamic and frozen regions, frozen residue range, type number, etc. It is unclear if anchor domains are embedded in the scaffold structure and they are adding the intrinsically disordered regions, which are not visible in the structure.

We thank the reviewer for alerting us of these issues. In our revision, we move the table to Supplementary Table 2. In response, we now clarify these terms in the caption of **Supplementary Table 2 on Page 33**. We also point the reviewer to the supplementary data file NPC_Model_I_homopolymer.lampstrj, in which the structural data of the simulation model are compiled.

14) Equation 14 – Isn’t $A(\text{fret}) = 0$ after acceptor photobleaching? Doesn’t this mean that the first term doesn’t belong in Equation 15? Perhaps the photobleaching is incomplete and a small signal remains. Some comment should be made here.

We apologize for the confusion about the subscripts here. In our revision, Equation 14 is now 16. We define A_D , A_{FRET} , and A_{bg} as the species fraction of the three components before photobleaching. Here we performed a complete photobleaching, which was verified by the observation that no acceptor

signal could be detected by 660 nm laser excitation. After a complete acceptor photobleaching, the FRET population (A_{FRET}) shows the same lifetime as the donor only, i.e., from $A_{\text{FRET}} \int_0^\infty \rho(R) e^{-\frac{t}{\tau_D} \left[1 + \left(\frac{R_0}{R}\right)^6\right]} dR$ to $A_{\text{FRET}} e^{-\frac{t}{\tau_D}}$. We want to separate this newly generated donor-only population after photobleaching (denoted by A_{FRET}) from the original population of donor-only (denoted by A_D) in the equation. In response, we clarified the definition.

We thus revised our statement **on Page 2 in the Supplementary Text** to “where A_D , A_{FRET} , and A_{bg} are the species fraction, i.e., the initial intensities (at $t = 0$) for the three components before acceptor photobleaching, respectively”.

Appendix I

Extended answer to comment on live-cell labelling

1. The useability of genetic code expansion and click-labelling for live-cell labelling and how this compares to self-labelling-protein techniques

The genetic code expansion system we used in this work is compatible with live-cell labelling. We have tested various live-cell dyes, including Janelia Fluor 549-tetrazine (JF549-tz, Tocris), Janelia Fluor 646-tetrazine (JF 646-tz, Tocris), silicon rhodamine-tetrazine (SiR-tz, Spirochrome), and TAMRA-tetrazine (click chemistry tools). These rhodamine-based dyes have been demonstrated with high fluorogenicity for live-cell labelling when linked to the ligands of self-labelling protein (SLP) tags, e.g., HaloTag, SNAP-tag, and CLIP-tag in the literature²⁶. Owing to the sheer size of the SLP tags and the inherently limited freedom of labelling, we could not use them to probe multiple distance distributions of the same protein. Therefore, we chose smaller-sized, H-tetrazine derivatives of live-cell dyes coupled with genetic code expansion (GCE) systems to label noncanonical amino acids (nCAAs) that are site-specifically incorporated in the target protein. Compared to SLP derivatives of rhodamine-based dyes, however, the fluorogenicity of the tetrazine derivatives highly reduces, with the measured turn-on ratios of 1-4 in PBS solution *in vitro*¹⁰. The fluorogenicity can be even lower for *in-cell* labelling due to the complex cellular environments.

The fluorogenicity of new (silicon) rhodamine dyes is based on an environmentally sensitive equilibrium between a closed, nonfluorescent spirocyclic and an open, fluorescent quinoid form. For SLP tags, the specific interactions between their surface residues of the ligand and the rhodamine's xanthen ring are key to the high fluorogenicity (see structures adapted from point-by-point ref 27 in Extended Revision Response Fig. 1 below), where the specific interaction switches the equilibrium of rhodamines toward the fluorescent form. However, the tetrazine derivatives of the rhodamine dyes do not have such surface residues to interact with, which can lower the imaging contrast.

Extended Revision Response Fig. 1 EM structure showing that the specific interactions between the surface residues of (A) Halotag7 and (B) SNAP-tag ligands and the rhodamine's xanthen ring. The specific interaction switches the equilibrium of rhodamines toward the fluorescent form, which are critical to their high fluorogenicity. Adapted from²⁷.

To test various live-cell dyes in GCE system, we first transiently expressed mCerulean3^{116TAG}-vimentin (Extended Revision Response Fig. 2) as the reporter protein in the presence of trans-cyclooct-2-en-l-lysine (TCO*A, a noncanonical amino acid (nCAA) that can be labelled) or t-butyloxycarbonyl-l-lysine (BOC, a control nCAA that cannot be labelled). Noted, that the truncated proteins (those failing to incorporate nCAAs), i.e., mCerulean3(1-115 aa), do not have fluorescent signal and cannot dock on vimentin. For live-cell labelling with cell-permeable dyes, at 20-24 hrs post-transfection, COS-7 cells were washed with fresh normal culture medium supplemented with 10 mM HEPES and 50 μ M BOC and incubated at 37 °C for 2 hrs to get rid of the residual TCO*A. Then the cells were incubated with 250 nM cell-permeable dye in the serum-free culture medium at 37 °C for 45 min. After the labelling, the cells were washed with fresh normal culture medium 4 times in 2 hrs. Cells were imaged

immediately at room temperature in the culture medium supplemented with 10 mM HEPES without phenol red. The lid of the imaging dish was kept closed during cell imaging.

For comparison, we also labelled the cells with cell-impermeable dyes by semi-permeabilizing the plasma membrane with low-dosage digitonin (same as what we did in the manuscript). For the TCO*A groups in Extended Revision Response Fig. 2, the fluorescent signals of various dyes colocalized with the vimentin structure in the mCerulean channel, proving that the tetrazine-functional dyes could react with TCO*A. The BOC groups showed the background labelling due to non-specific sticking of the dye, where the cell-permeable dyes had higher background labelling than cell-impermeable dyes.

We then expressed and labelled NUP98 structure with a single amber site (NUP98^{221TAG}) with various dyes using the labelling conditions described above in COS-7 cells. Because NUPs are much less abundant than cytoskeletal “model” systems, the fluorescent signal of NUPs (Extended Data Fig. 2) is dimmer compared to the vimentin in Extended Revision Response Fig. 2 (the intensity display range was adjusted to 0-1000 for all dye channels (0-2000 for vimentin)). For live-cell dyes, we could hardly identify a clear nuclear envelope from the background signals (which can be observed in BOC group and without ncAA groups) unless we highly overexpressed NUP98.

Summary: For all tested membrane-permeable dyes, we find workable conditions for “simpler” highly abundant cytoskeletal proteins or highly overexpressed NUPs, in line with the published works for live-cell labelling using conventional GCE/amber suppression technology^{11,13,28}. However, the same conditions are by far not good enough for low protein expression levels of NUPs.

The success of SLP tags for low-abundant proteins is partly due to the higher fluorogenic effect enabled by the protein surface, which has been further engineered to enhance fluorogenicity in more recent studies²⁹.

The above data and explanation have been included into the **main text on Page 4 and 6, new Fig. 2** in the main text, **Supplementary Text Page 2**, and **Extended Data Fig. 2 and 6**.

Extended Revision Response Fig. 2 Labelling of mCerulean3^{116TAG}-vimentin with various cell-permeable and cell-impermeable tetrazine dyes in COS-7 cells. The reporter protein was expressed in the presence of trans-cyclooct-2-en-L-lysine (TCO*A, an nCAA that can be labelled) or t-butylloxycarbonyl-L-lysine (BOC, a control nCAA that cannot be labelled). Cells are labelled with the cell-permeable dyes include Janelia Fluor 549 (JF549), Janelia Fluor 646 (JF646), silicon rhodamine (SiR), and TAMRA under living cell conditions. For the cell-impermeable dyes (AZDye594 and LD655, the FRET dye pair used in the manuscript), we semi-permeabilized cell with low-dosage digitonin. Intensity display range was set as 0-500 for mCerulean images and 0-2000 for other images. (Scale bars: 20 μ m)

References

1. Hülsmann, B. B., Labokha, A. A. & Görlich, D. The permeability of reconstituted nuclear pores provides direct evidence for the selective phase model. *Cell* **150**, 738–751 (2012).
2. Hellenkamp, B. *et al.* Precision and accuracy of single-molecule FRET measurements—a multi-laboratory benchmark study. *Nat. Methods* **15**, 669–676 (2018).
3. Wang, J. *et al.* A molecular grammar governing the driving forces for phase separation of prion-like RNA binding proteins. *Cell* **174**, 688–699.e16 (2018).
4. Tran, H. T. & Pappu, R. V. Toward an accurate theoretical framework for describing ensembles for proteins under strongly denaturing conditions. *Biophys J* **91**, 1868–1886 (2006).
5. des Cloizeaux, J. & Jannink, G. *Polymers in Solution: Their Modelling and Structure*. (Oxford University Press, 2010).
6. Ribbeck, K. & Görlich, D. The permeability barrier of nuclear pore complexes appears to operate via hydrophobic exclusion. *EMBO J* **21**, 2664–2671 (2002).
7. Yang, W., Gelles, J. & Musser, S. M. Imaging of single-molecule translocation through nuclear pore complexes. *Proc Natl Acad Sci U S A* **101**, 12887–12892 (2004).
8. Paci, G., Zheng, T., Caria, J., Zilman, A. & Lemke, E. A. Molecular determinants of large cargo transport into the nucleus. *Elife* **9**, e55963 (2020).
9. Chowdhury, R., Sau, A. & Musser, S. M. Super-resolved 3D tracking of cargo transport through nuclear pore complexes. *Nat. Cell Biol.* **24**, 112–122 (2022).
10. Beliu, G. *et al.* Bioorthogonal labeling with tetrazine-dyes for super-resolution microscopy. *Commun. Biol.* **2**, 261 (2019).
11. Arsić, A., Hagemann, C., Stajković, N., Schubert, T. & Nikić-Spiegel, I. Minimal genetically encoded tags for fluorescent protein labeling in living neurons. *Nat. Commun.* **13**, 314 (2022).
12. Nikić, I. *et al.* Debugging eukaryotic genetic code expansion for site-specific click-PAINT super-resolution microscopy. *Angew. Chem. Int. Ed.* **55**, 16172–16176 (2016).
13. Mihaila, T. S. *et al.* Enhanced incorporation of subnanometer tags into cellular proteins for fluorescence nanoscopy via optimized genetic code expansion. *Proc. Natl. Acad. Sci.* **119**, e2201861119 (2022).
14. Kong, X., Nir, E., Hamadani, K. & Weiss, S. Photobleaching Pathways in Single-Molecule FRET Experiments. *J. Am. Chem. Soc.* **129**, 4643–4654 (2007).
15. Hofmann, H. *et al.* Polymer scaling laws of unfolded and intrinsically disordered proteins quantified with single-molecule spectroscopy. *Proc Natl Acad Sci U S A* **109**, 16155–16160 (2012).
16. Fuertes, G. *et al.* Decoupling of size and shape fluctuations in heteropolymeric sequences reconciles discrepancies in SAXS vs. FRET measurements. *Proc. Natl. Acad. Sci.* **114**, E6342–E6351 (2017).
17. Tan, P. S. *et al.* Two differential binding mechanisms of FG-nucleoporins and nuclear transport receptors. *Cell Rep.* **22**, 3660–3671 (2018).
18. Milles, S. *et al.* Plasticity of an ultrafast interaction between nucleoporins and nuclear transport receptors. *Cell* **163**, 734–745 (2015).
19. Flory, P. J. The configuration of real polymer chains. *J Chem Phys* **17**, 303–310 (1949).
20. Ng, S. C. & Görlich, D. A simple thermodynamic description of phase separation of Nup98 FG domains. *Nat. Commun.* **13**, 6172 (2022).
21. Lakowicz, J. R. *Principles of fluorescence spectroscopy*. (Springer, 2006).
22. Lakowicz, J. R. *Principles of fluorescence spectroscopy*. (Springer, 2006).
23. Kalita, J. *et al.* Karyopherin enrichment and compensation fortifies the nuclear pore complex against nucleocytoplasmic leakage. *J. Cell Biol.* **221**, e202108107 (2022).
24. Schaffer, J. *et al.* Identification of Single Molecules in Aqueous Solution by Time-Resolved Fluorescence Anisotropy. *J. Phys. Chem. A* **103**, 331–336 (1999).
25. Koshioka, M., Sasaki, K. & Masuhara, H. Time-dependent fluorescence depolarization analysis in three-dimensional microspectroscopy. *Appl. Spectrosc.* **49**, 224–228 (1995).

26. Grimm, J. B. *et al.* A general method to improve fluorophores for live-cell and single-molecule microscopy. *Nat Methods* **12**, 244–250 (2015).
27. Wilhelm, J. *et al.* Kinetic and Structural Characterization of the Self-Labeling Protein Tags HaloTag7, SNAP-tag, and CLIP-tag. *Biochemistry* **60**, 2560–2575 (2021).
28. Uttamapinant, C. *et al.* Genetic Code Expansion Enables Live-Cell and Super-Resolution Imaging of Site-Specifically Labeled Cellular Proteins. *J. Am. Chem. Soc.* **137**, 4602–4605 (2015).
29. Frei, M. S. *et al.* Engineered HaloTag variants for fluorescence lifetime multiplexing. *Nat. Methods* **19**, 65–70 (2022).

Reviewer Reports on the First Revision:

Referee #1 (Remarks to the Author):

The revised manuscript from Yu et al. is significantly improved. The authors also adequately addressed the issues I originally raised regarding the application of the GCE technology in mammalian cells (context dependence, variability of expression for different double mutants, etc.). My only suggestion would be to incorporate these key strategies that the authors have eluded to in their response letter in the manuscript. These are some persistent challenges in the field, and the solutions that the authors have developed would be useful for the community. Looking forward to seeing this in print!

Abhishek Chatterjee

Referee #2 (Remarks to the Author):

The authors have revised their manuscript and the revisions are extensive, fully responsive, and persuasive. The apparent scaling exponent remains a mystery to me given the direction in which finite size effects should work. However, the analysis is comprehensive, and the issue, unresolved for now, is something that will require a technical deep dive that blends numerical simulations and scaling theory - well outside the scope of the current MS. The correlation hole analysis is persuasive as is the new stickers and spacers model. I have no further revisions to request. I believe this MS is timely and at the leading edge of its field. Finally, we have a glimpse into the organization of the hole. These models will provide a clear path forward for modeling selective transport and it will be front and center in terms of enabling our understanding how molecules within condensates are networked with one another.

Referee #3 (Remarks to the Author):

This is an outstanding manuscript. The extensive edits and lengthy explanations to my previous comments have largely satisfied my concerns. The in vivo experiments and the potential pitfalls and limitations will be helpful for more widespread applications and for improving the method. I provide below a list of grammar/spelling issues to assist with presentation and identify some other minor issues. Also, there are issues raised by the new Supplementary Fig. 20 (Fig. S20) that should be addressed/clarified.

Both R and RE are used in the manuscript for the same thing and this confusion should be clarified/rectified. For example, R is used in equations 12, 13, 15, and 17 but RE is used in Eq. 18, Supplementary Fig. 20, and throughout the text. While RE is defined as the end-to-end distance, in this manuscript it is frequently used for the distance between two fluorophores. Simply R might be better as the segment length of interest. Adding to the confusion, there is the variable (R or RE, which varies for each monomer chain in a conformationally-dependent manner), and an average parameter ($\langle RE^2 \rangle^{1/2}$, perhaps better denoted as rms RE, or R with subscript 'RMS'), which is

presumably invariant for a set of conditions.

It would be helpful to clarify how $\langle RE^2 \rangle^{1/2}$ is calculated from the simulations. I am presuming that fits such as those in Fig. S20 are used to estimate $\langle RE^2 \rangle^{1/2}$. The fits are particularly poor for low x for the blue and yellow data. While the orange data fit much better, the expression used raises questions. To explain, I step back a bit as Fig. S20 and the caption needs further explanation/clarification. The expression for $pF(x)$ (line 502, Fig. S20 caption) was presumably derived from Eq. 13 by a change of coordinates. For clarity, this should be more explicitly explained. It is not clear why the new function $pF(x)$ is introduced rather than simply using the change of coordinate expression $p(x) = (4\pi x^2) / ((3/2\pi)^{1.5}) \cdot \exp(-1.5x^2)$. Presumably this formalism is how the distance data are plotted and fit in Fig. S20, i.e., not using the authors' expression for $pF(x)$. The use of $p(x)$ and $pF(x)$ in the figure, caption, and y-axis label is confusing as it is not clear if these refer to the same expression. More importantly, since the exponents and pre-factors are known for $p(x)$, why are these allowed to all vary for the fits? Doesn't this imply that Eq. 13 isn't an appropriate model? The only reasonable variable here seems to be the scaling factor, a , which could be non-unity if the data are inappropriately normalized. Now returning to the primary concern. The deviations from the expected expression are unexplained, and there is a potential important explanation that has not been discussed. From my understanding, the assumption behind Eq. 13 is that it applies to a single polymer within a homogeneous solvent. What happens if the volume (conformational space) accessible to the polymer is limited? This is certainly the case for a polymer tethered to the NPC scaffold. While the data may in fact fit 'fairly well' to the assumed simple model, the concern is whether the inaccessible volume might influence the scaling law, and hence, the primary conclusion of the paper about a good vs. poor solvent. As a somewhat extreme example, note that the scaling law will be different for polymers attached to the inside wall of a hollow cylinder. While the experimental data inside the NPC very clearly are linearly related with low error, if the model used to determine the scaling law does not apply to the conditions used, it is not clear if the dividing line between good and poor solvent is $v = 0.5$. The simulations should be more carefully interpreted to determine the scaling law under the conditions of the experiment where excluded volume may influence the analysis. If $p(x)$ cannot be fit to the simulated data, the $\langle RE^2 \rangle^{1/2}$ should be obtained by weighted integration under the curve, rather than the assumed model. The simulations for the polymers in solution and in condensates should be used as controls as the scaling law from Eq. 13 is expected to be accurate under these conditions.

The non-linearity of the log-log inset in Fig. 4A hints at some deviation from the assumed scaling law and suggests a range of applicability. Are the curves in Fig. 4A fits or just connections to guide the eye? How were the values determined in Fig. 4A? – by fits such as those in Fig. S20? Can the curves in Fig. 4A be fit to a scaling law? If so, it would be helpful to indicate the scaling law for each curve. The y-axis in Fig. 4A is presumably $\langle RE^2 \rangle^{1/2}$ and not RE .

Assuming that the simulation algorithm is considered verified by the experimental data, the authors could determine if the scaling law is similar for all the FG-Nups. Density dependent differences may be observed (i.e., within the pore or at the periphery of the FG-Nup distribution).

While the author's explanation why the lifetime curves are so similar for the different FRET pairs is now clear (a high fraction of donor only population), it is not clear why this component cannot be

simply subtracted out to give the 'corrected' lifetime curves. This would make the differences more obvious for most readers. Supplementary Fig. 7 supports my previous point that the corrected intensity measurements (donor only and background populations subtracted out) yield much stronger signals and hence should give more robust results. Distance calculations made from intensity measurements assuming the Gaussian chain model (Supplementary Fig. 5) should either verify the main result or demonstrate that the approach does not work.

Other (minor) issues:

The accepted terminology is "permeabilized cells", not 'semi-permeabilized cells". The cells are either permeabilized or they are not. While the permeabilization state of the sub-cellular organelles is not addressed by the term, it is understood/known that the nuclear envelope remains intact under the permeabilization conditions.

In Fig. 2A, the 'anchor' domain is indicated as beginning at residue 738, but the anchor point in the MD model is residue 595. Explain briefly somewhere in the manuscript.

Lines 712-716 & Extended Data Fig. 4 – it is unclear how the authors measured fundamental anisotropies or what they expect the reader to take away from the measurements and discussion. Usually, if the anisotropy of the sample is smaller than the fundamental anisotropy, the fluorophores are sufficient rotationally mobile such that the kappa-squared assumption is safe. The fundamental anisotropy is not the measured anisotropy, but the maximum anisotropy that can be measured for a given angle between the excitation and emission dipoles (i.e., when the fluorophore is immobile).

Fig. S5B – While the static model yields Gaussian like distributions, the Gaussian chain model does not (the distribution is more log-normal-like). Doesn't this imply that the data in Extended Fig. 8 should not be fit with a bi-Gaussian?

Extended Data Fig. 9 – Do the Nup98 chains for the simulations include the GLEBS domain, which was deleted for the experimental measurements? Why does Fig. S20 summarize distance information for the entire length of the disordered Nup98 chain rather than for a segment tested in the experiments? Doesn't the GLEBS domain influence the end-to-end distance?

Spelling/grammar/usage issues:

p. 2, Summary Paragraph – 'Visualize' (lines 34 and 44) is too strong of a word. They are using fluorescence, yes, but they are not obtaining images. 'Probe' or 'interrogate' would be more appropriate

line 91 – 'expanded' should likely be 'extended'

line 92-93 – "could be combined well" reads better as 'were combined'

lines 165, 322 – delete 'very'; it is unnecessary and undefined

Line 314 – delete 'transport'. It is unnecessary, and the sentence works better as a general statement.

Line 342 – under what conditions is $v \approx 1$ for a polymer brush? This corresponds to perfectly extended chains, which seems entropically impossible. A reference is needed here. This should also be addressed in Fig. 3 (which certainly cannot occur at the grafting density shown).

Line 343 – For clarity, ' $v \approx 0.3$ ' should be moved after 'forest model' on the previous line.

Line 355 – 'Sophisticated' is unnecessary

Line 356 – Add 'alone' after 'methods'

Line 358 – Add 'the tested' before 'membrane-permeable'

Line 383 – 'exact' is overly precise.

Fig. 3 – For the key, it would be simpler to use 'in condensates', 'inside NPCs', and 'in solution'

Eq. 12 – what is the mathematical expression for $E(R)$? Is this the equation used for FRET efficiency in Fig. S5 and in the supplementary info text (p. 6)? Probably should give this an equation number so it is easier to refer to.

Supplementary Info:

Numerous Supplementary Figures have reduced resolution/small text. These could easily be enlarged to make better use of space.

Line 114 – 'obtained' instead of 'come' would be better.

Line 132 - 'encouraging' instead of 'ensuring' would be better.

Line 290 – 'separation' is misspelled.

Fig. S1A – what do the boxes & lines represent?

Fig. S4 & S9 – The slopes are not 1 – is there any explanation?

Fig. S8 – It would be helpful to indicate the Flory limits (perhaps as dashed horizontal lines?)

Fig. S13 – the lines going all the way to the right margin are confusing. It would be clearer if these end at the end of the protein. The numbers are largely unreadable.

Fig. S15 – the experiment corresponds to the green data in Fig. 3, right? Why are there only 5 points and not 6?

Fig. S18 – 'Side' and 'front' nomenclature is confusing. Perhaps 'arc' and 'radial', with an illustration?

Author Rebuttals to First Revision:

RESPONSE TO REVIEWER COMMENTS

Reviewer comments in black

Our response in blue

Referee #1 (Remarks to the Author):

The revised manuscript from Yu et al. is significantly improved. The authors also adequately addressed the issues I originally raised regarding the application of the GCE technology in mammalian cells (context dependence, variability of expression for different double mutants, etc.). My only suggestion would be to incorporate these key strategies that the authors have eluded to in their response letter in the manuscript. These are some persistent challenges in the field, and the solutions that the authors have developed would be useful for the community. Looking forward to seeing this in print! (Abhishek Chatterjee)

We thank again the reviewer for his positive assessment of our work. We have already added these key strategies in the discussion in the main text as well as in the supplementary text.

Referee #2 (Remarks to the Author):

The authors have revised their manuscript and the revisions are extensive, fully responsive, and persuasive. The apparent scaling exponent remains a mystery to me given the direction in which finite size effects should work. However, the analysis is comprehensive, and the issue, unresolved for now, is something that will require a technical deep dive that blends numerical simulations and scaling theory - well outside the scope of the current MS. The correlation hole analysis is persuasive as is the new stickers and spacers model. I have no further revisions to request. I believe this MS is timely and at the leading edge of its field. Finally, we have a glimpse into the organization of the hole. These models will provide a clear path forward for modeling selective transport and it will be front and center in terms of enabling our understanding how molecules within condensates are networked with one another.

We thank the reviewer for praising our work!

Referee #3 (Remarks to the Author):

This is an outstanding manuscript. The extensive edits and lengthy explanations to my previous comments have largely satisfied my concerns. The in vivo experiments and the potential pitfalls and limitations will be helpful for more widespread applications and for improving the method. I provide below a list of grammar/spelling issues to assist with presentation and identify some other minor issues. Also, there are issues raised by the new Supplementary Fig. 20 (Fig. S20) that should be addressed/clarified.

We thank the reviewer for the dedicated comments and for the great effort in improving our manuscript! It is much appreciated.

Both R and RE are used in the manuscript for the same thing and this confusion should be clarified/rectified. For example, R is used in equations 12, 13, 15, and 17 but RE is used in Eq. 18, Supplementary Fig. 20, and throughout the text. While RE is defined as the end-to-end distance, in this manuscript it is frequently used for the distance between two fluorophores. Simply R might be better as the segment length of interest. Adding to the confusion, there is the variable (R or RE, which varies for each monomer chain in a conformationally-dependent manner), and an average parameter ($\langle RE^2 \rangle^{1/2}$, perhaps better denoted as rms RE, or R with subscript 'RMS'), which is presumably invariant for a set of conditions.

We thank the reviewer for alerting us about the inconsistent nomenclature. We also realized that the previous definition of R_E could be confusing for the readers. Therefore, we now define the root-mean-square inter-residue distance as $\sqrt{\langle r^2 \rangle} \equiv R_E$, where r refers to a specific inter-residue distance with its probability as $p(r)$. And indeed, we are particularly grateful for the reviewer to point this out, as in some cases we even confused this ourself, and reported RMS even when the figure reported mean R, which is of course wrong. We have now checked everything carefully again, and revised accordingly in the Main text, Extended data, and SI. None of the conclusions were affected by addressing this issue.

It would be helpful to clarify how $\langle RE^2 \rangle^{1/2}$ is calculated from the simulations. I am presuming that fits such as those in Fig. S20 are used to estimate $\langle RE^2 \rangle^{1/2}$. The fits are particularly poor for low x for the blue and yellow data. While the orange data fit much better, the expression used raises questions. To explain, I step back a bit as Fig. S20 and the caption needs further explanation/clarification. The expression for $pF(x)$ (line 502, Fig. S20 caption) was presumably derived from Eq. 13 by a change of coordinates. For clarity, this should be more explicitly explained. It is not clear why the new function $pF(x)$ is introduced rather than simply using the change of coordinate expression $p(x) = (4\pi x^2) / ((3/2\pi)^{1.5}) \cdot \exp(-1.5x^2)$. Presumably this formalism is how the distance data are plotted and fit in Fig. S20, i.e., not using the authors' expression for $pF(x)$. The use of $p(x)$ and $pF(x)$ in the figure, caption, and y-axis label is confusing as it is not clear if these refer to the same expression. More importantly, since the exponents and pre-factors are known for $p(x)$, why are these allowed to all vary for the fits? Doesn't this imply that Eq. 13 isn't an appropriate model? The only reasonable variable here seems to be the scaling factor, a , which could be non-unity if the data are inappropriately normalized. Now returning to the primary concern. The deviations from the expected expression are unexplained, and there is a potential important explanation that has not been discussed. From my understanding, the assumption behind Eq. 13 is that it applies to a single polymer within a homogeneous solvent. What happens if the volume (conformational space) accessible to the polymer is limited? This is certainly the case for a polymer tethered to the NPC scaffold. While the data may in fact fit 'fairly well' to the assumed simple model, the concern is whether the inaccessible volume might influence the scaling law, and hence, the primary conclusion of the paper about a good vs. poor solvent. As a somewhat extreme example, note that the scaling law will be different for polymers attached to the inside wall of a hollow cylinder. While the experimental data inside the NPC very clearly are linearly related with low error, if the model used to determine the scaling law does not apply to the conditions used, it is not clear if the dividing line between good and poor solvent is $v = 0.5$. The simulations should be more carefully interpreted to determine the scaling law under the conditions of the experiment where excluded volume may influence the analysis. If $p(x)$ cannot be fit to the simulated data, the $\langle RE^2 \rangle^{1/2}$ should be obtained

by weighted integration under the curve, rather than the assumed model. The simulations for the polymers in solution and in condensates should be used as controls as the scaling law from Eq. 13 is expected to be accurate under these conditions.

We thank the reviewer for the rigorous help to make this an excellent manuscript and for seeking clarification of SI Fig. S20 (now Fig. S21). We apologize for not explaining this well enough. We reply first to point (A) about how the root-mean inter-residue distance (we now define as $R_E = \sqrt[2]{\langle r^2 \rangle}$) is calculated from the simulations, then to (B) how the polymer model affects the scaling relation, and finally to (C) how SI Fig. S21 was adapted in response to the reviewer comments.

(A) Calculation of polymer extension from simulations

We now consistently report root-mean-square (RMS) inter-residue distances. From the simulations, the RMS distances were calculated by averaging the squared distance r between the residue pairs probed in the FLIM-FRET experiments and then taking the square root of the average, $R_E = \langle r^2 \rangle^{0.5}$. The averages were calculated over the production part of the molecular dynamics simulations and over all NUP98 chains (except in SI Fig. S20, where we resolve distances of the NUP98 chains according to where in the NPC scaffold they are anchored).

(B) How the polymer model affects the scaling relation

We thank the reviewer for challenging us to make the analysis of the fluorescence lifetime consistent with the distance distributions observed in the molecular dynamics (MD) simulations. As we show in the updated SI Fig. S21 (previously S20), the polymers in the MD simulations are well described by des Cloizeaux's theory for self-avoiding random walks. To strengthen this point, we now compare the distance distributions for the NUP98 residue pairs probed by the FLIM-FRET experiments, as suggested by the reviewer. As a hallmark of this theory, the distance distribution approaches zero at short distances even after accounting for the Jacobian $r^2 dr$. The distance distribution in des Cloizeaux's theory is also the basis of the so-called SAW- ν model of polymers, which has been widely used to analyse single-molecule fluorescence distance measurements. In response to the referee's challenge, we thus now use the SAW- ν model to analyse the fluorescence distance measurements in the nuclear pore complex. Based on the excellent agreement of the SAW- ν model with the simulation data (as shown in the updated SI Fig. S21), we consider the simulation model to be fully consistent with the fluorescence distance analysis.

Specifically, we now performed a global fitting of our experimental data to SAW- ν model as described by Zheng et al (2018)¹

$$\rho(r) = A \frac{4\pi}{\sqrt[2]{\langle r^2 \rangle}} \left(\frac{r}{\sqrt[2]{\langle r^2 \rangle}} \right)^{2+g} e^{-B \left(\frac{r}{\sqrt[2]{\langle r^2 \rangle}} \right)^\delta} \quad [1]$$

where $\sqrt[2]{\langle r^2 \rangle} \equiv R_E$. A and B are constants determined by the normalization condition $\int_0^\infty p(r) dr = 1$ and $\int_0^\infty p(r) r^2 dr = R_E^2$. The exponents in the SAW- ν model follow des Cloizeaux's theory and satisfy $\delta = 1/(1 - \nu)$, and $g = (\nu - 1)/\nu$ with $\nu \approx 1.1615$.

As closure relation for proteins, the SAW- ν model of Zheng et al (2018)¹ uses

$$R_E = bN_{res}^\nu \quad [2]$$

with $b = 0.55$ nm (which is estimated for proteins in good solvent¹).

From our fit of the SAW- ν model, we obtained the scaling exponent as $\nu = 0.56 \pm 0.001$ (see updated Extended Data Fig. 7). The perfect agreement with $\nu = 0.56 \pm 0.03$ obtained from our analysis using Gaussian chains shows the robustness of our scaling exponent for different polymer models and further strengthens our analysis.

(C) Clarifications to SI Fig. S21 and the effects of NPC confinement on polymer extension

We thank the reviewer for pointing out some notational issues and possible confusions concerning the SI Fig. S21 (previously S20). To address these issues, we now (i) show the distributions $p(x)$ of the inter-residue distances (for residue pairs probed by the experiments) without removing the Jacobian $x^2 dx$, and compare these distributions directly to the predictions (ii) of de Cloizeaux's theory (solid line) and (iii) of Flory's Gaussian-chain model (dashed line) defined in the caption as $p_F(x) = 4\pi \left(\frac{3}{2\pi}\right)^{3/2} x^2 \exp(-1.5x^2)$. Importantly, de Cloizeaux's theory accounts accurately for all inter-residue distance distributions probed in the experiments. De Cloizeaux's theory is also the basis of the SAW- ν model now used to analyze the experiments, as shown in Extended Data Fig. 7. The excellent agreement between de Cloizeaux's theory for self-avoiding walks and the distributions sampled in the MD simulations inside the NPC, as shown in the updated SI Fig. S21, addresses also the concern that the scaling relation might be affected by the restricted volume inside the NPC scaffold. Further supporting this point, we now show in the new SI Fig. 20d that the scaling law Eq. 18 applies both across all NUP98 chains and individually depending on their location in the NPC (see following point). The simulations thus provide a further control for the accuracy of the scaling law.

The non-linearity of the log-log inset in Fig. 4A hints at some deviation from the assumed scaling law and suggests a range of applicability. Are the curves in Fig. 4A fits or just connections to guide the eye? How were the values determined in Fig. 4A? – by fits such as those in Fig. S20? Can the curves in Fig. 4A be fit to a scaling law? If so, it would be helpful to indicate the scaling law for each curve. The y-axis in Fig. 4A is presumably $\langle RE^2 \rangle^{1/2}$ and not RE.

In Fig. 4A, the lines are guides to the eye, as is now mentioned in the caption. In response to the questions raised, we now performed a distance scaling analysis for the NUP98 chains overall and resolved by the different anchoring points in the NPC. The scaling-law fits are shown in SI Fig. S20d for the MD simulations of model II with a complete set of FG-NUPs for $\tilde{\epsilon} = 0.42$. See also Response Figure 1 below. The average and standard deviation of the scaling exponents in the NPC simulations are $\nu = 0.521 \pm 0.038$. For reference, the experimental scaling exponent for NUP98 is $\nu = 0.56 \pm 0.03$, in good agreement.

Response Fig. 1 Distance scaling of NUP98 FG domains grouped by the grafting positions in NPC model II. RMS distance (R_E) of beads on NUP98 FG domain in the NPC as function of residue separation N_{res} for homopolymer model II on a log-log scale. The interaction strength is $\tilde{\epsilon} = 0.42$. The 8-fold symmetric human NPC contains $8 \times 6 = 48$ NUP98 chains in six positions (see Supplementary Tables 2 and 3 and Supplementary Figure 19). Lines show fits to a scaling law, $R_E = a N_{res}^\nu$, with parameters a and ν . The fitting parameter values are $a = 5.88 \text{ \AA}$ and $\nu = 0.567$ for UCc chains; $a = 8.23 \text{ \AA}$ and $\nu = 0.493$ for UNc chains; $a = 6.5 \text{ \AA}$ and $\nu = 0.549$ for Uci chains; $a = 7.84 \text{ \AA}$ and $\nu = 0.497$ for Uco chains; $a = 8.94 \text{ \AA}$ and $\nu = 0.474$ for Uni chains; $a = 6.53 \text{ \AA}$ and $\nu = 0.544$ for Uno chains and $a = 7.2 \text{ \AA}$ and $\nu = 0.520$ for all NUP98 chains. The average and standard deviation of scaling exponents in simulations are $\nu_{sim} = 0.521 \pm 0.038$. The experimental scaling exponent for NUP98 is shown as black circles ($\nu = 0.56 \pm 0.03$).

Assuming that the simulation algorithm is considered verified by the experimental data, the authors could determine if the scaling law is similar for all the FG-Nups. Density dependent differences may be observed (i.e., within the pore or at the periphery of the FG-Nup distribution).

We thank the reviewer for asking us to use the simulations to probe for possible variations in the scaling behaviour. In response, also to the preceding point, we now performed a distance scaling analysis for the NUP98 chains at the different anchoring points, as is now shown in Supplementary Figure S20 and in Response Figure 1 (above). In this way, we could show that the different environments with different FG-NUP densities slightly affect the scaling behaviour, giving a range of scaling exponents ν from 0.47 to 0.57. The lowest exponent is found in the inner ring, where the density is comparably high, and the highest in the cytoplasmic ring, where the density is comparably low.

While the author's explanation why the lifetime curves are so similar for the different FRET pairs is now clear (a high fraction of donor only population), it is not clear why this component cannot be simply subtracted out to give the 'corrected' lifetime curves. This would make the

differences more obvious for most readers. Supplementary Fig. 7 supports my previous point that the corrected intensity measurements (donor only and background populations subtracted out) yield much stronger signals and hence should give more robust results. Distance calculations made from intensity measurements assuming the Gaussian chain model (Supplementary Fig. 5) should either verify the main result or demonstrate that the approach does not work.

We thank the reviewer for asking these insightful questions. We would like to point out that Supplementary Fig. 2 (previously SI Fig. 7) is a plot of lifetime decay differences before and after acceptor photobleaching, $I_{\text{diff}}(t) = A_{\text{FRET}} \left\{ e^{-\frac{t}{\tau_D}} - \int_0^\infty \rho(R) e^{-\frac{t}{\tau_D} [1 + (\frac{R_0}{R})^6]} dR \right\} \otimes IRF$, as described by SI Eq. 17, which reports on FRET-only population, but it is not the lifetime curve of the FRET signal of the donor. To acquire a plot of lifetime curve of the FRET signal (with donor-only and background signals subtracted out), $I_{\text{FRET}}(t) = \left\{ \int_0^\infty \rho(R) e^{-\frac{t}{\tau_D} [1 + (\frac{R_0}{R})^6]} dR \right\} \otimes IRF$, we have to first fit the lifetime curves with our model to know the ratio of each population and then perform the subtraction. We now performed such a calculation, and show the lifetime curves of the FRET signal of the donor in new Extended Data Figure 5b, where more pronounced differences can be seen.

We also thank the reviewer for challenging us to perform an intensity-based FRET analysis. We now perform such an analysis by assuming the Gaussian chain model as requested. The FRET efficiency E is calculated by,

$$E = \frac{I_{A_FRET}^D}{\gamma_{cor} I_{D_FRET}^D + I_{A_FRET}^D} \quad [3]$$

where $I_{A_FRET}^D$ is the acceptor fluorescence intensity of FRET signal detected upon donor excitation, $I_{D_FRET}^D$ is the donor fluorescence intensity of FRET signal detected upon donor excitation, and γ_{cor} is the factor accounting for the detection efficiency of the acceptor and donor channels. The total intensity of donor channel upon donor excitation $I_{D_total}^D$ can be described as,

$$I_{D_total}^D = I_{D_only}^D + I_{D_FRET}^D + I_{D_bg}^D \quad [4]$$

where $I_{D_only}^D$ is the donor-only signal, and $I_{D_bg}^D$ is the background signal. The total intensity of acceptor channel upon donor excitation $I_{A_total}^D$ is given by,

$$\begin{aligned} I_{A_total}^D &= I_{A_direct}^D + I_{A_FRET}^D + I_{D_leakage}^D \\ &= \delta I_{A_total}^A + I_{A_FRET}^D + \alpha I_{D_total}^D \end{aligned} \quad [5]$$

where $I_{A_direct}^D$ is the direct excitation of acceptors by the donor laser, which can be corrected with the acceptor fluorescence detected upon acceptor excitation $I_{A_FRET}^A$ by the factor δ , and $I_{D_leakage}^D$ is the leakage of donor channel into the acceptor channel, which can be corrected with $I_{D_total}^D$ by the factor α .

As we are at the ensemble level, we have to fit the lifetime profiles of donor channel ($I_{D_total}^D(t)$) to SI Eq. 15 in SI to determine the fractions of each population by assuming a

Gaussian chain model. Note that A_{FRET} determined by SI Eq. 15 is the FRET species fraction, not the intensity fraction. The intensity fraction of the FRET signal S_{FRET} can be calculated as,

$$S_{\text{FRET}} = \frac{I_{D_{\text{FRET}}}^D}{I_{D_{\text{total}}}^D}$$

$$= \frac{A_{\text{FRET}} \int_0^\infty \rho(R) e^{-\frac{t}{\tau_D} [1 + (\frac{R_0}{R})^6]} dR \otimes \text{IRF}}{\left\{ A_D e^{-\frac{t}{\tau_D}} + A_{\text{FRET}} \int_0^\infty \rho(R) e^{-\frac{t}{\tau_D} [1 + (\frac{R_0}{R})^6]} dR + A_{\text{bg}} \sum_{i=1}^N \alpha_i e^{-\frac{t}{\tau_{\text{bgi}}}} \right\} \otimes \text{IRF}} \quad [6]$$

By substituting Eq. 4-6 into Eq. 3, we can obtain the FRET efficiency E as,

$$E = \frac{I_{A_{\text{total}}}^D - \delta I_{A_{\text{total}}}^A - \alpha I_{D_{\text{total}}}^D}{\gamma S_{\text{FRET}} I_{D_{\text{total}}}^D + I_{A_{\text{total}}}^D - \delta I_{A_{\text{total}}}^A - \alpha I_{D_{\text{total}}}^D} \quad [7]$$

Because we always first measured ~5 min with 560 nm laser excitation, and then ~30 sec with 660 nm laser excitation, we added a correction factor n for $I_{A_{\text{total}}}^A$ to compensate for the different measurement duration compared to $I_{A_{\text{total}}}^D$ and $I_{D_{\text{total}}}^D$. We also experimentally determined the correction factors σ_{DD} , σ_{DA} , and σ_{AA} for $I_{D_{\text{total}}}^D$, $I_{A_{\text{total}}}^D$, and $I_{A_{\text{total}}}^A$ to compensate for the photobleaching. Therefore, the corrected FRET efficiency is,

$$E = \frac{\sigma_{DA} I_{A_{\text{total}}}^D - \sigma_{AA} n \delta I_{A_{\text{total}}}^A - \alpha \sigma_{DD} I_{D_{\text{total}}}^D}{\gamma S_{\text{FRET}} \sigma_{DD} I_{D_{\text{total}}}^D + \sigma_{DA} I_{A_{\text{total}}}^D - \sigma_{AA} n \delta I_{A_{\text{total}}}^A - \alpha \sigma_{DD} I_{D_{\text{total}}}^D} \quad [8]$$

We calculated E based on this Eq. 8 on a single-cell basis for 3 mutants with shorter, middle, longer chain lengths ($N_{\text{res}}=30\text{aa}$, 117aa , 285aa , as shown in Response Fig. 2A) and obtained R_E using the data shown in Supplementary Fig. 5a. For each of the three mutants, the mean from ~100 measured cells is shown and the error bars in Response Fig. 2A show standard deviation. We called this method semi-intensity-based, because S_{FRET} for each mutant has to be derived from lifetime analysis. As shown in Response Fig. 2B below, R_E calculated from semi-intensity-based agrees well from the results in lifetime-based analysis, which verifies our main result, as requested by the reviewer.

Response Fig. 2 Comparison between the (A) FRET efficiency E and (B) root-mean-square inter-residue distance R_E obtained from lifetime-based analysis and semi-intensity-based analysis by assuming a Gaussian chain model. 3 mutants with shorter, middle, longer chain lengths ($N_{res}=30aa, 117aa, 285aa$) were analysed. Error bar in (A) represent the standard deviation of ~ 100 cells for each mutant.

Other (minor) issues:

The accepted terminology is “permeabilized cells”, not ‘semi-permeabilized cells’. The cells are either permeabilized or they are not. While the permeabilization state of the sub-cellular organelles is not addressed by the term, it is understood/known that the nuclear envelope remains intact under the permeabilization conditions.

We changed the term as ‘permeabilized cells’ as suggested in the Main text, Extended data, and SI.

In Fig. 2A, the ‘anchor’ domain is indicated as beginning at residue 738, but the anchor point in the MD model is residue 595. Explain briefly somewhere in the manuscript.

We thank the reviewer for alerting us of this inconsistency. We corrected Fig. 2A to indicate that according to the structural model, the NUP98 anchoring region starts already at position 595.

Lines 712-716 & Extended Data Fig. 4 – it is unclear how the authors measured fundamental anisotropies or what they expect the reader to take away from the measurements and discussion. Usually, if the anisotropy of the sample is smaller than the fundamental anisotropy, the fluorophores are sufficient rotationally mobile such that the kappa-squared assumption is safe. The fundamental anisotropy is not the measured anisotropy, but the maximum anisotropy that can be measured for a given angle between the excitation and emission dipoles (i.e., when the fluorophore is immobile).

We thank the reviewer for pointing out that our nomenclature was inconsistent with the standard and can lead to confusion. What we referred to in lines 712-716 & Extended Data

Fig. 4 was the measured fluorescence anisotropy, not the fundamental anisotropy. In response, we revised the text and Extended Data Fig. 4. Since the measured fluorescence anisotropy was always below 0.3, we can assume $\kappa^2 \sim 2/3$.

Fig. S5B – While the static model yields Gaussian like distributions, the Gaussian chain model does not (the distribution is more log-normal-like). Doesn't this imply that the data in Extended Fig. 8 should not be fit with a bi-Gaussian?

We thank the reviewer for asking the question. We want to point out that Fig. S5b and Extended Data Fig. 8 are different plots. Fig. S5b describes how the theoretical FRET efficiency changes depends on polymer chain lengths. It does not include any photon statistics inherent to the measurement. In contrast, each of the plots in Extended Fig. 8 describes the single molecule FRET histogram for a given mutant. In each plot, there are two peaks. The first peak originates from the so-called donor-only species, where the acceptor was absent, either due to bleaching or insufficient labelling. The second peak gives the FRET ratio. The width is primarily related to photon statistics, which varies in each detection channel depending on FRET efficiency. The benefit of single-molecule measurements is that these two populations can be nicely separated and the centre of mass can be nicely extracted, in particular when alternating donor and acceptor laser excitation², as we did here.

Extended Data Fig. 9 – Do the Nup98 chains for the simulations include the GLEBS domain, which was deleted for the experimental measurements?

We thank the reviewer for asking us to clarify this point. In response, the GLEBS domain was present in the *in situ* NPC experiments and deleted for most of the *in vitro* experiments. *In vitro* phase separation of Nup98 is triggered by rapid transfer from denaturing buffer to physiological buffer. How fast a folded domain folds vs. misfolds is currently speculation and indeed, we did observe morphological differences between droplets formed by the NUP98 constructs with GLEBS and without GLEBS, which is in line with related observations in the literature³. For *in vitro* condensates, even samples without GLEBS domain still harden out over time, so some underlying kinetics are going on, potentially due to partial misfolding/aggregation. In contrast to our measurements *in situ* (stable for >2 hrs), the lifetime curves for *in vitro* condensates are not stable over hours and the signal slowly changes. We thus always limited ourselves to the first 5 minutes of freshly formed droplets, where they still show liquid-like characteristics. In the simulations, the GLEBS domain was included consistently, as there was no aggregation issue. In particular, for the simulations of the NPC, we used NUP98(2-615). In the simulations of the condensate and isolated chain, we used NUP98(1-499), which also includes the GLEBS domain. We now alert the reader of this point and note that the GLEBS domain is outside the residue segments probed by the experiments (we only probed the FG domain). We note here that based on recent NMR measurements⁴, the GLEBS domain is unstructured in solution with the exception of a very short helix (see Fig. 1f in response ref 4). Notably, for the *in vitro* condensates, with and without GLEBS the measured scaling is in line with good solvent conditions (we now show both experimental data in Extended Data Fig. 10d (previously Extended Data Fig. 9d)).

Why does Fig. S20 summarize distance information for the entire length of the disordered Nup98 chain rather than for a segment tested in the experiments?

We thank the reviewer for raising this point. In response, we now show in the updated Supplementary Figure S21 the distributions for selected inter-residue distances probed by the FLIM-FRET experiments. We compare these distributions to the polymer theories of de Cloizeaux and Flory.

Doesn't the GLEBS domain influence the end-to-end distance?

In response, we now state the GLEBS domain is outside the residue segments whose distances are probed in the simulations or experiments.

Spelling/grammar/usage issues:

p. 2, Summary Paragraph – ‘Visualize’ (lines 34 and 44) is too strong of a word. They are using fluorescence, yes, but they are not obtaining images. ‘Probe’ or ‘interrogate’ would be more appropriate

We revised accordingly.

line 91 – ‘expanded’ should likely be ‘extended’

We revised accordingly.

line 92-93 – “could be combined well” reads better as ‘were combined’

We revised accordingly.

lines 165, 322 – delete ‘very’; it is unnecessary and undefined

We deleted ‘very’.

Line 314 – delete ‘transport’. It is unnecessary, and the sentence works better as a general statement.

We deleted ‘transport’.

Line 342 – under what conditions is $v \approx 1$ for a polymer brush? This corresponds to perfectly extended chains, which seems entropically impossible. A reference is needed here. This should also be addressed in Fig. 3 (which certainly cannot occur at the grafting density shown).

We thank the reviewer for asking the question. Here we referred to polymer brushes as the surface-tethered polymer chains with the distance between neighbouring chains s , smaller than twice the Flory radius R_F , as in contrast to the ‘mushroom’ regime where $s > 2R_F$. The thickness L of the polymer brush can be expressed as $L \approx a(\sigma)^{1/3}N$, where σ is the grafting density, a is monomer dimension, and N is degree of polymerization (see Eq. 10 on page 1071 in Ref 46 as cited in the main text). As $R_E \propto L$, one can conclude that $R_E \sim N$. That is, the exponent of N is ~ 1 for densely grafted polymer brushes.

In response, we revised the illustration of the polymer brush in Fig. 3 with higher density.

Line 343 – For clarity, '($v \approx 0.3$)' should be moved after 'forest model' on the previous line.

We revised accordingly.

Line 355 – 'Sophisticated' is unnecessary

We deleted 'sophisticated'.

Line 356 – Add 'alone' after 'methods'

We added 'alone'.

Line 358 – Add 'the tested' before 'membrane-permeable'

We added 'the tested'.

Line 383 – 'exact' is overly precise.

We deleted 'exact'.

Fig. 3 – For the key, it would be simpler to use 'in condensates', 'inside NPCs', and 'in solution'

We revised accordingly. Thank you!

Eq. 12 – what is the mathematical expression for $E(R)$? Is this the equation used for FRET efficiency in Fig. S5 and in the supplementary info text (p. 6)? Probably should give this an equation number so it is easier to refer to.

We now added the expression for $E(r)$ in Eq.12. Thank you!

Supplementary Info:

Numerous Supplementary Figures have reduced resolution/small text. These could easily be enlarged to make better use of space.

We adjusted the Supplementary Figures with better resolution/enlarged text as suggested.

Line 114 – 'obtained' instead of 'come' would be better.

We revised as suggested.

Line 132 - 'encouraging' instead of 'ensuring' would be better.

We revised as suggested.

Line 290 – ‘separation’ is misspelled.

We revised as suggested.

Fig. S1A – what do the boxes & lines represent?

We added in the legend of Fig. S3a (previously Fig. S1A) that “The box limits represent the range between the first and third quartiles for each mutant, the centre lines show the median, the central square shows the mean, and the ends of the whiskers extend to 1.5× the interquartile range.”

Fig. S4 & S9 – The slopes are not 1 – is there any explanation?

We apologize for the confusion in Supplementary Fig. S4. We now replot the reference line as $y = x$ in Fig. S4 and recalculated the $R^2=0.96$.

We note that Supplementary Fig. S10 (previously Fig. S9) shows a linear relation between χ and $\tilde{\epsilon}$, for which the slope is not expected to be one.

Fig. S8 – It would be helpful to indicate the Flory limits (perhaps as dashed horizontal lines?)

In new SI Fig. S9A (previously Fig. S8), we added horizontal dotted and dashed lines to indicate the Flory estimates for self-avoiding walks ($R_g = N^{0.6}\sigma/\sqrt{6} = 101.8\text{\AA}$) and random coils ($R_g = N^{0.5}\sigma/\sqrt{6} = 54.7\text{\AA}$), respectively. In the inset of SI Fig. S9B, we added the normalized distributions of the end-to-end distance of NUP98 chain in the MARTINI model and the homopolymer model in the limit of weak cohesion as well as the Flory distribution of a random coil.

Fig. S13 – the lines going all the way to the right margin are confusing. It would be clearer if these end at the end of the protein. The numbers are largely unreadable.

We revised Fig. S13 as suggested (see new SI Fig. 14).

Fig. S15 – the experiment corresponds to the green data in Fig. 3, right? Why are there only 5 points and not 6?

We thank the reviewer for pointing this out and apologize for the mistake. As mentioned above, we worked in vitro over the course of experiments with different Nup98 constructs (with and without GLEBS domain) and had a data swap here, which we now corrected in the new SI Fig. 16 (previously Fig. S15). We now explain in the caption that we have two experimental data sets for the NUP98 condensate, with and without GLEBS domain.

Fig. S18 – ‘Side’ and ‘front’ nomenclature is confusing. Perhaps ‘arc’ and ‘radial’, with an illustration?

We thank the reviewer for the suggestion. We revised SI Fig. 19 (previously Fig. S18) accordingly.

Response references

1. Zheng, W. *et al.* Inferring properties of disordered chains from FRET transfer efficiencies. *J. Chem. Phys.* **148**, 123329 (2018).
2. Kapanidis, A. N. *et al.* Alternating-laser excitation of single molecules. *Acc. Chem. Res.* **38**, 523–533 (2005).
3. Ng, S. C., Güttler, T. & Görlich, D. Recapitulation of selective nuclear import and export with a perfectly repeated 12mer GLFG peptide. *Nat. Commun.* **12**, 4047 (2021).
4. Ibáñez de Opakua, A. *et al.* Molecular interactions of FG nucleoporin repeats at high resolution. *Nat. Chem.* **14**, 1278–1285 (2022).

Reviewer Reports on the Second Revision:

Referee #3 (Remarks to the Author):

This is an outstanding manuscript. The authors have satisfactorily addressed my questions and concerns.

While the manuscript is ready to go as is, I provide a few grammatical corrections and comments for the authors to address as they see fit.

lines 26-27 – ‘...for transport between the nucleus and cytosol’ reads better

lines 39-40 – ‘...transport between the nucleus and cytoplasm’ reads better

line 54 – ‘passage’ instead of ‘crossing’

lines 145-146 – ‘...determined the average spatial distances...’

lines 311-312 – ‘...barrier of the NPC,...’

lines 523-524 – ‘The permeability of the droplet-like condensates formed was measured rapidly while they still...’ is more accurate.

Fig. 3 – The protein in the NPC was labeled with a different dye combination (and presumably the linkers) than in solution and in condensates. This should be noted in the caption.

lines 600-602 – reference for the TB composition? I do not understand how PEG6000 avoids osmotic shock, as I expect it to freely cross the NPC permeability barrier. Osmotic balance requires a non-permeable osmolyte to balance the water concentration inside and outside the nucleus, thus preventing swelling.

line 615 – what was the incubation time before imaging dextran permeability?

lines 616-618 – references for purification of importin beta, Ran and NTF2?

line 701 – why is the refractive index = 1.52 for droplets at the surface? The refractive index is for the medium in which the fluorophores are sitting for the measurement (not, presumably, for the glass that is nearby). What manufacturer's datasheet?

line 710 – 'expected' is better than 'supposed'

lines 709-717 – the theoretical maximum anisotropy is 0.4 when the excitation and emission dipoles are parallel. This is normally a pretty safe assumption, but not always. Significant deviation could lead to a maximum anisotropy of 0.28 (e.g., angle of 27 degrees), in which case the anisotropy measurement would not support free rotation. It would be best to note that they are assuming parallel (or near parallel) excitation and emission dipoles.

RESPONSE TO REVIEWER COMMENTS

Reviewer comments in black

Our response in blue

Referee #3 (Remarks to the Author):

This is an outstanding manuscript. The authors have satisfactorily addressed my questions and concerns. While the manuscript is ready to go as is, I provide a few grammatical corrections and comments for the authors to address as they see fit.

We thank the referee for helping further improve our manuscript!

lines 26-27 – ‘...for transport between the nucleus and cytosol’ reads better

We revised accordingly.

lines 39-40 – ‘...transport between the nucleus and cytoplasm’ reads better

We revised accordingly.

line 54 – ‘passage’ instead of ‘crossing’

We revised accordingly.

lines 145-146 – ‘...determined the average spatial distances...’

We revised accordingly.

lines 311-312 – ‘...barrier of the NPC,...’

We revised accordingly.

lines 523-524 – ‘The permeability of the droplet-like condensates formed was measured rapidly while they still...’ is more accurate.

We revised accordingly.

Fig. 3 – The protein in the NPC was labeled with a different dye combination (and presumably the linkers) than in solution and in condensates. This should be noted in the caption.

We added the information about the dye combination in the caption.

lines 600-602 – reference for the TB composition? I do not understand how PEG6000 avoids osmotic shock, as I expect it to freely cross the NPC permeability barrier. Osmotic balance

requires a non-permeable osmolyte to balance the water concentration inside and outside the nucleus, thus preventing swelling.

We added reference for the TB composition. In our previous work (ref 41 in the main text), we observed that the nuclear envelop could be prevented from swelling by adding 5mg/mL PEG6000 during digitonin treatment.

line 615 – what was the incubation time before imaging dextran permeability?

We added the incubation time before imaging dextran permeability as 10 min.

lines 616-618 – references for purification of importin beta, Ran and NTF2?

We added the references accordingly.

line 701 – why is the refractive index = 1.52 for droplets at the surface? The refractive index is for the medium in which the fluorophores are sitting for the measurement (not, presumably, for the glass that is nearby). What manufacturer's datasheet?

We thank the reviewer for point that out. We now take the refractive index=1.426 for in vitro condensate as measured in ref 64, and recalculated R_0 and R_e accordingly. There is no substantial change to the data or conclusions, as there is no strong dependency on refractive index.

line 710 – 'expected' is better than 'supposed'

We have revised accordingly.

lines 709-717 – the theoretical maximum anisotropy is 0.4 when the excitation and emission dipoles are parallel. This is normally a pretty safe assumption, but not always. Significant deviation could lead to a maximum anisotropy of 0.28 (e.g., angle of 27 degrees), in which case the anisotropy measurement would not support free rotation. It would be best to note that they are assuming parallel (or near parallel) excitation and emission dipoles.

We have revised accordingly.